# Structural and electronic features enabling delocalized charge-carriers in CuSbSe$_2$

Yuchen Fu[1,14], Hugh Lohan[1,2,14], Marcello Righetto [3], Yi-Teng Huang [1,4], Seán R. Kavanagh [5], Chang-Woo Cho [6,7], Szymon J. Zelewski [4,8], Young Won Woo[2,9], Harry Demetriou [2,10], Martyn A. McLachlan [11], Sandrine Heutz [2,10], Benjamin A. Piot [6], David O. Scanlon [12], Akshay Rao [4], Laura M. Herz [3,13], Aron Walsh [2] & Robert L. Z. Hoye [1] ✉

Inorganic semiconductors based on heavy pnictogen cations (Sb$^{3+}$ and Bi$^{3+}$) have gained significant attention as potential nontoxic and stable alternatives to lead-halide perovskites for solar cell applications. A limitation of these novel materials, which is being increasingly commonly found, is carrier localization, which substantially reduces mobilities and diffusion lengths. Herein, CuSbSe$_2$ is investigated and discovered to have delocalized free carriers, as shown through optical pump terahertz probe spectroscopy and temperature-dependent mobility measurements. Using a combination of theory and experiment, the critical enabling factors are found to be: 1) having a layered structure, which allows distortions to the unit cell during the propagation of an acoustic wave to be relaxed in the interlayer gaps, with minimal changes in bond length, thus limiting deformation potentials; 2) favourable quasi-bonding interactions across the interlayer gap giving rise to higher electronic dimensionality; 3) Born effective charges not being anomalously high, which, combined with the small bandgap ($\leq 1.2$ eV), result in a low ionic contribution to the dielectric constant compared to the electronic contribution, thus reducing the strength of Fröhlich coupling. These insights can drive forward the rational discovery of perovskite-inspired materials that can avoid carrier localization.

Semiconductors based on heavy pnictogens (namely Sb$^{3+}$ and Bi$^{3+}$) have gained a surge of interest over the past few years because of their potential to replicate the defect tolerance of lead-halide perovskites (LHPs), whilst overcoming their toxicity and stability limitations[1–6]. Defect tolerance is the ability of semiconductors to achieve low non-radiative recombination rates and maintain high mobilities despite high defect densities, and it is believed that such an effect occurs in LHPs in part because of its unusual electronic structure at its band edges, which comes about from the strong contributions of the Pb 6$s^2$ electrons to the valence band density of states[7]. As such, there has been a focus on compounds based on heavy post-transition metal cations In$^+$, Sn$^{2+}$, Sb$^{3+}$, and Bi$^{3+}$, which

have valence $ns^2$ electrons and, unlike Pb, are fully compliant with regulations on elements that can be safely used in consumer electronics[8]. Heavier cations are preferable, as spin-orbit coupling increases with effective nuclear charge, which results in a smaller bandgap, thereby increasing the chances of dominant defects forming shallow traps[7]. Among these four cations, Sb$^{3+}$ and Bi$^{3+}$ are especially promising because they are not severely limited in supply or expensive (unlike In)[9], and their valence $ns^2$ electron pair is stable (unlike Sn$^{2+}$)[2]. Indeed, many Sb- and Bi-based inorganic semiconductors have been found to be more environmentally and thermally stable than LHPs[10–16], and have also been found to avoid the self-doping that is prevalent in Sn perovskites[17–20].

Early work developing solar absorbers from these heavy pnictogen-based compounds focussed primarily on their charge-carrier lifetimes, in addition to their bandgaps and absorption coefficients, with the assumption that there was no significant difference in the mobilities between these materials[1]. Surprisingly, some Bi-based thin film materials were found to exhibit lifetimes in the hundreds of nanoseconds to microseconds range[12,21,22], far exceeding the minority carrier lifetimes of conventional inorganic semiconductors (1–10 ns) or LHPs (~100 ns in polycrystalline thin films)[23]. Recently, it was realized that this slow long-time decay in the population of photogenerated charge-carriers came about from carrier localization, in which the wavefunctions of charge-carriers or excitons are confined to within a unit cell or smaller, leading to the formation of small polarons or self-trapped excitons[24,25]. Carrier localization substantially reduces mobilities and, therefore, limits diffusion lengths, despite the slow decay in the population of excitations[12,24,25]. For example, although $Cs_2AgBiBr_6$ halide elpasolites have photogenerated charge-carriers decaying with a time constant in the microsecond range, steady-state mobilities only reach up to ~10 $cm^2 \cdot V^{-1} \cdot s^{-1}$ in single crystals[26]. Electron diffusion lengths as short as 30 nm have been found in polycrystalline $Cs_2AgBiBr_6$, which partly accounts for the low photovoltaic power conversion efficiencies (PCEs) that are usually well below 4%[27]. Recent investigations into the wider family of bismuth-halide and bismuth-chalcogenide semiconductors have found carrier localization to be so prevalent that it is being described as a hallmark of these materials[10–12,14,28–31]. The effect of carrier localization on Sb-based compounds is not as well established. One of the best-studied of these materials is the antimony chalcogenide family of compounds ($Sb_2S_3$ and $Sb_2Se_3$). There are currently strong disagreements in the community regarding whether self-trapping occurs in these materials, limiting open-circuit voltages up to a maximum of 0.8 V in the case of $Sb_2S_3$[30,32–34], or whether the performance is instead limited by charged defects[35–37]. In $Cs_2AgSbBr_6$, on the other hand, charge-carrier localization proceeds on a picosecond timescale, similar to that in $Cs_2AgBiBr_6$, with alloying of the two materials exacerbating such effects, owing to localized charge-carriers being more susceptible to energetic disorder[38].

It is clear that the future development of pnictogen-based perovskite-inspired materials for optoelectronic devices urgently requires not only consideration of defects, but also insights into how charge-carrier localization may be avoided in these materials. Very recently, we provided hints in this direction with detailed spectroscopic and computational investigations into bismuth oxyiodide (BiOI)[39,40]. BiOI exhibits a red-shift in the photoluminescence (PL) spectrum compared to its optical bandgap, which would typically be considered to arise from self-trapping. However, we found that this red-shift can be fully accounted for by the coupling between charge-carriers and two longitudinal optical (LO) phonon modes generated coherently through photoexcitation. The delocalized nature of these large polarons was verified from computations of the wavefunction of the lowest-energy exciton, as well as magneto-optical spectroscopy measurements. The mobilities reached up to 83 $cm^2 \cdot V^{-1} \cdot s^{-1}$ in the in-plane direction, exceeding the mobilities of self-trapped carriers (typically ~10 $cm^2 \cdot V^{-1} \cdot s^{-1}$ or lower)[10–12,24–26,28]. In separate recent work, we showed through optical pump terahertz probe (OPTP) spectroscopy measurements of thin film samples that BiOI avoids charge-carrier localization in both the in-plane and out-of-plane directions[40]. Therefore, unlike most novel bismuth-halide semiconductors, BiOI exhibits band-like transport. We speculated that this was related not only to its layered nature, but also the large thickness of each layer[39], which could contribute to the delocalization of excitations. However, the detailed mechanisms, as well as the role of acoustic phonons and how they interact with charge-carriers, are yet to be determined.

Inspired by this recent work, herein we investigate a related layered Sb-based compound, $CuSbSe_2$. This material is a příbramite, which is the Se analog to the chalcostibite $CuSbS_2$, and has

experimentally- and computationally-determined bandgaps in the range of 0.9–1.2 eV[41–46]. This is smaller than the bandgaps found for most Sb- and Bi-based perovskite-inspired materials recently investigated (Supplementary Table 1), and is suitable for harvesting the near-infrared portion of the solar spectrum, which is a substantial fraction of the energy in the AM 1.5 G spectrum[47]. More broadly, the $ABZ_2$ family of materials (A = monovalent cation, B = $Sb^{3+}$ or $Bi^{3+}$, Z = chalcogen) have gained attention as promising pnictogen-based semiconductors. This is because $AgBiS_2$ photovoltaics recently reached a certified PCE of 8.85%[14], which is among the highest for any Bi-based solar absorber. Both $AgBiS_2$, and the related $NaBiS_2$ compound, were found to be stable in air and have slow decays in their photoexcited charge-carriers[12–14]. Our detailed investigations into $NaBiS_2$ showed that this was due to carrier localization, which was facilitated by localized S 3$p$ states that form in regions with high coordination of Na around S, likely capturing holes and leading to the formation of small hole polarons[12,48]. Recently, the presence of charge-carrier localization in $AgBiS_2$ was also reported, and the degree of localization tunable through cation disorder engineering[49]. $CuSbSe_2$ avoids the cation disorder found in both $NaBiS_2$ and $AgBiS_2$ owing to the $Cu^+$ and $Sb^{3+}$ cations having sufficiently different radii (60 pm and 76 pm, respectively)[50], as well as the stereochemical activity of the lone pair on $Sb^{3+}$, such that the smaller $Cu^+$ occupies tetrahedral sites, while $Sb^{3+}$ occupies trigonal pyramidal sites (which allows the lone pair on $Sb^{3+}$ to be projected out into space). Furthermore, the thickness of each layer in $CuSbSe_2$ (5.70 Å) is comparable to that of BiOI (6.14 Å)[39]. There is, therefore, a possibility that $CuSbSe_2$ may be able to avoid the charge-carrier localization found in $NaBiS_2$, $AgBiS_2$, and most Bi-halide compounds, and if so, the mechanism by which this occurs will be of paramount importance to learn how delocalized excitations can be achieved more broadly across the wider family of pnictogen-based perovskite-inspired materials.

In this work, we developed a novel thiol-amine-based solution-processing route to achieve phase-pure $CuSbSe_2$ thin films. The optical phonon modes present were determined through Raman and infrared (IR) spectroscopy, and the nature of excitations (i.e., whether free charge-carriers or excitons formed) was determined through Elliott model fitting of the optical absorption spectra, and correlated with computations of the exciton binding energy. To understand whether these excitations are localized, the charge-carrier kinetics were measured by transient absorption spectroscopy (using a femtosecond pulsed excitation laser), and photoconductivity transients by OPTP spectroscopy, along with measurements of temperature-dependent mobility. The underlying mechanisms behind the nature of excitations were established through calculations of the strength of coupling with acoustic phonons (acoustic coupling constant) and LO phonons (Fröhlich coupling constant), as well as calculations of the key parameters that influence these coupling constants, namely the deformation potential, dielectric tensor, bonding/anti-bonding nature of crystal orbitals at the band extrema, changes in bond lengths and interlayer spacing arising from distortions, as well as the Born effective charges the elements present. The understanding gained from investigating the case of $CuSbSe_2$ can provide insights into how we could design heavy pnictogen-based semiconductors with band-like transport, which will be critical for creating more promising earth-abundant solar absorbers.

## Results

### Structure, synthesis, and vibrational properties of $CuSbSe_2$ thin films

$CuSbSe_2$ has a layered structure (Fig. 1a), with an orthorhombic unit cell that is similar to that of chalcostibites (*Pnma* space group)[51,52]. The $CuSbSe_2$ layers are held together by van der Waals interactions. Each Sb atom is bonded to three Se atoms in a trigonal pyramidal geometry, while each Cu atom is bonded with four Se atoms in a

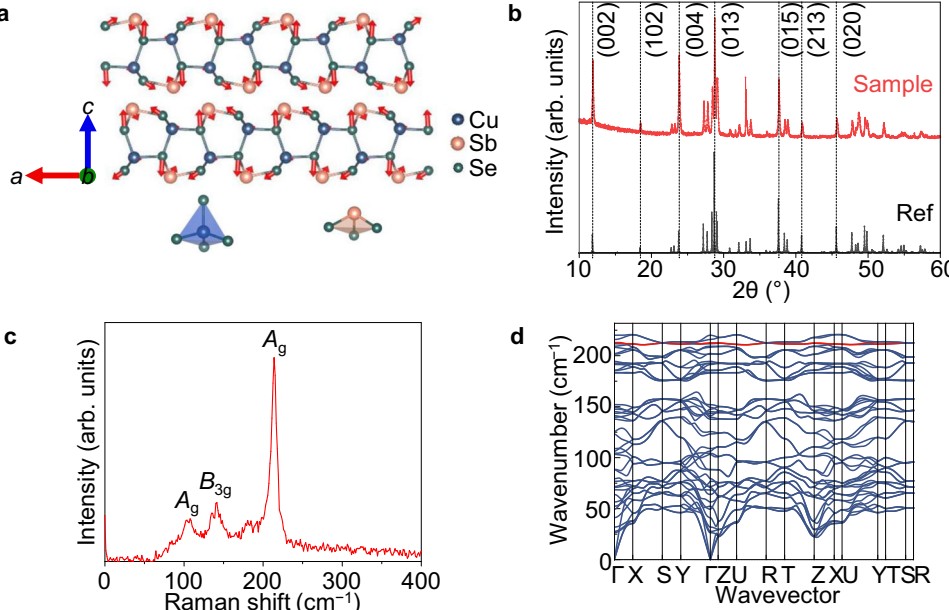

**Fig. 1 | Structural and phonon properties of CuSbSe₂. a** Crystal structure of CuSbSe₂, viewed along the *b* axis, and with the dominant $A_g$ Raman mode shown in red arrows. The bonding environments of Cu (blue spheres) and Sb (pink spheres) are illustrated below the crystal structure. Se represented by green spheres. **b** X-ray diffraction (XRD) pattern of solution-processed thin films compared with the reference pattern of CuSbSe₂ (ICSD database, ID 418754; detailed fitting in Supplementary Fig. 1a, b). The Miller indices of the dominant peaks are indicated. **c** Raman spectrum of spin-coated CuSbSe₂ thin film with phonon modes of the most intense peaks labeled. Spectrum is the average of 10 scans. **d** Phonon dispersion curve of CuSbSe₂. The band containing the dominant $A_g$ mode is highlighted in red.

tetrahedral arrangement. The CuSe₄ tetrahedra and SbSe₃ trigonal pyramids share corners (of Se). The distance between Sb and Se atoms separated by the interlayer gap (3.26 Å) is too large to form full covalent bonds between these atoms. From the structure shown in Fig. 1a, it can be seen that there is static stereochemical activity of the $5s^2$ lone pair on $Sb^{3+}$[53–68]. This stereochemical activity is also found in CuSbS₂[69–71], and occurs because the Sb 5s and chalcogen valence *p* orbitals are close enough in energy for mixing, such that a second-order Jahn-Teller distortion can occur[72]. Indeed, the Sb 5s lone pair is also stereochemically active in Sb₂Se₃[72,73], indicating that the orbital energy levels of the Sb 5s and Se 4p states are close enough in energy to interact.

Previous efforts at growing CuSbSe₂ focussed on vacuum-based approaches (e.g., sputter deposition[74], close-space sublimation[75]), methods that have long reaction times (e.g., fusion method[76] or selenization of metal precursors[77]), or processes involving the use of toxic precursors (e.g., hydrazine solvent[41,78]). Solution-processing is advantageous in requiring less capital-intensive equipment than vacuum-based processing[79–81], but at the same time, it is critical to avoid the use of toxic solvents[82]. More recently, a more benign solvent system than hydrazine, comprised of a thiol-amine mixture, has been found to be effective in dissolving chalcogenide precursors and successfully used to deposit absorber layers of photovoltaic devices, such as Cu₂ZnSn(S,Se)₄[83,84], Cu(In, Ga)Se₂[85,86], and CuIn(S, Se)₂[87]. In this work, we developed the synthesis of phase-pure CuSbSe₂ thin films by this novel thiol-amine-based solution-processing route for the first time, as detailed in Methods. To achieve crystalline films, we dried the films at 100 °C for 2 min in an N₂-filled glovebox, before crystallizing at 400 °C for 2 min in a tube furnace filled with Ar (-1200 mTorr pressure). The details of the optimization of the thiol-amine processing route for CuSbSe₂ are in Supplementary Note 1. Pawley fitting of the X-ray diffraction (XRD) pattern of these films with the reference pattern for CuSbSe₂ (ICSD database, coll. code 418754) showed that all measured peaks were accounted for by the příbramite phase (Fig. 1b and details in Supplementary Fig. 1).

Raman and Fourier-transform infrared (FTIR) spectroscopy were employed to determine the dominant optical phonon modes present in CuSbSe₂. For the *Pnma* space group ($D_{2h}^{16}$), there are four Raman-active mode symmetries ($A_g$, $B_{1g}$, $B_{2g}$, and $B_{3g}$), along with three IR active mode symmetries ($B_{1u}$, $B_{2u}$, and $B_{3u}$)[41,88]. In the Raman spectrum measured from the thin film samples (Fig. 1c), three obvious peaks at $105.7 \pm 0.2$ cm⁻¹ ($A_g$), $141.7 \pm 0.6$ cm⁻¹ ($B_{3g}$), and $213.7 \pm 0.2$ cm⁻¹ ($A_g$) can be observed, and all of them have been reported to come from the příbramite phase of CuSbSe₂[74,89–91]. The results agree with our calculated phonon spectrum for CuSbSe₂ (Fig. 1d and Supplementary Table 3). In the FTIR spectrum (Supplementary Fig. 2), two relatively strong peaks centered at $182.8 \pm 0.2$ cm⁻¹ and $223.1 \pm 0.1$ cm⁻¹ were observed. According to our calculations, these two peaks can be assigned to the $B_{2u}$ and $B_{3u}$ modes, respectively. These Raman and FTIR measurements are, therefore, consistent with the phase-purity of the spin-coated CuSbSe₂ thin films prepared after the heat treatment at 400 °C for 2 min.

It is worth noting that the intensity of the $A_g$ mode at ≈213 cm⁻¹ is much higher than other Raman-active modes (Fig. 1c). We also constructed the phonon dispersion curve (Fig. 1d), and the band of the dominant $A_g$ mode at $213.7 \pm 0.2$ cm⁻¹ is highlighted. The vibrations associated with this mode are calculated and illustrated by the red arrows in Fig. 1a, showing this to be an intralayer breathing mode. The phonon density of states is shown in Supplementary Fig. 2 and compared with the FTIR and Raman spectra.

Finally, we note that CuSbSe₂ contains Cu in the +1 oxidation state, whereas the +2 oxidation state is typically more thermodynamically stable under ambient conditions. We therefore examined the chemical stability of the cation species in the optimized CuSbSe₂ films by X-ray photoelectron spectroscopy, as detailed in Supplementary Note 3. We found from the Cu 2p core levels and LMM Auger peaks that Cu remained in the +1 oxidation state after storage in ambient air (with -80% relative humidity) for 3 weeks, and Sb also remained in the +3 oxidation state. However, we found that a layer of oxide (likely Sb₂O₃) formed on the surface of the films after storage in air, whereas there

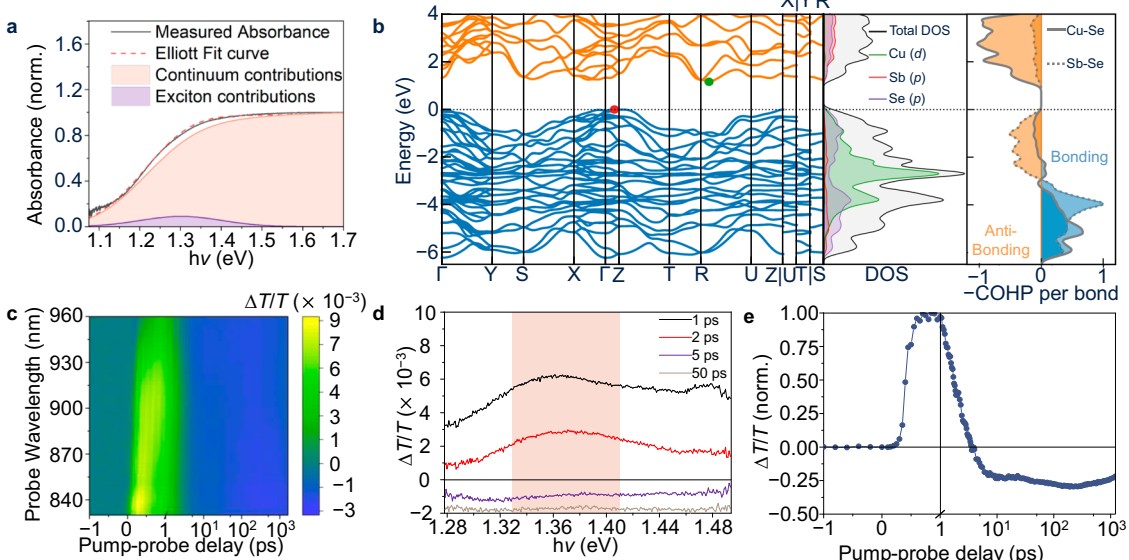

**Fig. 2 | Optical and electronic properties of CuSbSe₂. a** Comparison between the measured optical absorbance curve (black solid line) and fit with the Elliott model (red dashed line). The contributions from the exciton and continuum to the optical absorption spectrum are represented by the areas shaded purple and pink, respectively. **b** Electronic band structure of CuSbSe₂ (left panel; the highest occupied state set to 0 eV), along with the electronic density of states curves (middle panel), and crystal orbital Hamilton population (COHP) diagram (right panel). The bonding and anti-bonding interactions are represented by blue and orange, respectively, in the right panel. The red circle in the left panel indicates the valence band maximum, while the green circle indicates the conduction band minimum. **c** Short-time transient absorption (TA) signal color map of CuSbSe₂ films excited by an 800 nm wavelength pump (150 fs pulse width, 41 μJ cm⁻² pulse⁻¹ fluence, with 500 Hz repetition rate), along with **d** short-time TA spectra for pump-probe delays of 1, 2, 5, and 50 ps, and **e** its normalized ground state bleach (GSB) signal kinetics. The GSB kinetics were acquired by averaging the signals from 1.33 to 1.41 eV (pink shaded area in **d**) and normalized to the maximum $\triangle T/T$ value. Data shown for parts (**d**, **e**) are the average of five scans.

was no evidence of cuprous oxide or hydroxide species, showing Cu(I) to remain stable in its tetrahedral environment in the structure. We also found CuSbSe₂ to be phase stable in ambient air over this 3-week period (Supplementary Fig. 6a), and is also more stable under damp-heat conditions (85% relative humidity, 85 °C temperature, and under 1-sun illumination) than lead-halide perovskites (Supplementary Fig. 6b).

**Optoelectronic properties of CuSbSe₂**

Having developed phase-pure samples and understood the dominant phonon modes in CuSbSe₂, we next needed to understand the nature of excitations and their kinetics. The black solid line in Fig. 2a shows the measured optical absorbance curve of CuSbSe₂, and the electronic structure is shown in Fig. 2b. The fit to the optical absorption spectrum (red dashed line in Fig. 2a) was obtained from Elliott's theory[92], following a previously reported procedure[93], while the absorbance spectrum of CuSbSe₂ over a wider photon energy range is illustrated as Supplementary Fig. 7a. We note that despite a significant lineshape broadening ($\Gamma \sim 90$ meV), the fit matches with the measured spectrum well (Fig. 2a). The deconvolution of the excitonic and continuum contributions yields a weak and broad excitonic contribution, described by an exciton binding energy ($E_b$) of 9 ± 4 meV. This matches well with the density functional theory (DFT) calculations we made on CuSbSe₂, from which we obtained an $E_b$ of 8.7 meV, as estimated using the Wannier-Mott hydrogenic model[94]. Given that these $E_b$ values are well below k$T$ at room temperature (26 meV), we would expect CuSbSe₂ to predominantly exhibit free charge-carriers rather than bound excitons.

We also note that the absorbance curve shows a shoulder at ~1.4 eV (details in Supplementary Fig. 7b), which could either arise from excitons or from the electronic structure of the material. To distinguish between these two possibilities, we computed the optical absorption spectrum (Supplementary Fig. 7c) from the frequency-dependent dielectric tensor using hybrid DFT (HSE06 functional)[95].

The computed absorption spectrum reproduced the experimentally observed shoulder in absorption. Our calculations were carried out in the independent particle approximation[96], and therefore phonon-assisted transitions and polaronic/excitonic effects were not considered. This analysis shows that the shoulder in the absorption spectrum of CuSbSe₂ arises because of its electronic structure.

To understand the kinetics of the free charge-carriers in CuSbSe₂, short-time transient absorption (TA) spectroscopy was employed. In short-time TA measurements, the sample was excited with the 800 nm wavelength pulsed laser. After excitation, probe pulses, comprising a broadband near-IR spectrum, were used to measure the relative change in transmittance ($\Delta T/T$) of the sample at certain delays after pumping, with pump-probe delays ranging from 1 to 1000 ps. The positive ground state bleach (GSB) signal on a $\Delta T/T$ scale is usually proportional to the photoexcited carrier population near the band edges. The decay in the GSB signal, therefore, reflects the depopulation of charge-carriers near the band edges. Meanwhile, negative photo-induced absorption (PIA) can also occur, and the possible origins of PIA include self-trapping, absorption related to defect states or excitation to higher energy states. Strong PIA signals can interfere with the GSB signals. The results of short-time TA measurements are shown in Fig. 2c–e. In the short-time TA spectrum (Fig. 2d), we can observe a broadband GSB signal centered at approximately 1.36 eV. However, the positive GSB signal was pulled down by a strong PIA signal within 5–10 ps, which can also be observed in the normalized TA signal kinetics (Fig. 2e). The strong PIA signal makes it difficult to estimate the charge-carrier lifetime of CuSbSe₂ films. As for long-time TA measurements (355 nm wavelength pump, with pump-probe delays ranging from 1 to 1000 ns), the GSB signal was completely suppressed by a PIA signal, and no GSB signal could be observed (Supplementary Fig. 9a–c). The strong PIA signal in TA measurements made it necessary to use other techniques to better understand charge-carrier kinetics in CuSbSe₂. Nevertheless, the breadth of the GSB observed in the short-time TA measurements, along with the absence of PIA on

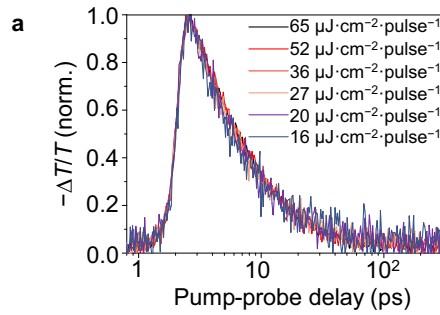

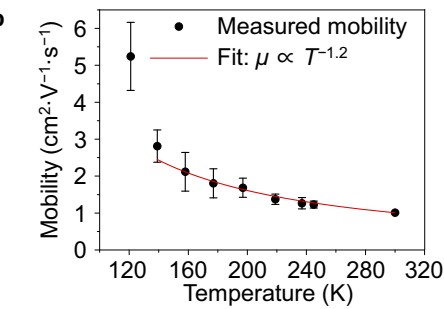

**Fig. 3 | Spectroscopic and temperature-dependent studies on carrier-phonon coupling in CuSbSe₂. a** Normalized comparison between the fluence-dependent optical pump terahertz probe (OPTP) transients measured for CuSbSe₂ thin films following 400 nm wavelength pulsed excitation. **b** Temperature-dependent mobility of CuSbSe₂ thin films determined using Hall effect measurements, fit using a power law model that indicates $\mu \propto T^{-1.2}$. The point at the lowest temperature (121 K) is not included in the fit due to its higher standard deviation than the other data points. The error bars represent the standard deviation between two samples, which were prepared and measured with identical parameters.

either side initially, is consistent with these excitations originating from charge-carriers rather than excitons.

## Experimental investigation into charge-carrier-phonon coupling in CuSbSe₂

To gain more in-depth insights into the nature of the excitations in CuSbSe₂, we employed optical pump terahertz probe (OPTP) measurements. The fractional change in transmitted terahertz (THz) field amplitude $(-\Delta T/T)$ is monitored in OPTP measurements with sub-picosecond time resolution following a 400-nm wavelength pulsed excitation (see Methods for details). The measured $-\Delta T/T$ signal is proportional to the photoconductivity $\Delta\sigma$ of the studied thin film, making it ideal for investigating charge-carrier localization processes. As demonstrated for several Bi-based semiconductors, the charge-carrier localization process yields a photoconductivity decay on the sub-picosecond timescale[10,12,28], as a result of the lower mobility of localized charge-carriers. By comparison, defect-assisted trapping would cause a slower decay in the photoconductivity because charge-carriers need to diffuse to the defect states before they are trapped (reducing mobility), or undergo non-radiative recombination (reducing the photoexcited charge-carrier population)[10]. The different timescales of photoconductivity decay can provide insights into the trapping mechanisms inside materials, especially since free charge-carriers rather than excitons form in CuSbSe₂ (see end of Supplementary Note 4 for details).

In the case of CuSbSe₂, we found that 50% of the original OPTP signal was lost over a period of 6.7 ps, while taking 50 ps for 92% of the original signal to be lost (Fig. 3a). This is a slower decay than observed from short-time TA measurements (Fig. 2e), likely because the GSB kinetics had the rise in PIA superimposed upon it, obscuring the real decay of the photogenerated charge-carriers. The decay observed in the OPTP measurements is significantly slower than previously reported for other bismuth-based semiconductors, which have charge-carrier dynamics dominated by localization processes. For example, for Cs₂AgBiBr₆, Cu₂AgBiI₆, NaBiS₂, and non-heat treated AgBiS₂ with inhomogeneous cation disorder, 50% of the peak OPTP signal was lost after only 0.5–2 ps (Supplementary Table 1)[10–12,28]. The OPTP kinetics for CuSbSe₂ are, therefore, consistent with delocalized free charge-carriers diffusing to defect states and undergoing non-radiative recombination, rather than undergoing carrier localization.

To verify that delocalized large polarons instead of small polarons form in CuSbSe₂, we measured the temperature dependence of the mobility through Hall effect measurements. If large polarons dominate charge-carrier transport, then the overall mobility should decrease as the temperature increases because increasing temperature can lead to more phonons that charge-carriers can couple to[25,97–99]. On the contrary, small polarons show an increase in mobility with increasing temperature, since small polarons can only hop between lattice sites, and increasing the temperature provides more thermal energy for hopping transport[10,28,97,98].

In this study, we measured the Drude mobilities at different temperatures ranging from 120 to 300 K. Samples were prepared in the van der Pauw configuration, with gold contacts evaporated on the four corners. Figure 3b shows the measured temperature-dependent mobility of CuSbSe₂ films. A clear decrease in mobility with an increase in temperature was observed, which is consistent with the behavior of large polarons. We fit a power law model to the mobility data, and found the exponent to be –1.2 (Fig. 3b), however, interpreting the scattering mechanism based on this exponent has recently been revealed as not straightforward[100]. Resistivities also decreased with increasing temperature (Supplementary Fig. 8), due to an increase in carrier concentration as more charge-carriers were thermally excited across the bandgap. Importantly, the mobilities measured in these CuSbSe₂ films are higher than typically found for small polarons. Hall effect measurements gave a macroscopic mobility of $1.01 \pm 0.01\ \mathrm{cm^2 \cdot V^{-1} \cdot s^{-1}}$ at room temperature (Fig. 3b, 300 K), while the delocalized intra-grain mobility value extracted from the initial photoconductivity of OPTP measurements was $4.7 \pm 0.2\ \mathrm{cm^2 \cdot V^{-1} \cdot s^{-1}}$ (refer to Supplementary Note 5 for details of how the mobility was determined from OPTP measurements). This difference in mobility values can be attributed to the different length scales of the Hall effect and OPTP measurements. Hall effect measurements investigate charge-carrier transport throughout the whole sample, while the mobility extracted from the initial OPTP signal represents the transport within a shorter range, usually well within one grain[25,27,28,48].

Even though the mobility values extracted over different length scales are different, comparing mobilities obtained from different materials over the same length scale is informative to put the nature of charge-carrier transport in context. BiOI, as another pnictogen-based compound which can avoid charge-carrier localization, has a peak OPTP mobility of ~3 $\mathrm{cm^2 \cdot V^{-1} \cdot s^{-1}}$ at 295 K for polycrystalline samples. In single crystals, the mobilities for BiOI obtained from time-of-flight measurements are 26 $\mathrm{cm^2 \cdot V^{-1} \cdot s^{-1}}$ (out-of-plane) and 83 $\mathrm{cm^2 \cdot V^{-1} \cdot s^{-1}}$ (in-plane)[39]. By contrast, materials undergoing carrier localization (e.g., NaBiS₂, Cs₂AgSbBr₆, Cs₂AgBiBr₆ and AgBiS₂ without heat treatment) exhibit a substantial reduction in mobility from an initial delocalized state to localized mobilities in the range of 0.03–1.3 $\mathrm{cm^2 \cdot V^{-1} \cdot s^{-1}}$ 2 ps after excitation, as obtained from OPTP measurements (Supplementary Table 1). These OPTP mobilities are all lower than that for CuSbSe₂, despite CuSbSe₂ having higher effective masses (Supplementary Table 1). This is consistent with the delocalized nature of charge-carriers in CuSbSe₂, and emphasizes the importance of avoiding carrier localization.

**Table 1 | Calculated properties related to carrier-phonon coupling in $CuSbSe_2$ along different principal axes**

|  | a | b | c | Average[a] |
|---|---|---|---|---|
| $a_o$ (Å) | 6.457 | 4.034 | 14.929 |  |
| $E_d^{VBM}$ (eV) | 1.16 | 1.93 | 2.11 | 1.73 |
| $E_d^{CBM}$ (eV) | 6.60 | 6.32 | 6.62 | 6.51 |
| $C_{iii}$ (GPa) | 75.5 | 81.7 | 60.4 | 41.6 |
| $g_{ac}^{VBM}$ | $1 \times 10^{-3}$ | $3 \times 10^{-3}$ | $3 \times 10^{-3}$ | $2 \times 10^{-3}$ |
| $g_{ac}^{CBM}$ | $7 \times 10^{-3}$ | $1.0 \times 10^{-2}$ | $1.0 \times 10^{-2}$ | $9 \times 10^{-3}$ |
| $\epsilon_\infty$ | 10.1 | 12.5 | 11.4 | 11.3 |
| $\epsilon_{stat}$ | 12.0 | 40.4 | 16.5 | 23.0 |
| $m_h^*$ | 1.44 | 1.30 | 2.38 | 1.60 |
| $m_e^*$ | 0.29 | 0.41 | 0.94 | 0.43 |
| $\alpha_h$ | 0.55 | 1.77 | 1.17 | 1.59 |
| $\alpha_e$ | 0.25 | 0.99 | 0.73 | 0.82 |
| $E_b$ (meV) | 22.3 | 2.6 | 33.6 | 8.7 |

[a]For details on how averaging for each quantity was carried out, see Supplementary Note 8. $a_o$: lattice parameter; $E_d^{VBM}$: acoustic deformation potential of the valence band maximum; $E_d^{CBM}$: acoustic deformation potential of the conduction band minimum; $g_{ac}$: acoustic coupling constant; $C_{iii}$: Diagonal component of the elastic tensor $\epsilon_\infty$: dielectric constant at high frequency; $\epsilon_{stat}$: static dielectric constant; $m_h^*$: effective mass of holes (related to electronic conductivity); $m_e^*$: effective mass of electrons (related to electronic conductivity); $\alpha_h$: Fröhlich coupling constant of holes; $\alpha_e$: Fröhlich coupling constant of electrons. $E_b$: Wannier-Mott binding energies.

## Theoretical insights into charge-carrier-phonon coupling in $CuSbSe_2$

Having experimentally demonstrated an absence of charge-carrier localization in $CuSbSe_2$, which is unusual compared to most recently investigated pnictogen-based perovskite-inspired materials[10,12,24,25], we aim now to verify this computationally and establish the underlying factors enabling this behavior.

To understand the delocalized nature of charge-carriers, we used deformation potential theory[101], which aims to describe the effect that a long-wavelength acoustic wave has on the electronic structure of a semiconductor as it propagates through the material. The core assumption of this theory is that the wavelength of the propagating wave is large compared to the characteristic size of the unit cell, and so the propagating wave can be described by a homogenous strain (please refer to Supplementary Fig. 11). We can use this assumption to describe the first-order scattering potential of any long-wavelength acoustic phonon as the change in band edge position as we apply a strain to a structure via a quantity known as the acoustic deformation potential ($E_d^{n\mathbf{k}}$), which is described by Eq. 1:

$$E_d^{n\mathbf{k}} = \frac{\delta \epsilon_{n\mathbf{k}}}{\delta \mathbf{S}_{\alpha\beta}} \qquad (1)$$

In Eq. 1, $\epsilon_{n\mathbf{k}}$ is the energy of band $n$ at wavevector $\mathbf{k}$, and $\mathbf{S}_{\alpha\beta}$ is the uniform stress tensor[99]. Please refer to Supplementary Note 7 for more details on how we calculated the acoustic deformation potential, especially Supplementary Fig. 12 and the associated discussion beneath it. The values in Table 1 are calculated self-consistently with a fixed internal structure using the HSE06 exchange-correlation function. The average values for the deformation potentials of $CuSbSe_2$ (1.73 eV for VBM; 6.51 eV for CBM, as shown in Table 1) are much lower than those of $Cs_2AgBiBr_6$ (13.7 eV for VBM; 14.7 eV for CBM), which undergoes charge-carrier localization, and comparable to the values of $CsPbBr_3$ (2.2 eV for VBM; 6.3 eV for CBM), which has delocalized charge-carriers[10].

Deformation potential theory successfully describes phonon-limited mobility in materials where acoustic phonon scattering is dominant, but fails to describe self-trapping due to polaron formation,

a problem addressed by the continuum model of ref. 102. Building on foundations laid by Bardeen and Shockley, Toyozawa predicted a discontinuous and large increase in the effective mass of a free carrier (a so called "self-trapped" state) in a 2D material where a dimensionless coupling constant exceeds unity. This constant is known as the acoustic coupling constant ($g_{ac}$), and is given by Eq. 2:

$$g_{ac} = \frac{E_d^2}{Ca_0} \cdot \frac{m}{3\pi\hbar^2} \qquad (2)$$

where $E_d$ is the acoustic deformation potential, $C$ the elastic constant, $a_o$ the lattice parameter, $m$ the mass of the charge-carrier considered and $\hbar$ the reduced Planck's constant. For values much less than one, we do not expect localization due to acoustic coupling. Charge-carrier localization can be expected even in stiff materials if they have large deformation potentials, due to the square proportionality seen in Eq. 2. Using this model, we can see that low acoustic deformation potentials are consistent with polarons being large in $CuSbSe_2$.

The strength of coupling between charge-carriers and longitudinal optical (LO) phonons is described by the Fröhlich coupling constant, $\alpha$, given by Eq. 3.

$$\alpha = \frac{e^2}{4\pi\epsilon_0} \left( \frac{1}{\epsilon_\infty} - \frac{1}{\epsilon_{stat}} \right) \sqrt{\frac{m^*}{2\omega_{LO}\hbar^3}} \qquad (3)$$

In Eq. 3, $\epsilon_0$ is the vacuum permittivity while $\epsilon_\infty$ and $\epsilon_{stat}$ are the calculated optical and static dielectric constants, respectively. $m^*$ is the (conductivity) effective mass of the free charge-carrier considered, while $\omega_{LO}$ is the effective longitudinal optical (LO) phonon frequency, and $\hbar$ is the reduced Planck's constant. The values of these properties are shown in Table 1. $\omega_{LO}$ is 138 $cm^{-1}$, and was calculated as an average over all Γ-point modes weighted by the dipole moment they produce (since Fröhlich coupling arises due to interactions between charge-carriers and optical phonon modes producing local dipoles)[99]. The average Fröhlich coupling constants of holes and electrons ($\alpha_h = 1.59$, $\alpha_e = 0.82$) are both in the weak regime, lower than those found in $ABZ_2$ materials like $NaBiS_2$ ($\alpha_h = 2.92$, $\alpha_e = 1.40$)[12], $AgBiS_2$ ($\alpha_h = 1.63$, $\alpha_e = 1.09$)[14], as well as methylammonium lead iodide perovskites (2–3)[24]. The low Fröhlich coupling constants are well below the range typically considered to be strong[103], showing that carrier localization due to coupling with LO phonons should not occur in $CuSbSe_2$.

We additionally performed state-of-the-art calculations using the ShakeNBreak method[104] to explicitly model polarons in $CuSbSe_2$ as dilute charges in a 64-atom supercell (four times the volume of a unit cell), as shown in Supplementary Fig. 10. To do this, we added an extra unpaired electron or hole to the unperturbed supercell, and allowed the system relax to a local minimum. By inspecting the charge densities of electrons and holes (representative of their wavefunctions) in the relaxed structures, we found that no localized states (i.e., 0D states confined to within a unit cell) occurred. Rather, the polaronic states were delocalized over the entire supercell, and would therefore have wavefunctions well exceeding a unit cell. This supports the conclusion that small electron and hole polarons do not form in $CuSbSe_2$. Note that other, more sophisticated methods exist to calculate electron-phonon coupling and model polarons (including ab initio molecular dynamics[105–107], AHC theory[108–110], Quantum Monte-Carlo simulations[111–113], the special displacement method[114–116], and the ab initio theory of polarons of ref. 117), but these are cutting-edge methods and go beyond the scope of this work. Nevertheless, fully investigating them in future work can lead to better quantitative agreement between theory and experiment, in addition to qualitative agreement.

In light of the results presented, it can be seen that carrier localization is not present in $CuSbSe_2$, and that this is due to the weak coupling between charge-carriers and both acoustic and optical

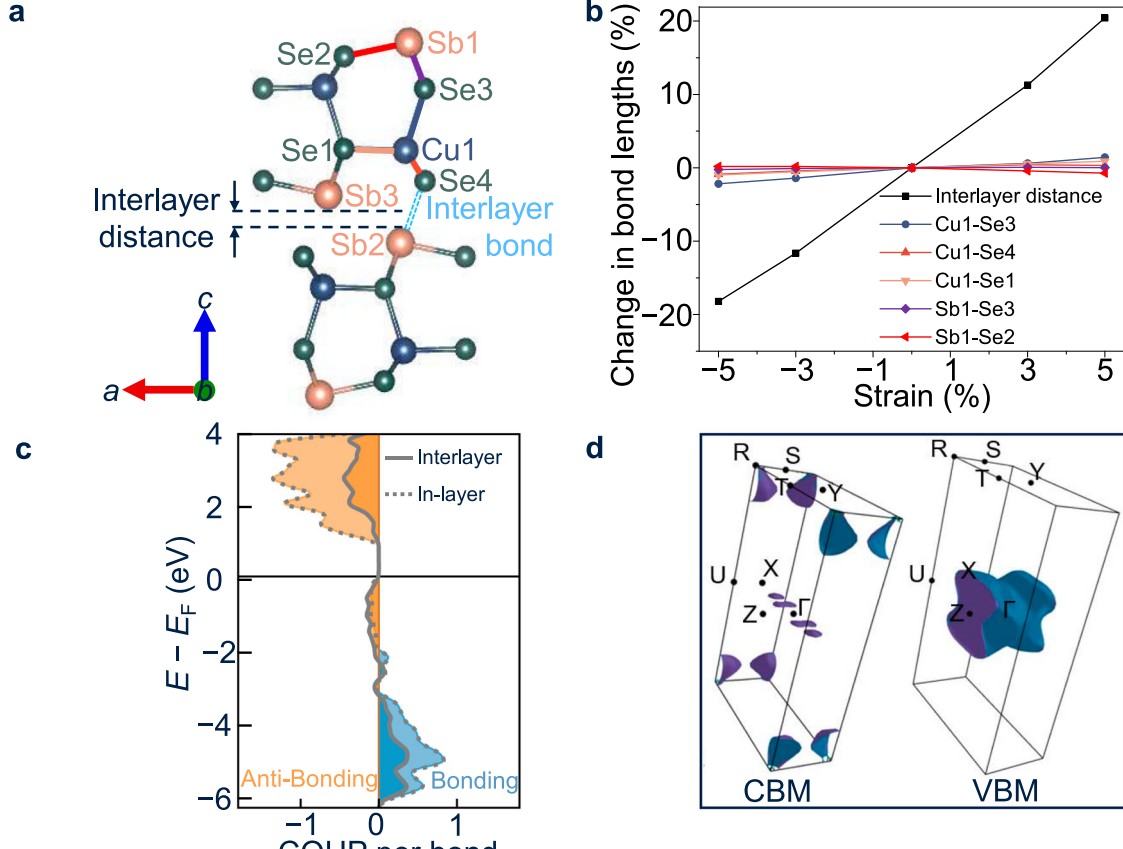

**Fig. 4 | Computational analysis of CuSbSe₂. a** Structure of CuSbSe₂, with key atoms labeled, and the interlayer distance defined as the perpendicular distance between Sb2 and Sb3. **b** Percentage changes in bond lengths and interlayer distance of CuSbSe₂ as a function of strain along the *c*-axis. All calculated bond lengths shown are after the relaxation of the atoms in the structure after distortion, i.e., calculations for equilibrated structures as shown (refer to discussion in Supplementary Note 7). A disproportionally large change in the interlayer distance is observed as compared to bond lengths for a given strain. **c** Calculated crystal orbital Hamilton population (COHP) per bond of in-layer (dash line) and interlayer (solid line) Sb-Se bonds. The bonding and anti-bonding interactions are represented by blue and orange, respectively. **d**, Fermi iso-surface 0.1 eV below the VBM (top figure) and above the CBM (bottom figures).

phonons. This is unusual among heavy pnictogen-based perovskite-inspired materials, and it is critical to unravel the underlying structural, electronic, and chemical factors. In the next three sub-sections, we will examine computationally how the crystal structure and bonding in CuSbSe₂ result in the low deformation potentials and low Fröhlich coupling constants, as well as the effect on the electronic dimensionality in both the valence and conduction bands.

**Understanding the cause of low deformation potentials in CuSbSe₂**

As mentioned previously, the low acoustic deformation potential $E_d$ of CuSbSe₂ is the prime factor in causing weak acoustic coupling. Since $E_d$ describes the change in band edge positions when the lattice is distorted, which is in turn influenced by the nature of bonding between atoms, analysis of the bonding environment can provide insights into the magnitude of the acoustic deformation potential.

The electronic structure of CuSbSe₂ is shown in the middle panel of Fig. 2b, with the orbital-projected density of states (pDOS) and band structure diagrams shown. On the far right of the same panel are crystal orbital Hamilton population (COHP) diagrams for Sb-Se and Cu-Se bonds, which allow us to determine whether these are bonding, anti-bonding or non-bonding interactions. With −COHP set as the horizontal axis, positive values represent bonding interactions, negative values indicate anti-bonding interactions between atoms, while values near the center line indicate non-bonding interactions. The more positive −COHP value of the Sb-Se bond reveals stronger

covalent bonding between Sb and Se atoms compared to Cu and Se atoms. We see the VBM of CuSbSe₂ mainly consists of Cu(*d*)-Se(*p*) anti-bonding states, with much weaker contributions from Sb(*s,p*)-Se(*p*), which are approximately non-bonding. These non-bonding states are the result of Sb *s*–*p* orbital mixing, which occurs through a second-order Jahn-Teller distortion, resulting in a familiar non-bonding lone-pair projected into the interlayer space.

From Table 1, it can be seen that there is more anisotropy in the deformation potentials at the VBM than CBM. This anisotropy in the VBM deformation potentials can be understood by considering the inequivalent distortions of the Cu-Se tetrahedra caused by strains along the principal crystallographic axes. Whilst all Cu-Se bonds are equivalent in both bond length and strength, these tetrahedra are arbitrarily rotated with respect to the principal axes. Thus, straining along these principal axes will cause inequivalent changes in the electronic structure at the VBM, leading to differences in the deformation potentials. For example, strain along the *a*-axis causes scissoring of pairs of Cu-Se bonds rather than significant changes in bond length, whereas strain along the *b*-axis distorts all four bonds.

The magnitude of the deformation potential is substantially reduced by structural relaxation in this flexible crystal structure, as outlined in Supplementary Note 7. We posit that quasi-2D structures with interlayer void space exhibits reduced deformation potentials compared to 3D structures, as strain-induced changes in bond length (due to the propagation of an acoustic wave) can be compensated by modulating the interlayer-spacing. To demonstrate this, we analyzed

the change in cation-anion bond lengths (after relaxation) as the $CuSbSe_2$ lattice was strained along the $c$-axis (i.e., perpendicular to $CuSbSe_2$ layers). The bonds considered are highlighted in Fig. 4a, and listed with the corresponding colors in Fig. 4b. When the strain reached ±5%, the changes in most bond lengths were below 1%, and the maximum change (Cu1-Se3 bond) was only around 2%. This is not explained by misalignment of the strain to bonding vectors, as we see differences of more than 4% for the same Cu1-Se3 bond in the unrelaxed case (i.e., for a uniform distribution of strain along the interatomic distances), and is also in contrast to the large change in interlayer distance of ±20% under ±5% $c$-axis strain. The fact that Cu-Se bonds exhibit more changes than Sb-Se bonds agrees with the COHP calculation results that indicate that the Cu-Se bonds are overall weaker due to the filled anti-bonding states in the VBM[118]. The strong relaxation in Cu-Se and Sb-Se bond lengths correlates well with the general reduction in acoustic deformation potentials presented in Supplementary Table 4, and suggests deformation potentials in Table 1, while low in absolute terms, are themselves an overestimation. This phenomenon should be considered when calculating deformation potentials in complex materials with similar structures to BiOI[39] and $CuSbSe_2$.

We also performed COHP calculations for the interlayer Sb-Se bonding interaction and derived an integrated crystal orbital Hamilton population (ICOHP) value as a measure of the covalent bonding strength (Fig. 4c). The much higher values of intralayer Sb-Se bonds (labeled in-layer in Fig. 4c) than interlayer Sb-Se bonds indicate that the interlayer covalent interaction is significantly weaker than the intralayer case, which is consistent with considering $CuSbSe_2$ as a layered material.

## High electronic dimensionality in CuSbSe$_2$

Electronic dimensionality also has an important effect on charge-carrier-phonon coupling, which can be described as a first approximation by the continuum model of Toyozawa, which considers both acoustic and optical phonon fields[102,119]. A 3D electronic structure can be advantageous by having an energy barrier against charge-carrier localization, but the energy barrier height should also be accounted for[28,120]. As for electronic 2D materials, the tendency to undergo strong coupling to acoustic phonons depends on the acoustic coupling factor $g_{ac}$. When $g_{ac} > 1$, barrierless charge-carrier localization is energetically favorable. On the contrary, for $g_{ac} < 1$, charge-carrier localization should not occur because the lattice energy increases as charge-carriers become more localized[120]. For 1D materials, this model predicts spontaneous localization in all cases. The electronic dimensionality of a semiconductor may be probed by analyzing the Fermi surfaces slightly above and below the CBM and VBM, which are shown for $CuSbSe_2$ in Fig. 4d. These surfaces are representative of the states occupied by free charge-carriers in the material as a result of thermal or optical excitation. Planar or columnar motifs are indicative of 1D and 2D structures respectively, indicating weak dispersion along the flat planar/axial direction(s). Meanwhile, ellipsoidal (closed-surface) motifs show dispersion in all directions and so are hallmarks of 3D electronic structures[73]. The VBM of $CuSbSe_2$ is unambiguously 2D in this transport regime due to the presence of a single columnar surface showing weak dispersion along the $c$-axis, while the CBM shows a number of ellipsoidal and closed rod-like structures, suggestive of an electronic structure that is 3D or close to 3D (Fig. 4d). The near-3D nature of the CBM is consistent with the lower CB being dominated by Sb-Se anti-bonding states, and there being weak interactions between the Sb and Se species across the interlayer gaps (Fig. 4c), which we refer to as quasi-bonding. By contrast, the 2D nature of the VBM is consistent with Cu-Se interactions, which dominate the upper VB, mostly occurring within each layer. The combination of the relatively high electronic dimensionality (especially in the CBM) and low $g_{ac}$ values overall are consistent with the band-like transport in $CuSbSe_2$.

This deviation in the electronic dimensionality from the structural dimensionality in $CuSbSe_2$ is consistent with what has been found in other pnictogen-based semiconductors. For example, although $Cs_2AgBiBr_6$ has a 3D crystal structure, its electronic dimensionality is significantly lower[121], which is one of the factors contributing to carrier localization in this material. As another example, although $Sb_2Se_3$ and $Sb_2S_3$ both have the same quasi-1D crystal structure, we see a 2D VBM in the former and a 3D VBM in the latter[73]. In the case of $CuSbSe_2$, although the electronic dimensionality of the VBM matches its quasi-2D structural dimensionality, the CBM has a 3D-like character. This emphasizes the importance of evaluating the electronic dimensionality and considering other important properties, such as the acoustic and Fröhlich coupling constants, in order to rationalize the nature of carrier–phonon interactions.

## Understanding weak Fröhlich coupling in CuSbSe$_2$

Another feature of $CuSbSe_2$ is its weak Fröhlich interaction, which primarily arises due to the small difference between the electronic and static dielectric constants, $\epsilon_\infty$ and $\epsilon_{stat}$ (refer to Eq. 1 and Eq. 3). This occurs when the ionic dielectric contribution is low relative to electronic contributions, and occurs due to both i) a high electronic dielectric contribution along all principal axes, and ii) a low ionic dielectric contribution, especially along the $a$- and $c$-axes (Table 1). The high electronic contribution is due to the small bandgap (since $\epsilon_\infty \propto E_g^{-0.5}$). In addition, the high density of states near the band edges will lead to a stronger interaction between electrons and light, giving a higher refractive index and higher $\epsilon_\infty$ (since $\epsilon_\infty \propto n^{0.5}$, where $n$ is the refractive index[122]). To understand the cause of the low ionic dielectric contribution, we calculated the Born effective charge (BEC) tensors for the different sublattices in $CuSbSe_2$ (Fig. 5). The Born effective charges ($Z_{\alpha,ij}^*$), also known as dynamical charges, describe the change in polarization in direction $i$ when the sublattice of atoms ($\alpha$) is displaced along direction j[123,124],

$$Z_{\alpha,ij}^* = \frac{\partial P_i}{\partial u_{\alpha,j}} \tag{4}$$

For materials with strong ionic-covalent bonding, the Born effective charges can be significantly larger than the formal oxidation states. The BECs for Cu1 are close to the oxidation state of the species, however, when considering the whole BEC tensor for Sb1 and Se1 atoms, we observe BEC values higher than the formal oxidation states with displacements along the $b$ direction, while net BECs (summing over columns) for displacements in the $a$ and $c$ directions are lower than the oxidation states. The low net dynamical charges of Sb1 and Se1 for $a$ and $c$ displacements are reflected by their close $\epsilon_\infty$ and $\epsilon_{stat}$ values, and contribute to the low Fröhlich coupling constants in these directions (Table 1). The anomalously large contributions of the Sb1 and Se1 atoms along the $b$-axis are of interest. These can be explained by either a change in the polarization of the Sb1 and Se1 atoms upon displacement, or a direct transfer of charge between the two species, however it is difficult to say for sure without further investigation[124]. The lone-pair on the Sb1 atom, and its origins in a symmetry-breaking interaction between hybridized Sb-$s$, $p$ orbitals, and Se $p$ orbitals[72] can explain this. Changes in the symmetry (e.g., via sublattice displacement) of the Sb-Se coordination sphere will change how the lone pair is expressed, leading to strong deviations in BEC. Nevertheless, despite the larger ionic contributions to the dielectric constant along the $b$-axis, Fröhlich coupling constants remain <2 (Table 1).

To put $CuSbSe_2$ in context, we compare the BEC values with those of other Sb- or Bi-based compounds, as well as $CH_3NH_3PbI_3$. The stable $6s^2$ lone pair of $Pb^{2+}$ results in anomalously higher BEC values than the formal valence of $Pb^{2+}$, hence leading to high ionic dielectric constants

$$Z^*_{\text{Cu1 (I)}} = \begin{pmatrix} 0.87 & 0 & -0.07 \\ 0 & 1.23 & 0 \\ 0.28 & 0 & 1.18 \end{pmatrix} \quad Z^*_{\text{Sb1 (III)}} = \begin{pmatrix} 2.53 & 0 & -0.62 \\ 0 & 5.65 & 0 \\ -1.50 & 0 & 2.06 \end{pmatrix}$$

$$\Sigma = \begin{bmatrix} 1.14 & 1.23 & 1.12 \end{bmatrix} \qquad\qquad \Sigma = \begin{bmatrix} 1.03 & 5.65 & 1.44 \end{bmatrix}$$

$$Z^*_{\text{Se1 (II)}} = \begin{pmatrix} -1.06 & 0 & -0.21 \\ 0 & -4.18 & 0 \\ -0.45 & 0 & -1.08 \end{pmatrix} \quad Z^*_{\text{Se2 (II)}} = \begin{pmatrix} -2.34 & 0 & 0.04 \\ 0 & -2.71 & 0 \\ 0.67 & 0 & -2.16 \end{pmatrix}$$

$$\Sigma = \begin{bmatrix} -1.51 & -4.18 & -1.29 \end{bmatrix} \qquad \Sigma = \begin{bmatrix} -1.67 & -2.71 & -2.12 \end{bmatrix}$$

displacement direction: $a$, $b$, $c$

**Fig. 5 | Calculated Born effective charge (BEC) of atoms in CuSbSe$_2$.** The BEC values represent the induced polarization when atoms are displaced, in this case, along the principal crystallographic axis directions. The net BEC values for displacements in the direction of each axis ($a$, $b$, or $c$) is shown below each tensor in square brackets.

($\epsilon_\infty = 6.1$; $\epsilon_{\text{stat}} = 25.7$[125]). These high dielectric constants lead to stronger Fröhlich interactions ($\alpha = 2$–$3$ for methylammonium lead iodide perovskites[24]) than CuSbSe$_2$, as discussed earlier. As Supplementary Table 5 shows, almost all pnictogen atoms in perovskite-inspired materials exhibit higher BEC values than their formal valences, despite the anisotropic values along certain directions. Compared to these compounds, the BEC values of Sb in CuSbSe$_2$ in the $a$ and $c$ directions are obviously lower than the formal oxidation state, resulting in the low ionic dielectric contribution, and, thus, weak Fröhlich interaction.

## Discussion

Based on our investigations, we propose that free volumes (e.g., interlayer gaps) in the structure can help minimize the effect of structural distortions on the bonding environment and lower the deformation potential. We proposed that this does not necessarily need to be in the form of a layered structure, but could also be achieved in motifs where there is a regular soft layer of species (e.g., molecular species) that do not contribute to orbitals at the band extrema. At the same time, quasi-bonding across these regular gaps between species contributing to the band-edge density of states is important for increasing the electronic dimensionality, which reduces the likelihood of self-trapping. This could be found more generally, for example, in materials that exhibit the stereochemical activity of the pnictogen cation (e.g., CuSbS$_2$[126], CuBiS$_2$[70], and CuBiSe$_2$[127]), resulting in a layered structure, with both the pnictogen and chalcogen placed about the interlayer gaps, allowing quasi-bonding to take place between them. Finally, materials with low ionic contributions to the dielectric constant are desired to minimize Fröhlich coupling, but this needs to be balanced with the effect on the capture cross-section of charged defects.

We believe that these insights, gained from investigating CuSbSe$_2$, are generalizable because the key structural and electronic features can be found in other materials, and the ways in which they affect carrier localization are rationalized based on the fundamentals of electron-phonon coupling theory (rather than bespoke theory specific to only CuSbSe$_2$). The important next step will be to test the wider applicability of these principles in broader sets of materials, particularly making use of the computational approaches we employed in this work. These efforts could ultimately lead to the development of simple descriptors for the high-throughput inverse design of pnictogen-based semiconductors with band-like transport.

On a more practical level specific to CuSbSe$_2$, whilst the charge-carrier mobilities ($1.01 \pm 0.01$ cm$^2 \cdot$V$^{-1} \cdot$s$^{-1}$ at room temperature) exceed those found for polycrystalline pnictogen-based semiconductors with self-trapping (Supplementary Table 1), they fall below the highest values achieved in single-crystal CuSbSe$_2$ (87 cm$^2 \cdot$V$^{-1} \cdot$s$^{-1}$)[44]. Given also that the intra-grain local mobility measured by OPTP exceeds the macroscopic mobility (as explained earlier), the charge-carrier mobilities of the CuSbSe$_2$ polycrystalline thin films prepared in this work are likely limited by grain boundary or structural defect scattering. Future efforts should, therefore, focus on reducing the density of these structural defects through processing or post-postprocessing strategies.

In conclusion, we have found CuSbSe$_2$ to be a heavy pnictogen-based chalcogenide that can avoid charge-carrier localization, which we determined through a combination of experiments and computations. A novel thiol-amine solution-processing method was employed to achieve phase-pure CuSbSe$_2$ thin films. OPTP measurements on CuSbSe$_2$ revealed a timescale of 6.7 ps to reach 50% photoconductivity decay, substantially slower than if carrier localization were present. Temperature-dependent Hall effect measurements confirmed the presence of large polarons based on the decrease in mobility with increases in temperature. Through DFT calculations, we found that both the acoustic and Fröhlich coupling constants are lower than those of many other heavy pnictogen-based materials, which supports the finding that CuSbSe$_2$ has weaker charge-carrier-phonon coupling. Whilst the effect of the deformation potential on the acoustic coupling strength, and relative size of the dielectric response factor on the strength of Fröhlich coupling are well established, it was not clear how these parameters could be tuned to achieve delocalized charge-carriers in heavy pnictogen-based semiconductors. In this work, we performed detailed computational investigations to reveal the factors involved, focusing on the bonding/anti-bonding nature of the crystal orbitals at the band extrema, and changes in bond lengths and interlayer spacing as a function of distortions, as well as the Born effective charges of ions. In particular, we show that deformation potentials can be minimized by having distortions to the unit cell due to the propagation of an acoustic wave relaxed through changes in geometry rather than bond length. This could be achieved through a layered structure, which provides sufficient degrees of freedom to allow bonds to mostly relax back to their equilibrium lengths following distortion. This could also be achieved by having groups of atoms contributing to the orbitals at band extrema (e.g., CuSe$_4$ tetrahedra) oriented at an angle to the principal axes, such that distortions are relaxed as changes in bond angles rather than bond length. Coupled with high electronic dimensionality (by having more than one species across interlayer gaps that

can form quasi-bonds), strong coupling to acoustic phonons is avoided. Meanwhile, the weak Fröhlich coupling is due to the high electronic contribution (mostly due to the small bandgap) and low ionic contribution to the dielectric constants. The latter arises from the Born effective charges of Sb, Cu, and Se not substantially deviating from their formal oxidation states (in contrast to lead-halide perovskites)[128]. This makes the important point that when it comes to materials design and the Born effective charge of species, there is a balance required between reducing Fröhlich coupling (lower BECs) and defect tolerance through dielectric screening (higher BECs). Overall, the insights made in this work are valuable for the future design of solar absorbers that have band-like transport.

## Methods

### CuSbSe$_2$ thin film deposition

About 0.1585 g Sb$_2$Se$_3$ (99.99% trace metals basis, Merck) and 0.0680 g Cu$_2$Se (99.5% metals basis, Alfa Aesar) were mixed in an empty vial, then 1 mL 1,2-ethylenediamine (for synthesis, Merck) and 0.1 mL ethane-1,2-dithiol (for synthesis, Merck) were added into the vial in an N$_2$-filled glovebox. Warning: the thiol-amine solvent system has to be processed in a glovebox because the thiol can react with humid air to produce H$_2$S, which is fatal if inhaled. The solution was firstly stirred at 70 °C for 10 min, then stirred at 30 °C overnight to fully dissolve precursors. Before spin coating, the solution was filtered with a 0.2-µm PTFE filter, and a 1.2 cm × 1.2 cm substrate was cleaned by sonication in acetone and isopropanol for 15 min, respectively. After the sonication cleaning, the substrate was blown dry with N$_2$, then UV-ozone treated for 20 min. Then 40 µL solution was spread onto the substrate, followed by spinning at 2000 rpm for 60 s. After spin coating, the sample was thermally treated on a hot plate at 100 °C for 10 min (ramp rate 30 °C min$^{-1}$). The sample, together with the hot plate, was then allowed to passively cool to room temperature. The cooling rate was estimated to be 5 °C min$^{-1}$. All of the above processes, except substrate cleaning, were performed in an N$_2$-filled glovebox, where the H$_2$O and O$_2$ levels were monitored and kept low (H$_2$O < 0.1 ppm; O$_2$ < 5 ppm). When the thermal treatment was completed, the sample was taken out of the glovebox and placed into a quartz tube for further heat treatment. The tube was firstly pumped to a pressure of ≈50 mTorr, then filled with Ar to reach a pressure of ≈1200 mTorr. Then the sample was heated to 400 °C (ramp rate 60 °C min$^{-1}$) and kept for 2 min, then cooled down naturally (estimated cooling rate: 10 °C min$^{-1}$) to obtain phase-pure CuSbSe$_2$ thin films (refer to Supplementary Fig. 1 for the X-ray diffraction patterns and phase-purity analysis).

### X-ray diffraction (XRD)

These measurements were performed in air at room temperature on a Bruker D8 Advance Eco instrument diffractometer. A copper K$_\alpha$ X-ray source ($\lambda(K_{\alpha 1})$ = 1.5406 Å; $\lambda(K_{\alpha 2})$ = 1.5444 Å) was utilized. Each measurement consisted of 4805 steps, with a dwell time of 0.35 s for each step. CuSbSe$_2$ film was deposited onto a 1.2 × 1.2 cm$^2$ single-crystal silicon substrate to minimize the background signal due to the substrate.

### Raman spectroscopy

Raman spectra were obtained in the air at room temperature with a Renishaw Raman system using a 532 nm wavelength continuous wave (cw) laser source. Before taking the measurements on CuSbSe$_2$, the equipment was calibrated by adjusting the characteristic Raman peak of the built-in silicon reference to 520 cm$^{-1}$. The final spectrum for each CuSbSe$_2$ sample was obtained by averaging 10 scans, where each scan took 5 s to collect. The CuSbSe$_2$ films were deposited onto 1.2 × 1.2 cm$^2$ glass substrates. The optical microscope built into the Raman spectrometer was used to focus the incident laser on the film surface before taking the Raman measurements.

### Fourier-transform infrared (FTIR) spectroscopy

FTIR spectra were obtained in dry N$_2$ at atmospheric pressure at room temperature with a Bruker Vertex 80 FTIR Spectrometer. The light source was a mid-infrared glowbar which is emissive from about 13000 to 40 cm$^{-1}$. The CuSbSe$_2$ films were deposited onto 7.5 × 2.5 cm$^2$ single-crystal silicon substrates. Before taking the measurements on CuSbSe$_2$, the wavenumber was calibrated by the mirror position, which was determined using the interference pattern of the HeNe laser, and the absolute reflectivity was calibrated using the blank 7.5 × 2.5 cm$^2$ single-crystal silicon substrate. The final spectrum for the CuSbSe$_2$ sample was obtained by averaging three scans, where each scan took 3176 s to collect.

### Absorption measurements

The absorption spectrum of CuSbSe$_2$ thin films on z-cut quartz substrates (Fig. 2a) was measured using a Fourier-Transform IR spectrometer (Vertex 80 v, Bruker) with a reflection-transmission accessory, a tungsten halogen lamp as light source, CaF$_2$ beamsplitter, and Si-detector. A silver mirror was used as a reflection reference, and the blank sample holder was used as a 100% transmission reference. The quartz substrate had a thickness of 2 mm and a diameter of 1.3 cm. The absorption coefficient α was calculated from Eq. 5:

$$\alpha = \frac{\ln\left(\frac{1-R}{T}\right)}{d} \tag{5}$$

where $R$ and $T$ are the reflectance and transmittance, respectively, of spin-coated CuSbSe$_2$ films, and $d$ is the film thickness. $R$ and $T$ were measured by UV-visible spectrophotometry within an integrating sphere, and $d$ was determined using a Dektak® stylus profilometer.

### Optical measurements

Long-time TA measurements were taken in air at room temperature. CuSbSe$_2$ films were deposited onto 1.2 × 1.2 cm$^2$ glass substrates. The third harmonic (355 nm) of an electronically controlled, Q-switched Nd:YVO$_4$ laser (Innolas Picolo 25) provided ~800 ps pump pulses. For short-time TA measurements, the fundamental Ti:Sapphire 800 nm wavelength laser provided ~150 fs pump pulses. Broadband near-IR probe pulses ranging from 800 to 980 nm were provided by a noncolinear optical parametric amplifier (NOPA) setup. Probe pulses were split into two beams by a beamsplitter. The other reference beam can then be used to calibrate shot-to-shot noise coming from the NOPA setup itself. This allows very weak signals to be measured. Both the probe and reference beams were detected by a Si dual-line array detector read out by a custom-built board from Stresing Entwicklungsbüro. The TA signals are expressed as $\frac{\Delta T}{T} = \frac{T_{\text{pump on}} - T_{\text{pump off}}}{T_{\text{pump off}}}$, where $T_{\text{pump on}}$ and $T_{\text{pump off}}$ represent the transmission with and without the pumping, respectively.

OPTP measurements were conducted at room temperature using a setup described in detail elsewhere[28]. Briefly, an amplified Ti-Saph laser system (Spectra-Physics, Spitfire) provides 800 nm wavelength pulses of 35 fs pulse duration and 5 kHz repetition rate. Single-cycle THz radiation pulses were generated via the inverse spin Hall effect upon photoexcitation of a spintronic emitter with the fundamental laser output[62]. THz detection was achieved by using a fraction of the fundamental laser output to gate the THz signal by free-space electro-optic (EO) sampling with a 1-mm-thick ZnTe (110) crystal. Here, a Wollaston prism was used to separate different circularly polarized components of the gate, which were then measured by a pair of balanced photodiodes. Samples were excited by frequency-doubled 400 nm pulses, obtained by second-harmonic generation in beta-barium-borate (BBO) crystal. During the OPTP measurements, the THz emitter, EO crystal, and samples are kept under vacuum at pressures below 10$^{-1}$ mbar. For OPTP measurements, samples were spin-coated CuSbSe$_2$ thin films on 2 mm thick circular z-cut quartz substrates with 1.3 cm diameter.

## Hall effect measurement

Samples for Hall effect measurements were prepared according to the van der Pauw method. A 100-nm thick gold was evaporated onto four corners of each $CuSbSe_2$ film sample as metal contacts ($0.2\,cm \times 0.2$ cm size). The substrate was a $1.2 \times 1.2\,cm^2$ glass substrate. Then the gold contacts were wired to the system for measurements. Hall effect measurement at room temperature (300 K) was performed in air with the Lake Shore 8400 Series under a 1 T magnetic field. The ohmic check was run before the Hall effect measurements to make sure the quality of metal contacts and electric connections was good.

For Hall effect measurements at lower temperatures, the same sample geometry was used. The measurements were carried out in a 16 T superconducting magnet with temperatures ranging from 1.3 to 300 K in a helium gas environment. The samples were mounted on the probe in the magnetic fields perpendicular to the ab-plane and glued with a GE-varnish. A combination of silver pastes and silver wires was used to make the electrical connection. After drying the pastes, it was confirmed that the contact resistances were acceptable within the order of a few ohms. The longitudinal ($\rho_{xx}$) and transverse resistivity ($\rho_{xy}$) were obtained using the van der Pauw technique with a current amplitude of 0.1–1 mA, and an alternating frequency of 3–17 Hz with the help of SR830 digital lock-in amplifier. The perpendicular magnetic fields were swept at a rate of 0.5 T $min^{-1}$ at a given temperature. All data were taken over a full range from −16 to 16 T, averaged in positive and negative fields to remove a small longitudinal resistance contribution to the measured voltage, which may arise from the van der Pauw geometry, and retain only the antisymmetric voltage component due to the Hall effect.

## First-principles calculations

For the computations made in this work, we carefully selected the functional to use that struck a balance between accuracy and computational cost. For example, we found that the regularized and restored SCAN ($r^2$SCAN) meta-GGA functional provided a much more accurate description of $\Gamma$-point modes, and was, therefore, more suitable for computing force constants and the phonon dispersion curve. PBE is also well known to poorly describe the electronic structure, and we, therefore, used hybrid functionals (details below) for computing properties requiring an accurate prediction of the bandgap, such as the dielectric constant. PBE was, however, suitable for DFPT calculations of ionic dielectric properties. It was not feasible to run all calculations using hybrid functionals due to computational cost and incompatibility of DFPT with hybrid/mGGA functionals.

Calculations of the crystal, electronic, phonon structure, and bulk-polaron partial charge density functions were carried out in the Kohn Sham density functional theory (KS-DFT) framework[129] using with the projector augmented wave (PAW) method[130] as implemented using the Vienna ab initio software package (VASP)[131]. The PBE.54 PAW potential set was used throughout (Cu 22Jun2005, Sb 06Sep2000, and Se 06Sep2000). Electronic structure calculations (Figs. 2b, 4c), including structural relaxation and deformation potentials, were carried out using the hybrid functional of Heyd, Scuseria, and Ernzerhof (HSE06)[95] hybrid functional using a $4 \times 6 \times 2$ $\Gamma$-centered grid and a plane wave cut-off of 300 eV. Band structure between high symmetry points was interpolated from a densely sampled uniform band structure calculation generated using the zero-weighted k-point method atop a weighted $4 \times 6 \times 2$ grid. The dielectric function was calculated in the single particle approximation using the linear optics routine of Gajdos et al. implemented in VASP[130–134] (LOPTICS = .TRUE.). The dielectric function was found to converge to one decimal place when the number of bands (NBANDS) was increased to 204. Projection from a plane wave to orbital basis and subsequent COHP[135] analysis was achieved using the postprocessing, analysis, and plotting tools Lobster[136–138] and LobsterPy[139]. Sumo[140] was used to plot the DOS. Gaussian broadening of 0.12 eV was applied to both COHPs and DOS. Deformation potential calculations were calculated using the method of Wei and Zunger[141–143]

with deformed structures generated and analyzed via the ab initio scattering and transport (AMSET) package[99]. The phonon band structure was calculated using the finite displacement method[144] within the harmonic approximation as implemented in Phonopy using a displacement of 0.15 Å[145,146], using the $r^2$SCAN meta-GGA functional[147]. It should be noted that within the harmonic approximation, finite temperature effects are neglected. Prior to supercell calculations, the $r^2$SCAN functional was used to perform a tight structural optimization over a $6 \times 8 \times 2$ $\Gamma$-centered grid and a plane wave cut-off of 500 eV. $3 \times 4 \times 1$ supercells were used to calculate force constants over a commensurate $2 \times 2 \times 1$ k-point grid. The plotting utility ThermoParser[148] was used to generate phonon band structure and DOS plots. The static dielectric and BECs were calculated via DFPT (IBRION = 8, LEPSILON = .TRUE.) using the functional of Perdew–Burke, and Ernzerhof (PBE)[149], using a plane wave cut-off of 300 eV and an $8 \times 14 \times 3$ k-point grid at a single q-point ($\Gamma$).

The search for low-energy bulk polarons was carried out using the ShakeNBreak method and package[104,150]. A 64-atom supercell was used to perform a spin-polarized calculation (ISPIN = 2) containing an unpaired electron or hole, which was enforced using NELECT and NUPDOWN = 1 INCAR tags at HSE06 level. Local distortions around atomic sites of $\pm 30\%$ and 0% were applied, followed by a stochastic rattle of all atoms in the cell with a standard deviation of 0.25 Å. An unperturbed supercell was also run. For holes, distortions centered on Cu, Se1, and Se2 were trialed, while for electrons, distortions centered on Sb were trialed. Distorted supercells were relaxed within the $\Gamma$-point approximation using the $\Gamma$ only version of VASP. Energy data was plotted using the tools within ShakeNBreak, while partial charge densities of bands containing unpaired hole and electron were generated using pymatgen[151] and plotted using VESTA.

## Data availability

The raw data (both experimental and computational) generated in this paper and the Supplementary Information can be found from the Oxford University Research Archive (ORA) Data Repository, with the link https://doi.org/10.5287/ora-dqngo81dy. The atomic coordinates of the three optimized structures used in this work are provided as Supplementary Data. DFPT and elastic constant calculations were carried out using the PBE functional and structure (Supplementary Data 1). Phonon dispersion calculations were carried out using the $r^2$SCAN functional and tightly optimized structure (Supplementary Data 2). Electronic structure calculations (including band structure, DOS, COHP, and deformation potentials) were carried out using the HSE06 functional and structure (Supplementary Data 3). In addition to providing these structures as supplementary data in CIF format, we have also included them in our ORA repository file in VASP POSCAR/OUTCAR format, as documented in the readme.txt.

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

## Acknowledgements

The authors thank Xinwei Wang and Alex Ganose for valuable discussions, Robert Jacobs for collecting the FTIR data, as well as Mark Isaacs for collecting the XPS data at the EPSRC National Facility for XPS (HarwellXPS), which is operated by Cardiff University and UCL under contract no. PR16195. Y.F., H.L., and R.L.Z.H. thank the UK Research and Innovation for funding through a Frontier Grant (no. EP/X022900/1), awarded via the European Research Council 2021 Starting Grant scheme. Y.-T.H. and R.L.Z.H. thank the Engineering and Physical Sciences Research Council (EPSRC) for funding (no. EP/V014498/2). Y.F., H.L., Y.-T.H., and R.L.Z.H. thank the Henry Royce Institute for support through the Industrial Collaboration Program, funded through EPSRC (no. EP/X527257/1). L.M.H. acknowledges support through a Hans Fischer Senior Fellowship from the Technical University of Munich's Institute for Advanced Study, funded by the German Excellence Strategy. L.M.H. and M.R. thank EPSRC for funding. S.R.K. acknowledges the Harvard University Center for the Environment (HUCE) for funding a fellowship. R.L.Z.H. also thanks the Royal Academy of Engineering for funding via the Research Fellowships scheme (no. RF\201718\17101).

## Author contributions

R.L.Z.H. conceived of and supervised this project as a whole. Y.F. developed the method for synthesizing $CuSbSe_2$ by solution processing and performed X-ray diffraction, Raman spectroscopy, Fourier-transform infrared spectroscopy, UV-visible spectroscopy, and room temperature Hall effect measurements, supervised by R.L.Z.H. H.L. performed the computations, with support from S.R.K., Y.W.W., and D.O.S., and supervised by A.W. M.R. performed the optical pump terahertz probe spectroscopy measurements and Elliot model fit, supervised by L.M.H. Y.-T.H. performed transient absorption spectroscopy, supervised by R.L.Z.H. and A.R. C.-W.C. performed low-temperature Hall effect measurements, supervised by B.A.P. S.J.Z. performed photothermal deflection spectroscopy measurements. H.D. helped optimize the heat treatment of $CuSbSe_2$ thin films, supervised by S.H. M.A.M. provided support on thin film deposition equipment. All authors contributed to writing the manuscript.

## Competing interests

The authors declare no competing interests.

## Additional information

[1]Inorganic Chemistry Laboratory, University of Oxford, South Parks Road, Oxford OX1 3QR, United Kingdom. [2]Department of Materials and Centre for Processable Electronics, Imperial College London, Exhibition Road, London SW7 2AZ, United Kingdom. [3]Department of Physics, University of Oxford, Clarendon Laboratory, Parks Road, Oxford OX1 3PU, United Kingdom. [4]Cavendish Laboratory, University of Cambridge, JJ Thomson Ave, Cambridge CB3 0HE, United Kingdom. [5]Harvard University Center for the Environment, Cambridge, Massachusetts 02138, USA. [6]Laboratoire National des Champs Magnétiques Intenses, CNRS, LNCMI, Université Grenoble Alpes, Université Toulouse 3, INSA Toulouse, EMFL, F-38042 Grenoble, France. [7]Department of Physics, Chungnam National University, Daejeon 34134, Republic of Korea. [8]Department of Experimental Physics, Faculty of Fundamental Problems of Technology, Wrocław University of Science and Technology, Wybrzeże Wyspiańskiego 27, 50-370, Wrocław, Poland. [9]Department of Materials Science and Engineering, Yonsei University, Seoul 03722, Republic of Korea. [10]London Centre for Nanotechnology, Imperial College London, Prince Consort Road, London SW7 2AZ, United Kingdom. [11]Department of Materials, Imperial College London, Molecular Sciences Research Hub, Wood Lane, W12 0BZ London, United Kingdom. [12]School of Chemistry, University of Birmingham, Birmingham B15 2TT, United Kingdom. [13]Institute for Advanced Study, Technical University of Munich, Lichtenbergstrasse 2a, D-85748 Garching, Germany. [14]These authors contributed equally: Yuchen Fu, Hugh Lohan. ✉e-mail: robert.hoye@chem.ox.ac.uk

