## [Transparent Peer Review file · Nature Communications]

Structural and Electronic Features Enabling Delocalized Charge-Carriers in CuSbSe₂

Corresponding Author: Professor Robert Hoyer

Version 0:

Reviewer comments:

Reviewer #1

(Remarks to the Author)

This manuscript sheds light on the reasons for carrier localization in certain inorganic semiconductors based on Sb³⁺ and Bi³⁺ cations. The authors have attempted to explain that a weak coupling between acoustic and optical phonons with the charge carriers is a deciding factor for the charge carrier delocalization in CuSbSe₂ material. The slower decay observed in the OPTPS measurements indicates the formation of delocalized free charge carriers as compared to the very fast decay in Bi-based materials. The inference obtained from the theoretical studies also supports the results of the different characterizations. Manuscript needs clarity and some aspects need better clarifications and details. Following are a few comments:

1. Authors say that "In this work, we developed a solution processing route to achieve phase-pure CuSbSe₂ thin films.", I don't understand this, is this altogether a new process or recipe or is it derived from the literature? Clarity is needed on what is the new part. Full details of the complete processing are required.
2. Characterization details are missing such as type of instrument used, conditions and parameters used need to be mentioned clearly.
3. Why does XRD data not appear so clean, why is there so much of background, there seem to be many other peaks! The data is not really a reflection of a high quality sample. Have you identified all the peaks, Please provide complete information.
4. What were the ramp rate and cooling rate before and after annealing? Also, annealing is a treatment which is often used for metals (for relieving stress and changing grain size), not for thin films, It should be labelled as heat treatment because it does lead to change in phase and crystallization. This is a wrong terminology. Even if whole world uses it, it is fundamentally incorrect.
5. I doubt if you can use Raman and FTIR for phase analysis in a complete sense. This is rather incorrect and inappropriate way to use these techniques. Raman is not exactly a finger printing technique unlike XRD, its spectra depends on a variety of factors. This is a very basic of materials characterization.
6. I don't think there is enough evidence to back this statement, the sample quality isn't as great as is being claimed: "Combining these XRD, Raman and FTIR measurements, we can conclude that the spin-coated CuSbSe₂ thin films are phase-pure after annealing at 400 °C for 2 min." The XRD plot itself is not an excellent one.
7. Why is the absorbance range so narrow? What was the measurement range?
8. On page 26, the authors mentioned "We posit that quasi-2D structures with interlayer void space exhibits reduced deformation potentials compared to 3D structures, as strain-induced changes in bond length can be compensated by modulating the interlayer-spacing." I would like the authors to elaborate on the origin of strain in CuSbSe₂ material and how the authors have controlled the degree of strain produced along the principal crystallographic axes? There is not so straightforward correlation as being claimed. You also need to provide better justifications.

9. For calculating the acoustic deformation potential, a more detailed explanation is needed. Current details are not sufficient.

10. A comprehensive explanation of the charge carrier mobility variation based on the type of polarons produced in the CuSbSe₂ system is needed and will be critical in understanding the role of polarons in dictating the charge carrier dynamics in these systems. Current details are very sketchy.

11. It is strange that error bars seem to be missing all over, they need to be provided.

Discussion is very hand-waiving without much convincing evidence. Overall, I think manuscript requires substantial improvement.

Reviewer #2

(Remarks to the Author)

This article effectively demonstrates that CuSbSe₂, a heavy pnictogen-based chalcogenide, can prevent charge carrier localization at the greatest extent, leading to a significant superiority over other pnictogen-based materials. The authors utilize both experimental methods and computational simulations to provide insightful observations. The thiol-amine solution processing method employed for obtaining phase-pure CuSbSe₂ thin films demonstrates practical feasibility. The photoconductivity decay timescale and temperature-dependent Hall effect measurements confirm the presence of large polarons in CuSbSe₂. The paper employs density functional theory (DFT) calculations to delve into the acoustic and Fröhlich coupling constants, revealing them to be lower compared to many other heavy pnictogen-based materials. The low deformation potentials and relatively high electronic dimensionality contribute to weak coupling to acoustic phonons, and the weak Fröhlich coupling arises from the small bandgap and low ionic contribution to dielectric constants.

This research holds significance for the design and optimization of solar absorbers. The unique characteristics of CuSbSe₂, particularly its ability to resist charge carrier localization, position it as a potential solar absorption material. This contributes to the advancement of more efficient photovoltaic materials. I recommend this work be published after the following concerns are addressed:

1. How does the preparation method of CuSbSe₂ affect its performance? Was the composition of precursor optimized specifically to gain the pure-phase compound? Can this preparation method be extended to similar materials?
2. How is the stability of CuSbSe₂? Have the authors done relevant investigations? As far as I am concerned, the valence states of cations demonstrate an unstable state of this compound.
3. What was the substrate material for depositing CuSbSe₂? Was it conductive or non-conductive? Have the authors investigated any applications of CuSbSe₂ on photovoltaic or photo-electrochemistry?

Reviewer #3

(Remarks to the Author)

Y. Fu et al performed an interesting joint experimental and computational study of carrier localization in CuSbSe₂. Unlike other Sb- and Bi-based inorganic semiconductors, they have demonstrated that CuSbSe₂ exhibits delocalized free carriers through mobility measurements as a function of temperature and optical pump terahertz probe spectroscopy; those results together with the solution processing method are the most important contributions.

The experimental results are supported by calculations of the mobility, acoustic deformation potential, and Fröhlich coupling constant. However, the level of theory used to evaluate mobilities and electron-phonon coupling strengths is not at the level of the state-of-the-art. For example, the calculation of the deformation potential relies on the rigid ion approximation whose validity in several systems is questionable. Similarly, the calculations for temperature-dependent mobilities in the supplemental material noticeably overestimate the experimental values, an aspect that lacks clarification in the manuscript. Additionally, it seems the authors might have overlooked important first-principles studies in calculating electron-phonon coupling, mobilities, and polarons in other compounds. The association between mobilities and large polarons is less convincing due to a lack of clear evidence for polaron formation in these systems.

The most appealing results of this manuscript are indeed the measurements (Raman spectra, X-ray diffraction, and mobilities) and the solution processing method which all seem to be of high quality. However, the manuscript does not meet Nature Communications' criteria as the authors' claims on the existence of large polarons, the theoretical approach, and the novelty in advancing a particular field are not convincing.

Some other comments, without a specific order, include:

1. It is not surprising that acoustic phonons lead to weak electron-phonon coupling effects which is the case for halide perovskites.
2. Could the authors provide more clarity regarding their confidence in assigning B_{2u} and B_{3u} modes in the FTIR measurement? Was this determination based on energy matching? It would enhance the understanding if the authors could

specify their level of confidence, perhaps using terms like "tentatively" to convey the certainty of the assignment.

3. Supl. Fig.3: I think combining figures (a) and (b) can facilitate direct comparison.

4. The figure descriptions in the main text are out of sequence. For example, the reference to Fig. 2b appears after the discussion of Fig. 3.

5. It appears that there might be confusion on the part of the authors regarding the definition of polarons. At variance with phonons (which represent vibrations around equilibrium positions), a polaron causes lattice distortion. However, polaron effects are not captured in their theoretical calculations. What is calculated here is the phonon-limited mobility due to LO phonons and not the polaron-limited mobility. The interpretation that the decrease in mobility implies the presence of large polarons is not necessarily correct.

6. AMSET is not defined either in the main manuscript or the supplemental material.

7. The authors could comment on why the calculated mobility is overestimated with respect to the experiment.

8. Surprisingly the band gap of the material is not reported.

9. How the average in Table 1 is taken? Some values do not represent the average, e.g. the binding energy.

10. I am uncertain about the authors' decision to relax the structure after applying strain. Their objective is to compute the acoustic deformation potential, and its derivatives are typically taken with respect to the structure at equilibrium.

11. The method for calculating the prediction of the Acoustic Deformation Potential (ADP) and Polar Optical Phonon (POP) contributions in Supplemental Fig. 5 is unclear.

12. It would be nice for the authors to comment that the temperature dependence of the effective masses and phonon frequencies (anharmonicity, renormalization of the phonon self-energy) are not taken into account. In general, it will be beneficial for the manuscript to state all the approximations involved in computing mobilities, deformation potentials, and Fröhlich coupling.

13. A discussion of how the mobility, effective masses, band gaps, dielectric constants, and effective charges are compared to other Sb- and Bi-based compounds could be beneficial to strengthen the points of the manuscript.

14. The use of three different levels of approximations (i) meta-GGA functional for force constants, (ii) hybrid functionals for electronic structure calculations, and (iii) PBE for DFPT is quite unusual and not justified. For example, why not use PBE for all calculations? Is there a particular reason?

15. Given the extensive discussion on mechanisms for weak charge-carrier-phonon coupling, there is also some uncertainty about whether a manuscript in the style of a "Communications" article is appropriate.

Version 1:

Reviewer comments:

Reviewer #2

(Remarks to the Author)

The response from the authors well addresses concerns and questions raised by me.

Reviewer #3

(Remarks to the Author)

I would like to thank the authors for their great effort in addressing both my comments and those of other reviewers. The manuscript's clarity has significantly improved, particularly regarding the computational part of the work. I agree with the removal of mobility calculations which indeed can potentially divert attention from the manuscript's key messages. Now the manuscript lacks any substantial quantitative comparison between calculations and experiments. Nonetheless, qualitatively, computational analysis remains valuable in interpreting the data.

The only comment made by the authors in the rebuttal letter and is not reflected in the manuscript is the following:

"In regard to other methods of calculating electron-phonon coupling, we agree that this is an exciting and rapidly growing area of research, but would stress that due to the sheer number of methods available (e.g., ab-initio MD, Quantum Monte-Carlo, the Special Displacement Method, the ab-initio theory of polarons of Sio et al.), an in-depth discussion of recent developments is outside the scope of this work."

I believe it's worth acknowledging the above methods in the main manuscript as promising avenues for future research. Using these methods could potentially lead to more accurate calculations of polarons and electron-phonon coupling, thereby facilitating a better quantitative comparison with experimental results. Including these approaches in the manuscript as future directions may enhance its scope and contribute to advancing the field.

After this comment is addressed, I believe the manuscript is suitable for publication in Nature Communications.

Reviewer #4

(Remarks to the Author)

In this revised manuscript, the authors explore factors that can delocalize carriers in pnictogen-based solar absorbers, using CuSbSe₂ as a case study. They propose that carrier delocalization in CuSbSe₂ is influenced by (a) weak coupling to acoustic phonons due to a low deformation potential, and (b) weak coupling to optical phonons due to a relatively low ionic contribution to the dielectric constant compared to the electronic contribution. While the paper is well-written and is praiseworthy for the synthesis advancement, I believe it does not provide insights or results significant enough to warrant publication in Nature Communications.

The assertion that reducing carrier coupling with phonons—whether acoustic or optical—is beneficial for delocalization is a known principle that researchers with a solid understanding of chemistry are likely familiar with. The insights provided, such as the relationship between low deformation potential and reduced carrier localization for reducing coupling with acoustic modes, or the impact of electronic versus ionic contributions to the dielectric constant in determining the coupling with optical modes, are generally well-understood principles in the field. In this context, the connection between layered structures and low deformation potential is also a predictable outcome, as is the expectation that high bandgap materials will have a lower electronic contribution to the dielectric constant.

Nevertheless, the manuscript presents interesting points regarding the nature of bonding/antibonding interactions near the valence band maximum (VBM) and conduction band minimum (CBM) and their impact on electronic dimensionality and deformation potential. Similarly, the discussion involving the effect of Born effective charges is also interesting. However, these insights are of limited practical use without known band structure data of a system. The authors do not provide sufficient guidance on predicting these properties from atomic composition or structure alone, which is a critical limitation in materials selection.

While the authors have addressed previous technical comments adequately, the manuscript does not convincingly advance the field of pnictogen-based solar absorbers. Therefore, I cannot recommend it for publication in its current form.

Nonetheless, the discussions are valuable and may be suitable for a different publication venue. Furthermore, the authors can investigate these concerns to improve the discussion part in the manuscript.

1. The authors are encouraged to give temperature dependent resistivity plot for more clarity of the carrier transport.
2. The authors should properly investigate exact temperature dependence of the mobility data and then discuss about the transport mechanism.
3. Can the authors compare the magnetoresistance of the materials having localized carriers with those having delocalized carriers? Magnetoresistance of small polaron and large polaron should be different.
4. The stability of CuSbSe₂, especially under operational conditions relevant to solar cells, is not thoroughly explored. Given the focus on earth-abundant and environmentally friendly materials, understanding long-term stability in diverse environmental conditions is essential.
5. While the work identifies mechanisms to avoid carrier localization in CuSbSe₂, it still acknowledges the prevalence of carrier localization in many pnictogen-based semiconductors. This limitation suggests that achieving truly delocalized charge-carriers remains a significant challenge, possibly limiting the broad applicability of the findings to other materials within this class. Furthermore, the unique structural features of CuSbSe₂, such as the stereochemical activity of the lone pair on Sb³⁺, might not be present in other materials, limiting the generalizability of the results.
6. Although the localization effect is minimal in CuSbSe₂, but the macroscopic mobility is still much lower for example compared to single crystalline MAPbI₃. For application in solar cells, this is a crucial aspect that needs to be addressed for CuSbSe₂.

Version 2:

Reviewer comments:

Reviewer #4

(Remarks to the Author)

Author satisfactorily answered all the comments now. The paper can be accepted as is.

The comments from the reviewers are in black, and our responses are in blue. All text quoted from the main text or Supplementary Information are in Times New Roman and italicized, with all changes highlighted in yellow.

Reviewer 1:

This manuscript sheds light on the reasons for carrier localization in certain inorganic semiconductors based on Sb^{3+} and Bi^{3+} cations. The authors have attempted to explain that a weak coupling between acoustic and optical phonons with the charge carriers is a deciding factor for the charge carrier delocalization in CuSbSe_2 material. The slower decay observed in the OPTPS measurements indicates the formation of delocalized free charge carriers as compared to the very fast decay in Bi-based materials. The inference obtained from the theoretical studies also supports the results of the different characterizations. Manuscript needs clarity and some aspects need better clarifications and details. Following are a few comments:

We thank the reviewer for their time in evaluating our paper, and for their detailed and constructive comments. We have now gone through to improve the clarity of our message, and provided further experiments to strengthen our conclusions, as detailed below.

1. Authors say that "In this work, we developed a solution processing route to achieve phase-pure CuSbSe_2 thin films.", I don't understand this, is this altogether a new process or recipe or is it derived from the literature? Clarity is needed on what is the new part. Full details of the complete processing are required.

Response 1.1: Indeed, the thiol-amine solution processing route for synthesising CuSbSe_2 thin films is a new processing route for this material itself, which we have developed as part of this work. We elaborated on this point on pages 11 and 12 of the results section:

*"Previous efforts at growing CuSbSe_2 focussed on vacuum-based approaches (e.g., sputter deposition⁷⁵, close-space sublimation⁷⁶), methods that have long reaction times (e.g., fusion method⁷⁸ or selenization of metal precursors⁷⁸), or processes involving the use of toxic precursors (e.g., hydrazine solvent^{41,79}). Solution-processing is advantageous in requiring less capital-intensive equipment than vacuum-based processing⁸⁰⁻⁸², but at the same time, it is critical to avoid the use of toxic solvents⁸³. More recently, a more benign solvent system than hydrazine, comprised of a thiol-amine mixture, has been found to be effective in dissolving chalcogenide precursors and successfully used to deposit absorber layers of photovoltaic devices, such as $\text{Cu}_2\text{ZnSn}(\text{S},\text{Se})_4$ ^{84,85}, $\text{Cu}(\text{In}, \text{Ga})\text{Se}_2$ ^{86,87} and $\text{CuIn}(\text{S}, \text{Se})_2$ ⁸⁸. **In this work, we develop a thiol-amine route to synthesize CuSbSe_2 thin films for the first time**, as detailed in Methods."*

In more detail: this thiol-amine route was firstly reported in 2013 and used to dissolve nine binary chalcogenide materials¹. Since then, the thiol-amine solvent system has shown the ability to dissolve a wide range of elements, chalcogenides and many other salts, including the two precursors used in this work, Cu_2Se ^{2,3} and Sb_2Se_3 ^{1,4}. Moreover, by mixing different precursor solutions together, followed by coating and heat treatment, a series of chalcogenide compounds, such as $\text{CuIn}(\text{S},\text{Se})_2$ ⁵ and $\text{Cu}_2\text{ZnSnS}_4$ ⁶, have been deposited as thin films. However, this thiol-amine route has never been used to deposit CuSbSe_2 . Inspired by these previous works and the present gap in the literature, we developed this novel thiol-amine route to deposit CuSbSe_2 thin films.

Compared to other reported methods for synthesizing CuSbSe_2 , our method is more cost-effective and faster. That is, unlike sputter deposition, spin coating does not require expensive equipment. As another example, selenizing metal precursors requires long deposition times >30 min⁷, whereas our thiol-amine route deposits the films within 1 min, and requires only a 2 min heat treatment afterwards. The thiol-amine route is also advantageous over the previously-reported hydrazine route for preparing CuSbSe_2 thin films because we can prepare our solutions with overnight stirring, whereas hydrazine processing required several days of continuous stirring to fully dissolve the precursors⁸. Moreover, ethane-1,2-dithiol and ethylenediamine are safer than hydrazine, which is highly toxic and causes severe harm to the skin, is highly corrosive, emits toxic fumes if heated to decomposition, and can self-ignite at low temperatures⁹. We do mention in the Methods section that the thiol-amine process needs to be completed in a glovebox because thiols will evolve H_2S when in contact with moist air, but this is easier to manage than the severe hazards associated with hydrazine.

We have provided the requisite details of the optimized process for preparing CuSbSe_2 thin films in the Methods section, as detailed in the sub-section labelled “*CuSbSe₂ thin film deposition*”. This includes the precursors we used, their purities and suppliers, as well as the masses and volumes of these precursors that we used to prepare the precursor solutions. We also specified the environment used to prepare the precursor solutions and deposit the films, as well as our optimised heat treatment protocol. All of these details are provided on pages 39 and 40 of the main text. If there are any other details that the reviewer would like us to provide, we would be glad to oblige.

Going back to the specific quote the reviewer raised, to clarify that the thiol-amine route to preparing CuSbSe_2 is new, we have modified the sentence on page 9 at the end of the introduction as follows:

“*In this work, we developed a novel thiol-amine-based solution processing route to achieve phase-pure CuSbSe_2 thin films.*”

In the results section on page 12, we have modified our explanation of the synthesis route as follows:

“*In this work, we develop the synthesis of phase-pure CuSbSe_2 thin films by this novel thiol-amine-based solution processing route for the first time, as detailed in Methods.*”

At the same time, we would like to emphasise that although this is one of the novelties of this work, it is not the most important conceptual advance. The core aim of this work was to understand the chemical and physical factors that enable pnictogen-based perovskite-inspired materials to avoid carrier localisation. We reveal CuSbSe_2 to exhibit band-like transport, in contrast to many previously-investigated pnictogen-halide compounds, making it an important case-study. We investigate this material in detail through both experiment and computations, providing important new insights into how the structure and combination of chemical species lead to low deformation potentials and low Fröhlich coupling constants (see Conclusions for full details). We therefore did not wish to detract too much from this with the discussion on the thiol-amine processing route, and hope that the clarification we have made in the quoted passages above serves to provide sufficient clarity on novelty without distracting from the main focus and main conceptual advance of this paper.

2. Characterization details are missing such as type of instrument used, conditions and parameters used need to be mentioned clearly.

Response 1.2: More characterization details have been added, as highlighted in yellow below:

“X-ray diffraction (XRD). These measurements were performed in air at room temperature on a Bruker D8 Advance Eco instrument diffractometer. A copper K_{α} X-ray source ($\lambda(K_{\alpha 1})=1.5406\text{\AA}$; $\lambda(K_{\alpha 2})=1.5444\text{\AA}$) was utilized. Each measurement consisted of 4805 steps, with a dwell time of 0.35 s for each step. CuSbSe_2 film was deposited onto a $1.2 \times 1.2 \text{ cm}^2$ single crystal silicon substrate to minimize the background signal due to the substrate.

Raman spectroscopy. Raman spectra were obtained in air at room temperature with a Renishaw Raman system using a 532 nm wavelength continuous wave (cw) laser source. Before taking the measurements on CuSbSe_2 , the equipment was calibrated by adjusting the characteristic Raman peak of the built-in silicon reference to 520 cm^{-1} . The final spectrum for each CuSbSe_2 sample was obtained by averaging 10 scans, where each scan took 5 s to collect. The CuSbSe_2 films were deposited onto $1.2 \times 1.2 \text{ cm}^2$ glass substrates. The optical microscope built in to the Raman spectrometer was used to focus the incident laser on the film surface before taking the Raman measurements.

Fourier transform infrared (FTIR) spectroscopy. FTIR spectra were obtained in dry N_2 at atmospheric pressure at room temperature with a Bruker Vertex 80 FTIR Spectrometer. The light source was a mid-infrared glowbar which is emissive from about 13000 to 40 cm^{-1} . The CuSbSe_2 films were deposited onto $7.5 \times 2.5 \text{ cm}^2$ single crystal silicon substrates. Before taking the measurements on CuSbSe_2 , the wavenumber was calibrated by the mirror position which was determined using the interference pattern of the HeNe laser, and the absolute reflectivity was calibrated using the blank $7.5 \times 2.5 \text{ cm}^2$ single crystal silicon substrate. The final spectrum for the CuSbSe_2 sample was obtained by averaging 3 scans, where each scan took 3176 s to collect.

Absorption measurements. The absorption spectrum of CuSbSe_2 thin films on z-cut quartz substrates (Figure 2a) was measured using a Fourier-Transform IR spectrometer (Vertex 80v, Bruker) with a reflection-transmission accessory, a tungsten halogen lamp as light source, CaF_2 beam splitter, and Si-detector. A silver mirror was used as a reflection reference, and the blank sample holder was used as a 100% transmission reference. The quartz substrate had a thickness of 2 mm and diameter of 1.3 cm...

Optical measurements. Long-time TA measurements were taken in air at room temperature. CuSbSe_2 films were deposited onto $1.2 \times 1.2 \text{ cm}^2$ glass substrates. The third harmonic (355 nm) of an electronically controlled, Q-switched Nd:YVO₄ laser (Innolas Picolo 25) provided ~800 ps pump pulses. For short-time TA measurements, the fundamental Ti:Sapphire 800 nm wavelength laser provided ~150 fs pump pulses...

OPTP measurements were conducted at room temperature using a setup described in detail elsewhere²⁸. Briefly, an amplified Ti-Sapphire laser system (Spectra-Physics, Spitfire) provides 800 nm wavelength pulses of 35 fs pulse duration and 5 kHz repetition rate... During the OPTP measurements, the THz emitter, EO crystal, and samples are kept under vacuum at pressures below 10^{-1} mbar. For OPTP measurements, samples were spin coated CuSbSe_2 thin films on 2 mm thick circular z-cut quartz substrates with 1.3 cm diameter.”

“Hall effect measurement. Samples for Hall effect measurements were prepared according to the van der Pauw method. 100 nm thick gold was evaporated onto four corners of each CuSbSe_2 film sample as metal contacts ($0.2 \text{ cm} \times 0.2 \text{ cm}$ size). The substrate was a $1.2 \times 1.2 \text{ cm}^2$ glass substrate. Then the

gold contacts were wired to the system for measurements. Then the gold contacts were wired to the system for measurements. Hall effect measurement at room temperature (300 K) was performed in air with the Lake Shore 8400 Series under 1 T magnetic field. Ohmic check was run before the Hall effect measurements to make sure the quality of metal contacts and electric connections was good.

For Hall effect measurements at lower temperatures, the same sample geometry was used. The measurements were carried out in a 16 T superconducting magnet with temperature ranging from 1.3 to 300 K in a helium gas environment. The samples were mounted on the probe in the magnetic fields perpendicular to the ab-plane and glued with a GE-varnish. A combination of silver paste and silver wires was used to make the electrical connection. After drying the paste, it was confirmed that the contact resistances were acceptable within the order of a few ohms. The longitudinal (ρ_{xx}) and transverse resistivity (ρ_{xy}) were obtained using the van der Pauw technique with a current amplitude of 0.1-1 mA, and an alternating frequency of 3-17 Hz with the help of SR830 digital lock-in amplifier. The perpendicular magnetic fields were swept at a rate of 0.5 T min^{-1} at a given temperature. All data were taken over a full range from -16 T to 16 T, averaged in positive and negative fields to remove a small longitudinal resistance contribution to the measured voltage which may arise from the van der Pauw geometry, and retain only the antisymmetric voltage component due to the Hall effect.”

3. Why does XRD data not appear so clean, why is there so much of background, there seem to be many other peaks! The data is not really a reflection of a high quality sample. Have you identified all the peaks, Please provide complete information.

Response 1.3: We apologise for the confusion, which may have come about from the small size of the figure, and the high number of peaks in the reference pattern of phase-pure CuSbSe_2 . The original X-ray diffraction pattern was taken relatively quickly over just 10 min, and this resulted in low resolution in the background of the scan. We have therefore re-made our CuSbSe_2 samples and retaken the measurements with much finer step size (increased from $0.05^\circ/\text{step}$ originally to $0.01^\circ/\text{step}$ now) and longer total acquisition time (25 minutes). In addition, we used a different diffractometer (Bruker D8 Advance system now, rather than the Bruker D2 phaser system originally). As shown in Fig. R1 below, we obtained a much cleaner background that follows the expected polynomial shape, as well as better-resolved peaks using the Bruker D8 Advance system. The background is now sufficiently clear for us to observe the minor peaks due to diffraction from K_β radiation (*) from the X-ray source, as well as the L_α radiation from W deposits on the Cu anode (#). We also can observe the (200) peak from the single crystalline silicon substrate (α), which has an associated W L_α line. Please note that we used single crystal silicon as the substrate in order to minimize the broad background common to amorphous (glass) substrates. These silicon peaks are sharper than the peaks from CuSbSe_2 , which is consistent with the single crystalline nature of the substrate. In the close-up of the diffraction peaks shown inset in Fig. R1a, we can also observe the reflections due to the $K_{\alpha 1}$ and $K_{\alpha 2}$ radiation from the X-ray source.

To quantify how well we can account for the peaks, we fit performed a profile fit on the new diffraction pattern, and excluded the two silicon substrate peaks. Using then the reference pattern of CuSbSe_2 (obtained from the Inorganic Crystal Structure Database, collection code: 418754), and accounting for diffraction from $K_{\alpha 1}$ and $K_{\alpha 2}$ radiation, we obtained a very close fit, with a goodness of fit of 1.18 (Supplementary Fig. 1). Without excluding the two silicon substrate peaks, we still obtained a close fit with a goodness of fit of 1.58, and the residuals then were dominated by the two silicon peaks.

We carefully considered whether any of the peaks could be due to Cu_3SbSe_3 , Cu_2Se or Sb_2Se_3 , and could not find any peak that matched any of these impurities. We can therefore conclude that the optimised CuSbSe_2 films are phase pure. Please refer to Supplementary Fig. 1c that compares films prepared with our optimised heat treatment that was phase-pure versus sub-optimal heat treatment that had Cu_3SbSe_3 impurities in it.

Figure R1. Comparison between the diffraction pattern of CuSbSe_2 taken on two different diffractometers. a, Bruker D8 Advance system (new measurements, made for this revision), obtained using a step size of $0.01^\circ/\text{step}$, taken over 25 min. All CuSbSe_2 peaks are indicated by dashed lines. We also indicate the very minor peaks due to reflections from K_β (*) and W L_α (#) radiation from the X-ray source, along with peaks from the crystalline silicon substrate (α) and the associated W L_α reflection ($\alpha_\#$). The inset shows the pattern and the peaks from the silicon substrate (α) and the associated W L_α reflection ($\alpha_\#$) within $2\theta = 30^\circ - 35^\circ$; **b**, Bruker D2 Phaser system (original measurements), obtained using a step size of $0.05^\circ/\text{step}$, taken over 10 min. The Miller indices of the dominant peaks are indicated.

Below is a comparison of Pawley fits of the two patterns taken by two different diffractometers:

Figure R2. Comparison of the fitting of the X-ray diffraction measurements of CuSbSe_2 taken using two different diffractometers. Fit to the diffraction data taken **a**, using the Bruker D8 Advance system, and **b**, using the Bruker D2 Phaser system. The reference pattern was obtained from the Inorganic Crystal Structure Database, collection code: 418754.

We have replaced Fig. 1b in the main text with the new measurements shown in Fig. R1a. We have also changed the Supplementary Information, such that the first figure shows the detailed analysis of the diffraction pattern, as well as the Pawley fitting in Fig. R2a.

4. What were the ramp rate and cooling rate before and after annealing? Also, annealing is a treatment which is often used for metals (for relieving stress and changing grain size), not for thin films, it should be labelled as heat treatment because it does lead to change in phase and crystallization. This is a wrong terminology. Even if whole world uses it, it is fundamentally incorrect.

Response 1.4: The details of ramp and cooling rates have been added to Methods, as below:

“After spin coating, the sample was thermally treated on a hot plate at 100°C for 10 min (ramp rate $30^\circ\text{C min}^{-1}$). The sample, together with the hot plate, was then allowed to passively cool to room temperature. The cooling rate was estimated to be 5°C min^{-1} . All of the above processes, except substrate cleaning, were performed in a N_2 -filled glovebox, where the H_2O and O_2 levels were monitored and kept low ($\text{H}_2\text{O} < 0.1\text{ ppm}$; $\text{O}_2 < 5\text{ ppm}$). When the thermal treatment was completed, the sample was taken out of the glovebox and placed into a quartz tube for further heat treatment. The tube was firstly pumped to a pressure of $\approx 50\text{ mTorr}$, then filled with Ar to reach a pressure of $\approx 1200\text{ mTorr}$. Then the sample was heated to 400°C (ramp rate $60^\circ\text{C min}^{-1}$) and kept for 2 min, then cooled down naturally (estimated cooling rate: $10^\circ\text{C min}^{-1}$) to obtain phase-pure CuSbSe_2 thin films (refer to Supplementary Fig. 1 for the X-ray diffraction patterns and phase-purity analysis).”

We have corrected the terminology regarding heat treatment throughout the whole paper as the reviewer suggested.

5. I doubt if you can use Raman and FTIR for phase analysis in a complete sense. This is rather incorrect and inappropriate way to use these techniques. Raman is not exactly a fingerprinting technique unlike XRD, its spectra depend on a variety of factors. This is a very basic of materials characterization.

Response 1.5: We thank the reviewer for this comment. Our main purpose for using Raman spectroscopy and FTIR was to identify the dominant Raman- and IR-active phonon modes present in the material. This is important for understanding what phonon modes charge-carriers can couple to in CuSbSe_2 , and provides experimental data to compare against our computations of the phonon dispersion curve. We have therefore reworked the discussion in this section to not detract from the main focus of the paper, as detailed below:

“Raman and Fourier transform infrared (FTIR) spectroscopy measurements were employed to determine the dominant optical phonon modes present in CuSbSe_2 . For the $Pnma$ space group (D_{2h}^{16}), there are four Raman active mode symmetries (A_g , B_{1g} , B_{2g} , and B_{3g}), along with three IR active mode symmetries (B_{1u} , B_{2u} , and B_{3u}) ...”

6: I don't think there is enough evidence to back this statement, the sample quality isn't as great as is being claimed: "Combining these XRD, Raman and FTIR measurements, we can conclude that the spin-coated CuSbSe_2 thin films are phase-pure after annealing at 400 °C for 2 min." The XRD plot itself is not an excellent one.

Response 1.6: We have answered the question in detail in Response 1.3. From taking more detailed X-ray diffraction measurements, we can only observe peaks that are fully consistent with $Pnma$ CuSbSe_2 , and we have a very good fit of the reference pattern to our measured pattern. The background of the retaken diffractogram also shows no anomalies now. We are therefore convinced that the CuSbSe_2 , prepared with the optimized heat treatment procedure detailed in Response 1.4, was phase pure.

We have deleted the sentence quoted by the reviewer. The determination of phase purity is now only based on the X-ray diffraction measurements. However, we can still say that the phonon modes we observed from Raman and FTIR are consistent with the přibramite phase of our CuSbSe_2 films. We have therefore kept this sentence from page 13 of the main text:

“These Raman and FTIR measurements are therefore consistent with the phase-purity of the spin-coated CuSbSe_2 thin films prepared after the heat treatment at 400 °C for 2 min.”

7. Why is the absorbance range so narrow? What was the measurement range?

Response 1.7: We have now included the absorbance measurements of CuSbSe_2 thin films obtained over a wider range of photon energies from 1.1 to 3.0 eV (Supplementary Fig. 6a). In Fig. 2a, our aim was to determine whether CuSbSe_2 is excitonic by fitting the Elliott model to the absorption edge. This model accounts for the continuum of a conventional semiconductor, superimposed with the absorption due to one excitonic peak. It does not

account for any intra-band transitions or higher-order excitonic states. To avoid these factors affecting the fitting, it was important to focus on the narrow range of energies close to the absorption edge. At the same time, we agree that it is useful for readers to see the absorption spectrum of CuSbSe₂ over a wider photon energy range, and we have therefore now included this (as below).

Supplementary Fig. 6 | Optical absorption properties of CuSbSe₂. a, Absorbance of three CuSbSe₂ thin film samples over a broad photon energy range. The three samples were prepared with identical parameters, and were heat treated at 400 °C following our optimized procedure, as described in the main text. The peak at 1.37 eV (900 nm wavelength) was caused by a change in the detector in the instrument...

8. On page 26, the authors mentioned “We posit that quasi-2D structures with interlayer void space exhibits reduced deformation potentials compared to 3D structures, as strain-induced changes in bond length can be compensated by modulating the interlayer-spacing.” I would like the authors to elaborate on the origin of strain in CuSbSe₂ material and how the authors have controlled the degree of strain produced along the principal crystallographic axes? There is not so straightforward correlation as being claimed. You also need to provide better justifications.

Response 1.8: We apologise to the reviewer for the confusion here. In this part of the discussion, we are aiming to understand the cause of the low deformation potentials that we found in CuSbSe₂. Deformation potential refers to the changes in the electronic structure as a material’s unit cell is distorted. In order to predict the deformation potential using DFT, we need to apply strains to a simulated unit cell. To be clear: in this quoted passage, we are not physically applying any strains to the material, we are distorting the input structure used in our computational studies.

The theory of deformation potentials, as originally presented by Bardeen and Shockley in 1950¹⁰, aims to describe the effect that a long wavelength acoustic wave has on electronic structure of a semiconductor as it propagates through the material, and how shifts in band positions caused by this propagating wave lead to a carrier-phonon coupling element. The core assumption of this theory is that the wavelength of the propagating wave is large compared to the characteristic size of the unit cell, and so the propagating wave can be described by a

homogenous strain. When discussing strain in CuSbSe₂, we are implicitly referring to the propagation of an acoustic phonon through the material. This has been clarified in the main text now, and the concept is now illustrated in a new Supplementary Fig. 9, copied below for convenience.

We have now modified the results section to better explain the principles behind deformation potential theory on page 23:

“To understand the delocalized nature of charge-carriers, we used deformation potential theory¹⁰², which aims to describe the effect that a long wavelength acoustic wave has on the electronic structure of a semiconductor as it propagates through the material. As the wavelength of the propagating wave is large compared to the characteristic size of the unit cell, it can be described by a homogenous strain (please refer to Supplementary Fig. 9). We can use this assumption to describe the first order scattering potential of any long-wavelength acoustic phonon as the change in band edge position as we apply strain to a structure via a quantity known as the acoustic deformation potential (E_d^{nk}), which is described by Eq. 1:

$$E_d^{nk} = \frac{\delta \epsilon_{nk}}{\delta S_{\alpha\beta}} \quad (1)$$

In Eq. 1, ϵ_{nk} is the energy of band n at wavevector \mathbf{k} , and $S_{\alpha\beta}$ is the uniform stress tensor.”

Supplementary Fig. 9 | Illustrating the effects of acoustic waves in crystals. A long wavelength acoustic phonon propagating through a material can be approximated locally as a homogenous strain to the unit cell. This is one of the key assumptions made in deformation potential theory⁴⁴.

In addition, we modified the quoted text to the following on page 29 of the main text:

“The magnitude of the deformation potential is substantially reduced by structural relaxation in this flexible crystal structure, as outlined in Supplementary Note 7. We posit that quasi-2D structures with interlayer void space exhibits reduced deformation potentials compared to 3D structures, as strain-induced changes in bond length (due to the propagation of an acoustic wave) can be compensated by modulating the interlayer-spacing.”

9. For calculating the acoustic deformation potential, a more detailed explanation is needed. Current details are not sufficient.

Response 1.9: The acoustic deformation potentials were calculated within the framework laid out by Wei and Zunger¹¹. While the original approach considered absolute deformation potentials with respect to an isotropic change in volume for cubic structures, we consider anisotropic deformation with respect to an arbitrary homogeneous strain element S_{ij} . These anisotropic deformation potentials (E_{ij}^n) have been successfully applied to a wide range of semiconductor materials to model acoustic deformation potential scattering¹². A diagram has been added to the Supplementary Information to illustrate this process.

Supplementary Fig. 10 | Schematic of calculation of acoustic deformation potential of band n at point k using a modified method of Wei and Zunger⁴⁵. **a**, Plot of band energy $\epsilon_{n,k}$ against homogenous strain $S_{\alpha\beta}$. The acoustic deformation potential is defined as the change in $\epsilon_{n,k}$ with respect to $S_{\alpha\beta}$. **b**, Change in band edge energy of a simple parabolic band under axial compression (red) and expansion (blue). **c**, Uniaxial compression (red) and expansion (blue) of a BCC unit cell. Note, it is implicitly assumed that the change in energy of core levels is negligible, and that the band edges are in equivalent electrostatic reference frames

We have now modified the theory part of the results section (on page 23) to capture the essence of how we calculated the acoustic deformation potential. We also added the schematic above to the Supplementary Information, and direct readers to this for further details just beneath Eq. 1 in the main text:

“We can use this assumption to describe the first order scattering potential of any long-wavelength acoustic phonon as the change in band edge position as we apply a strain to a structure via a quantity known as the acoustic deformation potential (E_d^{nk}), which is described by Eq. 1:

$$E_d^{nk} = \frac{\delta \epsilon_{nk}}{\delta S_{\alpha\beta}} \quad (1)$$

In Eq. 1, ϵ_{nk} is the energy of band n at wavevector \mathbf{k} , and $\mathbf{S}_{\alpha\beta}$ is the uniform stress tensor¹⁰⁰. Please refer to Supplementary Note 7 for more details on how we calculated the acoustic deformation potential, especially Supplementary Fig. 10 and the associated discussion beneath it.”

10. A comprehensive explanation of the charge carrier mobility variation based on the type of polarons produced in the CuSbSe₂ system is needed and will be critical in understanding the role of polarons in dictating the charge carrier dynamics in these systems. Current details are very sketchy.

Response 1.10: We believe the reviewer is mainly referring to Supplementary Fig. 5 (copied below for convenience) that we had in the original submission. We have carefully considered this point and have performed a new set of calculations to further support the conclusion that small polarons do not intrinsically form in this system, providing important computational evidence that supports our conclusions from experiment.

Supplementary Fig. 5 | Computational analysis of mobility. Calculated temperature-dependent total mobility (black spots) and mobilities when acoustic deformation potential (ADP, red upward triangles) scattering or polar optical phonon (POP, pink downward triangles) scattering dominates. The mobility values limited by ionized impurity scattering (IMP) are too high to be included in the figure.

We used state-of-the-art calculations using the *ShakeNBreak* method¹³ to explicitly model polarons in CuSbSe₂ as dilute charges in a supercell. As applied to bulk polarons, the ShakeNBreak method is effectively an evolution of the bond distortion method with the addition of a stochastic rattle to the perturbed/distorted structures. The bond distortion method has been used previously to search for low-energy polaronic states and thus calculate polaron properties such as formation energy and mobility in the dilute limit¹⁴. Perturbation of the structure is necessary, as while simply adding an extra charge (an unpaired electron or hole) to an unperturbed supercell and allowing the system to relax to its local minimum may in some cases result in spontaneous localisation of an electron/hole polaron, typically the structure must first be biased towards a polaronic configuration by distorting atoms around likely polaron sites before localization can occur. Intuitively one can think of this as overcoming the energy barrier between delocalised and localised solutions. These distorted structures are relaxed to their local minima in turn, and by comparing the energies of unperturbed and distorted (perturbed) structures, we can assess whether energy minimising distortions are present in the system. By inspecting the charge density (representative of the wavefunction) of the band

containing the unpaired electron or hole in each relaxed structure we can assess whether or not it represents a localised (0D) polaronic solution or otherwise. From our results using 64 atom supercells, we fail to find energy minimising localised solutions and conclude that polaronic states in CuSbSe₂ occupy a >1555.52 Å³ volume, which is exactly four times the volume of a unit cell volume and would therefore be classified as large polarons. These calculations were run at HSE06 level to negate self-interaction errors, which may have otherwise resulted in spurious delocalised solutions. Localization was assessed qualitatively by inspecting the partial charge density function of the band containing the unpaired electron or hole, with all solutions being > 0D by inspection. We have added the following to the Supplementary Information:

Supplementary Fig. 8 | Computations to directly determine whether small polarons form in CuSbSe₂. **a**, Addition of an unpaired hole followed by distortion around the Cu, Se1 and Se2 sites and/or rattling of the lattice results in a relatively small change in DFT total energy of about ~12 meV compared to the unperturbed state.. The partial electron density function of the unpaired hole for the **b**, unperturbed and the **c**, perturbed structures show a quasi-2D valence band as the electron density is confined to a single layer. **d**, Addition of an unpaired electron followed by distortion around the Sb site results in the formation of a metastable state 0.12 eV above the unperturbed structure. Analysis was performed on distortions around Sb because Sb 5p orbitals dominate the lower conduction band (refer to Fig. 2b in the main text). The partial electron density of the unpaired electron in the **e**, unperturbed structure shows a delocalised 3D solution, while the higher energy **f**, perturbed structure shows quasi-

1D character because the electron density is confined to ribbons running through the interlayer space. In this analysis, we considered a supercell that was four times the volume of a unit cell, and we did not observe any localisation of either the electron or hole charge density (0D character) to within one unit cell. That is, these computations support the conclusion that small polarons do not form in CuSbSe_2

Regarding the calculated variation in mobility with scattering by optical and acoustic phonons (that was Supplementary Fig. 5): after internal discussion, we have decided to remove this figure, as well as references to the calculated scattering data from the main text and Supplementary Information. The reason this data was included in the first instance is mostly historical. Many of the calculated quantities presented in this work, such as the dielectric constants, Γ -point modes, deformation potentials, and band structure, were originally calculated solely as inputs to the AMSET model. However, as we gathered more data the scope of the paper shifted further from a basic presentation of calculated/measured materials properties, and closer towards what we believe is a more interesting discussion on the strength and fundamental origins of the electron-phonon interaction in CuSbSe_2 . In this sense, while the calculated mobility data originally motivated computational work, it now falls out of the scope of the useful discussion in the paper's final form. We believe its removal is justified, as these data do not contradict the conclusions we present, however their inclusion distracts from the takeaway message of the work and has led to confusion amongst reviewers.

Furthermore, we believe that it is less confusing to readers to only read the experimentally-measured change in mobility as a function of temperature, shown in Fig. 3b in the main text. This decrease in mobility as temperature rises is fully consistent with the presence of large polarons (delocalised carriers) in our system. Small polarons would instead increase in mobility with a rise in temperature due to an increase in activated hopping transport. Therefore this experimental result helps us to verify the absence of carrier localisation in CuSbSe_2 .

Fig. 3 | Spectroscopic and temperature-dependent studies on carrier-phonon coupling in CuSbSe_2 ... b, Temperature-dependent mobility of CuSbSe_2 thin films determined using Hall effect measurements.

11. It is strange that error bars seem to be missing all over, they need to be provided.

Response 1.11: We have carefully gone back over the whole paper, and have found that it is only appropriate to add in error bars to the temperature-dependent Hall mobility results (Fig. 3b) and the texture coefficients (Supplementary Table 2), which have now been done. For all other results, this is not appropriate, as detailed below:

Fig. 1b XRD pattern: error bars are usually not provided to the measured X-ray intensities. The texture coefficients (Supplementary Table 2) are calculated based on the XRD patterns of three samples prepared with identical parameters. The small standard deviation shows the high level of similarity of the XRD patterns of these samples.

Fig. 1c Raman spectrum: error bars are usually not provided to the measured intensities. Our Raman spectrum is the average of ten scans. We have now specified this in the caption of Fig. 1c.

Fig. 1d Phonon dispersion curve: It is not typical to attach error bars here, as the error is dominated by systematic error of the underlying level of theory chosen, as opposed to machine noise. However, we can comment on benchmarks of the r^2 SCAN functional for predicting phonon dispersion. Ning *et al.* benchmarked LDA, PBE, and r^2 SCAN functionals against neutron scattering data for Si, GaAs, NiOx, and Fe. They found that r^2 SCAN described the non-magnetic materials quantitatively better than the other functionals tested. However, due to the scarcity of complete neutron scattering data, it is difficult to assign mean absolute relative errors as compared to experiment¹⁵.

Fig. 2a Optical absorbance curve and Elliott fit: we have obtained the optical absorbance curves of 3 samples, which were prepared with identical parameters (Supplementary Fig. 6a). The differences between samples can be the indication of error bars. The close results indicate the high quality of our samples and good repeatability. As for the Elliott fit, the average values were derived by performing a large number of Elliott fits. The error bar of fitted exciton binding energy was given in the main text to capture the uncertainty in the fitting:

“The deconvolution of the excitonic and continuum contributions yields a weak and broad excitonic contribution, described by an exciton binding energy (E_b) of 9 ± 4 meV.”

Fig. 2b Calculated electronic band structure of CuSbSe₂ along with electronic density of states curves and crystal orbital Hamilton population (COHP) diagram:

Similar to Fig. 1d, as the error is dominated by the systematic error of the underlying level of theory chosen and not random noise, we can only comment on the suitability of the HSE06 functional in describing the feature of interest, *i.e.*, the electronic structure. The hybrid functional, HSE06 has been found to describe key electronic structure descriptors, such as bandgap, much more accurately than PBE or other GGA functionals^{16,17}. We calculated a bandgap of 1.16 eV, which is well within the range of bandgaps reported in the literature for CuSbSe₂ (1.04 eV to 1.2 eV^{8,18-20}), and we are therefore confident about the accuracy of our calculations using HSE06.

Fig. 2c-e Short-time transient absorption (TA) results: error bars are usually not provided. Our results are the average of 5 scans, which we have now specified in the caption.

Fig. 3a Fluence-dependent optical pump terahertz probe (OPTP) transients: given the fluence-independent photoconductivity decay of CuSbSe₂, the comparison between the normalized OPTP transients under different fluences shows the level of sample-to-sample

variation. It can be observed that all curves exhibit the same decay timescale, indicating the good repeatability of our OPTP measurements.

Fig. 3b Temperature-dependent Hall mobility: error bars have been added. We thank the reviewer for pointing this out.

Fig. 4b Percentage changes in bond lengths and interlayer distance of CuSbSe₂ as a function of strain along the *c*-axis:

The systematic error in the description of structure is related to the forces in the system. Accurate modelling of phonon dispersion is often used as a benchmark for accuracy of forces. As discussed for Fig. 1d, the r²SCAN functional is expected to model forces in the system well, and the good agreement of predicted Γ -point mode frequencies with Raman and IR measurements in our system supports this conclusion.

Discussion is very hand-waiving without much convincing evidence. Overall, I think manuscript requires substantial improvement.

Response 1.12: We would like to take this opportunity to clarify and explain the points made in our discussion section.

The core motivation of this paper is to reveal the chemical-physical factors that enable heavy-pnictogen-based perovskite-inspired materials to avoid carrier localisation, which has been a severe challenge for these emerging solar absorbers. Carrier localisation means that the radial extent of the wavefunction of the electron/hole is restricted to within a unit cell, such that instead of charge-carriers being transported through drift/diffusion, they are transported through hopping between lattice sites. This leads to a substantial reduction in mobilities and diffusion lengths. Therefore, understanding the underlying principles that allow this problem to be overcome is of significant importance.

Carrier localisation takes place due to strong electron-phonon coupling. This can be due to strong coupling between charge-carriers and longitudinal optical (LO) phonons (Fröhlich coupling), or with acoustic phonons. The strength of coupling to LO phonons is described by the Fröhlich coupling constant (α , refer to Eq. 3 in the main text), and $\alpha > 10$ is considered to be in the strong coupling regime. The strength of coupling to acoustic phonons can be described by the acoustic coupling constant, g_{ac} (Eq. 2).

In the results section of the paper, we show that there is no carrier localisation in CuSbSe₂ because 1) the photoconductivity transient, measured by OPTP, decays too slowly (over 50 ps, whereas materials with localisation would decay within 1 ps), and 2) the mobility decreases with increases in temperature, consistent with LO phonon scattering. Self-trapped carriers would instead have their mobility increase with temperature as hopping is thermally activated. Also in the results section, we show computationally that both the Fröhlich and acoustic coupling are weak (Table 1). The latter is due to the low deformation potentials, which g_{ac} strongly depends on (refer to Eq. 1).

The aim of the discussion section is to therefore understand why these are the case for CuSbSe₂ in order to extract some chemical-physical principles that may be more generally applicable to other materials classes.

The discussion is structured to cover why 1) deformation potentials are low, and 2) Fröhlich coupling is weak in CuSbSe₂.

For the first point, we show that:

- Deformation potentials are low because the relaxation of a strained structure from the propagation of acoustic waves through the lattice is mostly in the interlayer gaps rather than through changes in the bond length. As discussed in Response 1.8, deformation potential describes how the band positions change in energy following strains along the principal axes. These strains increase in magnitude as the population of acoustic phonons are increased. Our analyses are rigorously performed using hybrid-DFT. These calculations are corroborated by analyses of the nature of bonding at the band extrema through Crystal Orbital Hamilton Projection (COHP) calculations, which show that the Sb-Se bonds are strong, which accounts for their negligible changes in bond length following distortion (see page 28 of the main text)
- The electronic dimensionality of CuSbSe₂ is high at both the valence and conduction band extrema. Through computations of Fermi iso-surfaces within the band edges, we show that while the valence band maximum is 2D, the conduction band minimum is nearly 3D. This is important, since 3D electronic dimensionality results in a barrier to carrier localisation, which can help to enable band-like transport. For 2D electronic dimensionality, carrier localisation is also not thermodynamically favourable if the g_{ac} is well below 1, as is the case here. The higher electronic dimensionality of the conduction band is due to electronic coupling between Sb and Se across the interlayer gaps.

For the second point, we show that the weak Fröhlich coupling is because of 1) a high electronic dielectric contribution along all principal axes, and 2) a low ionic dielectric contribution, especially along the *a*- and *c*-axes. If we examine Eq. 3 (copied below), a low ϵ_{stat} relative to ϵ_{∞} will result in a reduced α .

$$\alpha = \frac{e^2}{4\pi\epsilon_0} \left(\frac{1}{\epsilon_{\infty}} - \frac{1}{\epsilon_{stat}} \right) \sqrt{\frac{m^*}{2\omega_{LO}\hbar^3}} \quad (3)$$

To understand the underlying reasons, we analysed the Born effective charge tensors for different sub-lattices in CuSbSe₂, as detailed in Table 2 of the main text. We found that the sum of the Born effective charges along the principal axes are mostly not substantially higher than the formal charges of the species (unlike lead-halide perovskites), showing that there would not be a high ionic dielectric contribution. The high electronic dielectric contribution is due to the small bandgap of CuSbSe₂ (since $\epsilon_{\infty} \propto E_g^{-0.5}$) and high density of states near band-edges (see page 32 for a detailed discussion).

Overall, from these detailed discussions, we can suggest the following factors as being conducive to materials with delocalised charge-carriers:

- Gaps being present in the crystal structure, into which distortions to the lattice due to the population of phonon modes could be relaxed with minimal changes to bond lengths, minimising deformation potentials
- Electronic coupling between atoms across the interlayer gaps, such that the electronic dimensionality at band-edges is 2D or higher
- Low ionic dielectric contributions compared to the electronic contributions by having a small bandgap and species without Born effective charges substantially above the formal charges of the elements

These open up the future design of perovskite-inspired materials with band-like transport, and the next step would be to test these design principles across a wide range of materials both computationally and experimentally. We would look for materials that have a layered structure (similar to CuSbSe_2 and BiOI), or possibly a quasi-1D structure (such as Sb_2S_3 and similar materials). At the same time, we would look for materials that have electronic coupling across layers/ribbons, such that the electronic dimensionality is 2D or higher at both VBM and CBM in order for there to not be a thermodynamic driving force for carrier localization. This can occur if the species contributing to the orbitals at band-edges are next to each other across gaps. For example, if we look at CuSbSe_2 (Fig. 1a, copied below), Sb in one layer is next to Se in the other layer. This allows Sb-Se interactions across the gap. Since Sb-Se antibonding orbitals comprise the CBM, this allows 3D electronic dimensionality in the CBM. Flipping (hypothetically) alternating layers in CuSbSe_2 could therefore be detrimental and reduce the electronic dimensionality in the CBM. Materials with similar structural features (e.g., in BiOI) could therefore similarly have high electronic dimensionality. Finally, we would need to carefully consider the effects of lone pairs from the pnictogen cation, which, if expressed, can lead to high Born effective charges along particular directions. From these guiding design principles, we could identify possible materials. But, importantly, we can make use of the computational and experimental tools used in this paper to test how promising the proposed materials are for exhibiting band-like transport.

Fig. 1 | Structural and phonon properties of CuSbSe_2 . **a**, Crystal structure of CuSbSe_2 , viewed along the b axis, and with the dominant A_g Raman mode shown in red arrows. The bonding environments of Cu and Sb are illustrated below the crystal structure.

Therefore, realising the wide impact of our work requires the thorough and rigorous analyses we made in the discussion section that explain the underlying factors behind the experimental and computational observations we made in the results section.

To make these points clearer, we replaced the overarching sub-heading “*mechanisms for weak charge-carrier-phonon coupling in CuSbSe_2* ” that was in the original discussion section with three sub-sections to more clearly to guide readers through our points made. The sub-section headings added are:

- “*Understanding the cause of low deformation potentials in CuSbSe_2* ”
- “*High electronic dimensionality in CuSbSe_2* ”
- “*Understanding weak Fröhlich coupling in CuSbSe_2* ”

To better set out the key hypotheses that we rigorously test in the discussion section, we changed the opening of the discussion on page 27 to:

“*In light of the results presented, it can be seen that carrier localization is not present in CuSbSe_2 , and that this is due to the weak coupling between charge-carriers and both acoustic and optical phonons. This is unusual among heavy pnictogen-based perovskite-inspired materials, and it is critical to unravel*

the underlying factors. In this discussion, we will examine computationally how the structure and bonding in CuSbSe₂ result in the low deformation potentials and low Fröhlich coupling constants, as well as the effect on the electronic dimensionality in both the valence and conduction bands.”

Our conclusions already capture the three key points made above:

“Based on our investigations, we propose that the free volumes (e.g. interlayer gaps) in the lattice can help to minimize the effect of lattice distortions on the bonding environment and lower the deformation potential. At the same time, electronic coupling across the interlayer gap between species contributing to the band-edge density of states can increase the electronic dimensionality, which reduces the likelihood of self-trapping. Finally, materials with low ionic contributions to the dielectric constant are desired to minimize Fröhlich coupling, but this needs to be balanced with the effect on the capture cross-section of charged defects. These insights are valuable for the future design of solar absorbers that have band-like transport.”

Reviewer 2:

This article effectively demonstrates that CuSbSe₂, a heavy pnictogen-based chalcogenide, can prevent charge carrier localization at the greatest extent, leading to a significant superiority over other pnictogen-based materials. The authors utilize both experimental methods and computational simulations to provide insightful observations. The thiol-amine solution processing method employed for obtaining phase-pure CuSbSe₂ thin films demonstrates practical feasibility. The photoconductivity decay timescale and temperature-dependent Hall effect measurements confirm the presence of large polarons in CuSbSe₂. The paper employs density functional theory (DFT) calculations to delve into the acoustic and Fröhlich coupling constants, revealing them to be lower compared to many other heavy pnictogen-based materials. The low deformation potentials and relatively high electronic dimensionality contribute to weak coupling to acoustic phonons, and the weak Fröhlich coupling arises from the small bandgap and low ionic contribution to dielectric constants.

This research holds significance for the design and optimization of solar absorbers. The unique characteristics of CuSbSe₂, particularly its ability to resist charge carrier localization, position it as a potential solar absorption material. This contributes to the advancement of more efficient photovoltaic materials. I recommend this work be published after the following concerns are addressed:

We are grateful to the reviewer for their time in carefully evaluating our paper, and for their strongly positive appraisal. We have now addressed the three comments raised, as detailed below.

1. How does the preparation method of CuSbSe₂ affect its performance? Was the composition of precursor optimized specifically to gain the pure-phase compound? Can this preparation method be extended to similar materials?

Response 2.1: Indeed, we carefully optimized the heat treatment of CuSbSe₂ to obtain a phase-pure material to study in depth experimentally, so that we can directly compare these measurements with calculations made on the same phase. The detailed discussions of the effect of heat treatment temperature on the crystallinity and phase purity of our CuSbSe₂ thin films are provided in Supplementary Note 1, and are especially well captured in Supplementary Fig. 1c.

The concentration of the precursors used was according to the stoichiometry of CuSbSe₂ (the molar ratio of Cu₂Se:Sb₂Se₃ = 1:1), and we did not optimize this stoichiometry further after obtaining phase pure thin films. Indeed, the thiol-amine synthesis method is broadly applicable to chalcogenides. We wrote in the main text that this work is the first report of developing a thiol-amine synthesis route for the preparation of phase-pure thin films of CuSbSe₂. We took inspiration from the recent literature on the use of this alkalest solvent to prepare films from other chalcogenide compounds, such as Cu₂ZnSn(S,Se)₄^{21,22}, Cu(In, Ga)Se₂^{2,5} and CuIn(S, Se)₂²³. A review on the thiol-amine system summarizing the elements and compounds that could be dissolved by this system is provided in Ref. 24. Given the success of the synthesis of the compounds mentioned above, as well as our work on CuSbSe₂, and the fact that a quite wide range of materials can be dissolved by the thiol-amine solvent system, we believe that this method has good potential to be extended to wide range of materials.

We have captured these points in the results section, and have added a reference to the optimization details in Supplementary Note 1:

“Previous efforts at growing CuSbSe₂ focussed on vacuum-based approaches (e.g., sputter deposition⁷⁵, close-space sublimation⁷⁶), methods that have long reaction times (e.g., fusion method⁷⁸ or selenization of metal precursors⁷⁸), or processes involving the use of toxic precursors (e.g., hydrazine solvent^{41,79}). Solution-processing is advantageous in requiring less capital-intensive equipment than vacuum-based processing⁸⁰⁻⁸², but at the same time, it is critical to avoid the use of toxic solvents⁸³. More recently, a more benign solvent system than hydrazine, comprised of a thiol-amine mixture, has been found to be effective in dissolving chalcogenide precursors and successfully used to deposit absorber layers of photovoltaic devices, such as Cu₂ZnSn(S,Se)₄^{84,85}, Cu(In, Ga)Se₂^{86,87} and CuIn(S, Se)₂⁸⁸. In this work, we develop a novel thiol-amine-based solution processing route by mixing the Cu₂Se and Sb₂Se₃ solutions together then achieving phase-pure CuSbSe₂ thin films for the first time, as detailed in Methods. To achieve crystalline films, we dried the films at 100 °C for 2 min in a N₂-filled glovebox, before crystallizing at 400 °C for 2 min in a tube furnace filled with Ar (~1200 mTorr pressure). The details of the optimization of the thiol-amine processing route for CuSbSe₂ are in Supplementary Note 1.”

2. How is the stability of CuSbSe₂? Have the authors done relevant investigations? As far as I am concerned, the valence states of cations demonstrate an unstable state of this compound.

Response 2.2: We thank the reviewer for this question. Indeed, Cu is in the +1 oxidation state, so it is valid to ask whether Cu remains in this oxidation state, or whether it oxidizes to the +2 state over time when left in air. Sb and Se are both in their stable +3 and -2 oxidation states, respectively. We therefore used X-ray photoelectron spectroscopy (XPS) to examine the oxidation states and bonding environments of the cations of CuSbSe₂ thin films as-made, and after 3 weeks of storage in air. We focussed on the cations in this XPS investigation, since the reviewer asked specifically about the stability of the cations.

We obtained the survey spectra, Cu 2p core levels and Cu LMM Auger spectra, as well as the Sb 3d core levels. All thin film samples were prepared using the optimized solution processing and heat treatment procedure, as described in Methods. After heat treatment, the aged samples were stored in ambient air (room temperature and approximately 80% relative humidity) for 3 weeks before measurements.

As shown in Supplementary Fig. 4a and b, after three weeks of storage, the Cu 2p core levels and Cu LMM Auger spectra of the aged CuSbSe₂ thin films had no obvious change compared to the as-prepared samples. To confirm the valence of the Cu species, the satellite found in the Cu 2p spectra is usually used²⁵⁻²⁸. As Supplementary Fig. 4a shows, our Cu 2p spectrum exhibited weak satellite signals, indicating the Cu species in our CuSbSe₂ thin films to be Cu(I). Moreover, the Cu LMM Auger spectra of our CuSbSe₂ thin films (Supplementary Fig. 4b) showed similar features to the spectra of other Cu(I) species in different compounds²⁹, with the kinetic energy of the peak fitted to be 917.7 ± 0.2 eV, which is close to the reported kinetic energy of Cu LMM peaks in Cu₂Se (917.5 eV) and CuAgSe (917.6 eV)³⁰. These results show that the Cu(I) species in this material are indeed stable, and remain in the same tetrahedral environment over time.

Supplementary Fig. 4 | XPS spectra of CuSbSe₂ thin films (as-prepared and aged). a, Cu 2p core levels and b, Cu LMM Auger spectra of as-prepared (black line) and aged CuSbSe₂ samples (red line).

The Sb 3d core level spectra are shown in Supplementary Fig. 5 (copied below). Each spectrum is comprised of a doublet with binding energies 538.2 eV (3d_{3/2}, red line) and 528.8 eV (3d_{5/2}, red line). These binding energies, as well as the separation of 9.4 eV between them, are consistent with Sb(III)³¹. Prior reports of CuSbSe₂ grown by close-space sublimation and hot injection show similar doublets in their Sb 3d spectra^{32,33}.

In analyzing the Sb 3d core level measurements we obtained, we found that the spectra for both the fresh and aged samples could be fit with Sb bonded to Se (main peak, red line) and O (purple line). After 3 weeks of storage in ambient air, the O peak increased in size and slightly shifted in position, while the peak due to Sb bonded to Se remained at the same energy. The Sb 3d_{5/2} peak also overlapped with the appearance of an O 1s peak, indicating that Sb-O species formed at the surface of the CuSbSe₂ film.

The position of the Sb-O species in the Sb 3d_{3/2} peak has a binding energy of 539.5 eV, which is closer to the Sb 3d_{3/2} peak reported for Sb₂O₃ (539.8 eV) than for Sb₂O₅ (540.4 eV)³⁴. This confirms that the Sb in our films remained in the +3 oxidation state, and likely formed Sb₂O₃ on the film surface after ageing in air.

Supplementary Fig. 5 | XPS spectra of CuSbSe₂ thin films for Sb 3d. Measurements and fitting for **a**, as-prepared and **b**, aged CuSbSe₂ samples.

To capture these points, we have added the following to the main text, in the results section on page 14:

“Finally, we note that CuSbSe₂ contains Cu in the +1 oxidation state, whereas the +2 oxidation state is typically more thermodynamically stable under ambient conditions. We therefore examined the chemical stability of the cation species in the optimized CuSbSe₂ films by X-ray photoelectron spectroscopy, as detailed in Supplementary Note 3. We found from the Cu 2p core levels and LMM Auger peaks that Cu remained in the +1 oxidation state after storage in ambient air (with approximately 80% relative humidity) for 3 weeks, and Sb also remained in the +3 oxidation state. However, we found that a layer of oxide (likely Sb₂O₃) formed on the surface of the films after storage in air, whereas there was no evidence of cuprous oxide or hydroxide species, showing Cu(I) to remain stable in its tetrahedral environment in the structure.”

In addition, we created a new Supplementary Note 3, which details our XPS analysis, and also includes our methodology for collecting the XPS measurements.

3. What was the substrate material for depositing CuSbSe₂? Was it conductive or non-conductive? Have the authors investigated any applications of CuSbSe₂ on photovoltaic or photo-electrochemistry?

Response 2.3: We have added the details of the substrate materials used to the Methods section (highlighted in yellow). For the reviewer’s convenience, here is a summary:

- XRD: 1.2 × 1.2 cm² single crystal silicon substrate;
- Raman spectroscopy: 1.2 × 1.2 cm² glass substrate;
- FTIR spectroscopy: 7.5 × 2.5 cm² single crystal silicon substrate;
- UV-vis: 2 mm thick circular z-cut quartz substrates with 1.3 cm diameter (main text) or 1.2 × 1.2 cm² glass substrate (supplementary information)
- PDS: 2 mm thick circular Spectrosil® 2000 quartz substrates with 1 cm diameter;
- TA measurements: 1.2 × 1.2 cm² glass substrate;
- OTP measurements: 2 mm thick circular z-cut quartz substrates with 1.3 cm diameter.
- Hall effect measurements: 1.2 × 1.2 cm² glass substrate.

Thus, all substrates used for characterization were non-conductive substrates. For OTP, we used z-cut quartz rather than glass because glass absorbs terahertz radiation.

CuSbSe₂ has previously been investigated for photovoltaic applications, and currently the highest power conversion efficiency (PCE) is 4.7% for CuSbSe₂ films synthesized by co-sputtering³⁵. The best solar cell based on solution-processed CuSbSe₂ achieved 2.70% PCE, where hydrazine was used as the solvent³⁶. So far, no applications of CuSbSe₂ for photo-electrochemistry has been reported, as far as we are aware. This could be another promising direction of research for CuSbSe₂.

At the same time, we would like to emphasise that applications on photovoltaic and photo-electrochemistry are not the focus of this paper. The main novelty of this work is in the new insights into the chemical and physical factors that enable pnictogen-based perovskite-inspired materials to avoid carrier localization. By using a combination of advanced spectroscopic tools and computations, we show how the structure and combination of chemical species in CuSbSe₂ lead to low deformation potentials and weak Fröhlich coupling (see Conclusions for full details). We therefore did not wish to discuss the applications of CuSbSe₂ too much, since the greatest impact of this work comes from these chemical-physical insights that could be more widely applied to identify other classes of heavy pnictogen-based semiconductors that avoid carrier localization, as captured in the final sentence of our conclusions (copied below), and as the review noted in the opening of their appraisal.

“These insights are valuable for the future design of solar absorbers that have band-like transport.”

Reviewer 3:

Y. Fu et al performed an interesting joint experimental and computational study of carrier localization in CuSbSe₂. Unlike other Sb- and Bi-based inorganic semiconductors, they have demonstrated that CuSbSe₂ exhibits delocalized free carriers through mobility measurements as a function of temperature and optical pump terahertz probe spectroscopy; those results together with the solution processing method are the most important contributions.

The experimental results are supported by calculations of the mobility, acoustic deformation potential, and Fröhlich coupling constant.

We would like to begin by thanking the reviewer for their time in evaluating our paper, and for recognising the significance of our results in showing how pnictogen-based semiconductors with band-like charge-carrier transport could be designed, which will be critical for the field of emerging inorganic solar absorbers. Below, we have addressed each comment in detail.

However, the level of theory used to evaluate mobilities and electron-phonon coupling strengths is not at the level of the state-of-the-art. For example, the calculation of the deformation potential relies on the rigid ion approximation whose validity in several systems is questionable. Similarly, the calculations for temperature-dependent mobilities in the supplemental material noticeably overestimate the experimental values, an aspect that lacks clarification in the manuscript. Additionally, it seems the authors might have overlooked important first-principles studies in calculating electron-phonon coupling, mobilities, and polarons in other compounds. The association between mobilities and large polarons is less convincing due to a lack of clear evidence for polaron formation in these systems.

Response 3.1: We thank the reviewer for making these comments. We have made detailed responses below to their points. To help the reviewer, we would like to guide them to the specific responses that cover the points they made above in their overview statement in detail:

- Level of theory used: Response 3.15
- Calculation of deformation potential: Response 3.11
- Calculated temperature-dependent mobilities: Response 3.7, with more details in Response 3.12
- Comparison of our materials with other studies: Response 3.14

We thought it could be helpful to the reviewer if we provide a concise answer to all of these questions here, but we do encourage them to go over our detailed responses in each of these sections below.

We carefully evaluated the level of theory used, striking a balance between accuracy and computational cost. In question 3.15, the reviewer asked why we did not use PBE for all calculations. We did indeed find that PBE was the best level of theory for the initial structural relaxation and exploration due to its combination of accuracy and modest computational cost, however PBE does not describe the electronic structure well. Therefore, we used the more accurate HSE06 functional for properties that rely on the separation between filled and empty electronic states, such as the high-frequency dielectric constant. For force calculations/phonon dispersion curves, we found that PBE poorly described the Γ -point phonon modes, while r²SCAN provides more accurate force constants, and was therefore selected.

Regarding the deformation potential calculations, to clarify: we did not use the “rigid ion approximation”, as originally defined based on a fixed potential. Instead, we used the self-consistent potential from a frozen internal structure. We have reworded the main text on page 23, and also added a detailed note in Supplementary Note 7, to clarify this point. As we explain in Response 3.11, we evaluated the effects of allowing the atomic basis to relax following deformation, and found that there is an overall relative *decrease* in coupling strength in this case. Repeating these calculations (at HSE06 level) was therefore not deemed necessary, as we have demonstrated it would not change our conclusions. But we do make use of equilibrated structures later on when we analyze changes in bond length due to the propagation of acoustic waves (see Fig. 4 in the main text).

The deviation between the calculated and measured mobilities is likely due to effects such as optical deformation potential scattering, grain boundary scattering, or defect scattering, which are not captured in our work. Nevertheless, these calculations show that the mobilities are not limited by optical phonon or acoustic phonon scattering, which is consistent with the presence of large polarons. However, upon reflection, we decided that the deviation between these calculated values and experiment was too confusing for readers and have removed this (*i.e.*, Supplementary Fig. 5 in the original submission) from this revision.

We have added three new tables to the Supplementary Information, along with further discussion in the main text, to compare our mobility, dielectric constant, bandgap and Born effective charge with the literature, and to put the values for CuSbSe₂ into context. In doing so, we cite more of the first-principles papers on electron-phonon coupling the reviewer mentioned.

Finally, we appreciate the reviewer’s point about the lack of explicit calculations of polaron formation. We have therefore used state-of-the-art calculations using the *ShakeNBreak* method¹³, to explicitly model polarons in CuSbSe₂ as dilute charges in a supercell. As applied to bulk polarons, the *ShakeNBreak* method is effectively an evolution of the bond distortion method with the addition of a stochastic rattle to the perturbed/distorted structures. The bond distortion method has been used previously to search for low-energy polaronic states and thus calculate phonon properties, such as formation energy and mobility in the dilute limit¹⁴. Perturbation of the structure is necessary, as while simply adding an extra charge (an unpaired electron or hole) to an unperturbed supercell and allowing the system to relax to its local minimum may in some cases result in spontaneous localisation of an electron/hole polaron, typically the structure must first be biased towards a polaronic configuration by distorting atoms around likely polaron sites before localization can occur. Intuitively one can think of this as overcoming the energy barrier between delocalized and localized solutions. These distorted structures are relaxed to their local minima in turn, and by comparing the energies of unperturbed and distorted (perturbed) structures, we can assess whether energy minimising distortions are present in the system. By inspecting the charge of the band containing the unpaired electron or hole in each relaxed structure, we can assess whether distortions result in a localized (0D) polaronic solution or otherwise. From our results using 64 atom supercells, we conclude that polaronic states in CuSbSe₂ occupy a volume >1555.52 Å³, which is four times the volume of a unit cell volume and would therefore be classified as large polarons. These calculations were run at HSE06 level to negate self-interaction errors, which may have otherwise resulted in spurious delocalized solutions. Localization was assessed qualitatively by inspecting the partial charge density function of the band containing the unpaired electron or hole, with all solutions being >0D by inspection.

In regard to other methods of calculating electron-phonon coupling, we agree that this is an exciting and rapidly growing area of research, but would stress that due to the sheer number

of methods available (e.g., *ab-initio* MD, Quantum Monte-Carlo, the Special Displacement Method, the *ab-initio* theory of polarons of Sio *et al.*), an in-depth discussion of recent developments is outside the scope of this work. In relation to our choice of model, we chose Fröhlich and ADP couplings due to the fact that they are would be familiar to many readers, they are relatively low cost, and they provide strong qualitative insight into band-like systems such as the CuSbSe₂ system.

We have made the following addition to the results section of the paper to capture the above points on page 26:

“We performed state-of-the-art calculations using the ShakeNBreak method¹⁰¹ to explicitly model polarons in CuSbSe₂ as dilute charges in a 64-atom supercell (four times the volume of a unit cell), as shown in Supplementary Fig. 8. To do this, we added an extra unpaired electron or hole to the unperturbed supercell, and allowed the system relax to a local minimum. By inspecting the charge densities of electrons and holes (representative of their wavefunctions) in the relaxed structures, we found that no localized states (i.e., 0D states confined to within a unit cell) occurred. Rather, the polaronic states were delocalized over the entire supercell, and would therefore have wavefunctions well exceeding a unit cell. This supports the conclusion that small electron and hole polarons do not form in CuSbSe₂.”

In addition, we have added the following to Supplementary Note 7.

Supplementary Fig. 8 | Computations to directly determine whether small polarons form in CuSbSe₂. **a**, Addition of an unpaired hole followed by distortion around the Cu, Se1 and Se2 sites and/or rattling of the lattice results in a relatively small change in DFT total energy of about ~12 meV compared to the unperturbed state. Cu and the two Se sites were selected, since the valence band is dominated by these states (refer to Fig. 2b in the main text). The partial electron density function of the unpaired hole for the **b**, unperturbed and the **c**, perturbed structures show a quasi-2D valence band as the electron density is confined to a single layer. **d**, Addition of an unpaired electron followed by distortion around the Sb site results in the formation of a metastable state 0.12 eV above the unperturbed structure. Analysis was performed on distortions around Sb because Sb 5p orbitals dominate the lower conduction band (refer to Fig. 2b in the main text). The partial electron density of the unpaired electron

in the **e**, unperturbed structure shows a delocalised 3D solution, while the higher energy **f**, perturbed structure shows quasi-1D character because the electron density is confined to ribbons running through the interlayer space. In this analysis, we considered a supercell that was four times the volume of a unit cell, and we did not observe any localization of either the electron or hole charge density (0D character) to within one unit cell. That is, these computations support conclusions that small polarons do not form in CuSbSe₂

The details of how these calculations were performed are included in the Methods section of the main text.”

The most appealing results of this manuscript are indeed the measurements (Raman spectra, X-ray diffraction, and mobilities) and the solution processing method which all seem to be of high quality. However, the manuscript does not meet Nature Communications' criteria as the authors' claims on the existence of large polarons, the theoretical approach, and the novelty in advancing a particular field are not convincing.

Response 3.2: We are glad that the reviewer is satisfied with the quality of our work. We hope that our detailed responses, and associated changes made to the paper, as outlined in Response 3.1 above, address the reviewer's concerns on the theoretical approach we have taken. We also hope that the new calculations we have undertaken, detailed above in Response 3.1, address the reviewer's point about directly showing the absence of small polarons explicitly.

Regarding novelty, the core conceptual advance of the work is the discovery of the physical and chemical factors that enable pnictogen-based perovskite-inspired materials to avoid carrier localization, as the reviewer recognised in the opening of their comments. There has been a significant global effort to develop lower-toxicity alternatives to lead-halide perovskites, which has given rise to substantial work on Bi- and Sb-based materials. However, as we explained in the introduction, a significant drawback of Bi-based materials is the presence of carrier localization, and carrier localization has also been found to be a limiting factor in Sb-based materials:

“Carrier localization substantially reduces mobilities and therefore limits diffusion lengths ... Recent investigations into the wider family of bismuth-halide and bismuth-chalcogenide semiconductors have found carrier localization to be so prevalent that it is being described as a hallmark of these materials^{10-12,14,28-31}. The effect of carrier localization on Sb-based compounds is not as well established. One of the best-studied of these materials is the antimony chalcogenide family of compounds (Sb₂S₃ and Sb₂Se₃). There are currently strong disagreements in the community regarding whether self-trapping occurs in these materials, limiting open-circuit voltages up to a maximum of 0.8 V^{30,32-34}, or whether the performance is instead limited by charged defects³⁵⁻³⁷. In Cs₂AgSbBr₆, on the other hand, charge-carrier localization proceeds on a picosecond timescale, similar to that in Cs₂AgBiBr₆, with alloying of the two materials exacerbating such effects, owing to localized charge-carriers being more susceptible to energetic disorder³⁸ ... It is clear that the future development of pnictogen-based perovskite-inspired materials for optoelectronic devices urgently requires not only consideration of defects, but also insights into how charge-carrier localization may be avoided in these materials.”

Therefore, understanding how we could design perovskite-inspired materials to avoid carrier localisation is of paramount importance to push forward this broad and interdisciplinary field to discover and develop more efficient solar absorbers.

Very recently, we uncovered hints in this direction with the discovery that BiOI is a perovskite-inspired material that exhibits band-like transport^{37,38}. However, the reasons behind this were unclear, although we hypothesized that it could be related to BiOI being comprised of thick layers^{37,38}. This motivated us to investigate CuSbSe₂, another layered compound. Whilst there have been investigations into the defect tolerance of CuSbSe₂, whether or not this material undergoes carrier localization was unknown.

In the results section of the paper, we show that there is no carrier localization in CuSbSe₂ because 1) the photoconductivity transient, measured by OPTP, decays too slowly (with 92% of the initial signal decaying over 50 ps, whereas materials with localization would decay within 1 ps), and 2) the mobility decreases with increases in temperature, consistent with phonon-scattering limited transport. Self-trapped carriers would instead have their mobility increase with temperature as hopping is thermally activated. Also in the results section, we show computationally that both the Fröhlich and acoustic coupling are weak (Table 1). The latter is due to the low deformation potentials, which g_{ac} strongly depends on (refer to Eq. 1). These results are supported by the new calculations we made (detailed in Response 3.1).

Critically, in this work, we push forward the wider field by uncovering the reasons why CuSbSe₂ avoids carrier localization, which could be generalized to other perovskite-inspired materials. This is covered in detail in the discussion section. To summarize, we discovered that the following factors are conducive to materials with delocalized charge-carriers:

- Gaps being present in the crystal structure, into which distortions to the lattice due to the population of phonon modes could be relaxed with minimal changes to bond lengths, minimizing deformation potentials
- Electronic coupling between atoms across the interlayer gaps, such that the electronic dimensionality at band-edges is 2D or higher, which reduces the likelihood of the system being thermodynamically favoured to relax into a self-trapped state, especially when combined with the low acoustic coupling constant
- Low ionic dielectric contribution compared to the electronic contribution by having a small bandgap and species without Born effective charges substantially above the formal charges of the elements

These findings open up the future design of perovskite-inspired materials with band-like transport, and the next step would be to test these design principles across a wide range of materials both computationally and experimentally.

We captured this point on page 10 of the main text:

“The understanding gained from investigating the case of CuSbSe₂ can provide insights into how we could design heavy pnictogen-based semiconductors with band-like transport, which will be critical for creating more promising earth-abundant solar absorbers.”

To make these points clearer, we replaced the overarching sub-heading “*mechanisms for weak charge-carrier-phonon coupling in CuSbSe₂*” with three sub-sections in the discussion to more clearly to guide readers through our points made. The sub-section headings added are:

- “*Understanding the cause of low deformation potentials in CuSbSe₂*”
- “*High electronic dimensionality in CuSbSe₂*”
- “*Understanding weak Fröhlich coupling in CuSbSe₂*”

In order to better set out the key hypotheses that we rigorously test in the discussion section, we changed the opening of the discussion on page 27 to:

“In light of the results presented, it can be seen that carrier localization is not present in CuSbSe₂, and that this is due to the weak coupling between charge-carriers and both acoustic and optical phonons. This is unusual among heavy pnictogen-based perovskite-inspired materials, and it is critical to unravel the underlying factors. In this discussion, we will examine computationally how the structure and bonding in CuSbSe₂ result in the low deformation potentials and low Fröhlich coupling constants, as well as the effect on the electronic dimensionality in both the valence and conduction band.”

Our conclusions already capture the three key points made above:

“Based on our investigations, we propose that the free volumes (e.g. interlayer gaps) in the lattice can help to minimize the effect of lattice distortions on the bonding environment and lower the deformation potential. At the same time, electronic coupling across the interlayer gap between species contributing to the band-edge density of states can increase the electronic dimensionality, which reduces the likelihood of self-trapping. Finally, materials with low ionic contributions to the dielectric constant are desired to minimize Fröhlich coupling, but this needs to be balanced with the effect on the capture cross-section of charged defects. These insights are valuable for the future design of solar absorbers that have band-like transport.”

Finally, we mention that another novelty in this work is that it is the first report of the synthesis of CuSbSe₂ by a thiol-amine process. As a solution processing method, this is potentially more cost-effective than traditional vacuum-based approaches, or techniques requiring long post-treatment times. At the same time, the alkali solvent is less dangerous than hydrazine, which was previously used. To make this clearer, we modified page 12 of the main text:

“In this work, we develop the synthesis of phase-pure CuSbSe₂ thin films by this novel thiol-amine-based solution processing route for the first time, as detailed in Methods.”

Some other comments, without a specific order, include:

It is not surprising that acoustic phonons lead to weak electron-phonon coupling effects which is the case for halide perovskites.

Response 3.3: We thank the reviewer for this comment, but the link between carrier localization in pnictogen-based semiconductors and lead-halide perovskites is not straightforward. Indeed, if we compare CuSbSe₂ with (CH₃NH₃)PbI₃, these two compounds have very different crystal structures and very different phonon dispersion curves. Thus, we would not suspect the electron-phonon coupling behaviours to be similar. Even if we compare (CH₃NH₃)PbI₃ with Cs₂AgBiBr₆ double perovskite, which has a much more similar (but not the same) crystal structure, the carrier localization behaviours are still very different. Cs₂AgBiBr₆ exhibits self-trapping due to strong coupling to acoustic phonons, which arises due to its high deformation potentials³⁹. Compounding these difficulties, the frontier orbitals of Ag and Bi in Cs₂AgBiBr₆ are not at similar energies, and therefore the electronic dimensionality is well below 3D, likely close to 0D.

More broadly, if we look at recent literature on bismuth- and antimony-based perovskite-inspired materials, carrier localization has been found to be widely prevalent. We explained this in the introduction:

“Recent investigations into the wider family of bismuth-halide and bismuth-chalcogenide semiconductors have found carrier localization to be so prevalent that it is being described as a hallmark of these materials^{10-12,14,28-31}. The effect of carrier localization on Sb-based compounds is not as well established. One of the best-studied of these materials is the antimony chalcogenide family of compounds (Sb_2S_3 and Sb_2Se_3). There are currently strong disagreements in the community regarding whether self-trapping occurs in these materials, limiting open-circuit voltages up to a maximum of 0.8 V^{30,32-34}, or whether the performance is instead limited by charged defects³⁵⁻³⁷. In $Cs_2AgSbBr_6$, on the other hand, charge-carrier localization proceeds on a picosecond timescale, similar to that in $Cs_2AgBiBr_6$, with alloying of the two materials exacerbating such effects, owing to localized charge-carriers being more susceptible to energetic disorder³⁸.”

Therefore, the discovery of delocalized charge-carriers and weak coupling to acoustic phonons in $CuSbSe_2$ is surprising. Furthermore, we make the surprising finding that the Fröhlich coupling in $CuSbSe_2$ is weaker than in lead-halide perovskites, despite both being comprised of polar-covalent bonds. Understanding of the underlying reasons why, which we put forward in our work, are important for the field.

Could the authors provide more clarity regarding their confidence in assigning B_{2u} and B_{3u} modes in the FTIR measurement? Was this determination based on energy matching? It would enhance the understanding if the authors could specify their level of confidence, perhaps using terms like "tentatively" to convey the certainty of the assignment.

Response 3.4: FTIR modes were assigned by energy matching to the calculated Γ -point modes. Assignment of the B_{2u} mode is tentative because an IR-active B_{3u} mode of similar energy is also predicted. Assignment of the high frequency B_{3u} mode is much more confident, as the next nearest IR mode is 5 cm^{-1} away. We have added the following to the caption of Supplementary Fig. 2:

“The symmetries of the phonon modes were obtained by energy matching with the calculated phonon dispersion curve. The assignment of the B_{2u} mode is tentative, since a B_{3u} mode of similar energy is present. But the higher-energy B_{3u} mode is assigned with greater confidence, since the nearest IR-active mode is 5 cm^{-1} away.”

Supl. Fig.3: I think combining figures (a) and (b) can facilitate direct comparison.

Response 3.5: We thank the reviewer for the suggestion to combine together the experimentally-measured absorption coefficients and the calculated absorption coefficients. However, looking carefully at the experimental data, we can see that there is sub-bandgap absorption (Supplementary Fig. 6b in the revised SI), which is difficult to investigate by calculations. Furthermore, the exact values of the absorption coefficient between experiment and computations are different. Looking in detail at our computational analysis, we found that the absorption spectra for $CuSbSe_2$, determined from the imaginary component of the dielectric constant, varies based on the principal axis, and we have a polycrystalline film with texturing. Added to this, there is light scattering from the textured morphology of the $CuSbSe_2$ films. Overall, we found that we could not make a 1:1 direct comparison between the measured and calculated absorption spectra. Our main purpose in calculating the absorption spectra was to determine whether the shoulder in the measured absorption spectrum at 1.3 eV was due to excitonic effects or not. Our calculations, which accounted for no excitonic

effects, showed that this shoulder does indeed take place due to direct band-to-band transitions, as stated in the main text and in Supplementary Note 4:

“As discussed in the main text, both spectra [from experiment and computations] exhibit a similar shoulder near the absorption onset, which is consistent with the shoulder originating from the electronic structure of CuSbSe₂.”

Overall, we believe it best to keep the measured and calculated absorption spectra separate.

The figure descriptions in the main text are out of sequence. For example, the reference to Fig. 2b appears after the discussion of Fig. 3.

Response 3.6: We thank the reviewer for pointing this out. We have carefully gone back over the revised paper and ensured that all figures are introduced in sequence. In the specific case raised by the reviewer as an example, we have made the following change on page 14 of the main text:

“Having developed phase-pure samples and understood the dominant phonon modes in CuSbSe₂, we next needed to understand the nature of excitations and their kinetics. The black solid line in Fig. 2a shows the measured optical absorbance curve of CuSbSe₂, and the electronic structure is shown in Fig. 2b. The fit to the optical absorption spectrum (red dashed line in Fig. 2a) was obtained from Elliott’s theory⁹³ ...”

It appears that there might be confusion on the part of the authors regarding the definition of polarons. At variance with phonons (which represent vibrations around equilibrium positions), a polaron causes lattice distortion. However, polaron effects are not captured in their theoretical calculations. What is calculated here is the phonon-limited mobility due to LO phonons and not the polaron-limited mobility. The interpretation that the decrease in mobility implies the presence of large polarons is not necessarily correct.

The authors could comment on why the calculated mobility is overestimated with respect to the experiment.

Response 3.7: We combined these two sets of comments from the reviewer, because they relate to the same topic. In both cases, we believe the reviewer is referring to Supplementary Fig. 5 that was in the original submission (copied below). Indeed, polarons and phonons are different. We have provided clarification our theoretical approach (to answer the first part of the question) in Response 3.12.

The overestimation of the calculated mobilities to the measured mobilities implies that neither Polar Optical Phonon (POP) nor Acoustic Deformation Potential (ADP) scattering are limiting in this system. Other effects such as optical deformation potential scattering, grain boundary scattering, or defect scattering could all play a part in further reducing the calculated mobility closer to the reported experimental value.

Supplementary Fig. 5 | Computational analysis of mobility. Calculated temperature-dependent total mobility (black spots) and mobilities when acoustic deformation potential (ADP, red upward triangles) scattering or polar optical phonon (POP, pink downward triangles) scattering dominates. The mobility values limited by ionized impurity scattering (IMP) are too high to be included in the figure.

While we did not use the term ‘polaron mobility’ in the original submitted manuscript, we acknowledge that the discussion of the temperature dependence of the computational and experimental mobilities in tandem has led to some confusion. To clarify, the useful conclusion the AMSET model allows us to draw is that POP (Fröhlich-like) scattering dominates acoustic deformation potential (ADP) scattering, and is unrelated to its temperature dependence. Unlike the experimental Hall mobilities, the AMSET model cannot be used to infer the presence of small polarons based its temperature dependence, as it is a scattering model and assumes band-like transport *a priori*.

After internal discussion, we have decided to remove this figure from the Supplementary Information, along with associated references to this figure in the paper. The reason that this data was included in the first instance is mostly historical. Many of the calculated quantities presented in this work, such as the dielectric constants, Γ -point modes, deformation potentials, and band structure, were originally calculated solely as inputs to the AMSET model. However, as we gathered more data the scope of the paper shifted further from a basic presentation of calculated/measured materials properties, and closer towards what we believe is a more interesting discussion on the strength and fundamental origins of the electron-phonon interaction in CuSbSe₂. In this sense, while the calculated mobility data originally motivated computational work, it now falls out of the scope of the useful discussion in the paper’s final form. We believe its removal is justified, as these data do not contradict the conclusions we present, however their inclusion distracts from the takeaway message of the work and has led to confusion amongst reviewers.

We have double checked through the paper to make sure that we do not make any references to phonon-limited mobilities.

Finally, we would like to discuss also the statement from the reviewer that “*The interpretation that the decrease in mobility implies the presence of large polarons is not necessarily correct.*”

Transport measurements are widely considered a key tool in distinguishing between large and small polarons in a material⁴⁴. The hallmark of small polaronic systems are very small carrier mobilities (typically $\ll 1 \text{ cm}^2\text{V}^{-1}\text{s}^{-1}$) which increase with temperature (*i.e.*, are thermally activated). Conversely, large polarons behave like ‘dressed’ carriers in a band-like system

limited by phonon scattering and typically have larger mobilities which smoothly decrease with temperature due to increased phonon scattering. As the reviewer noted, there are situations where temperature-dependent mobility data can fall outside of these two well-behaved monotonic cases, which would make interpretation more difficult. However, the Hall effect measurement data presented in our manuscript does not fall within this category, and the conclusions we draw from it are consistent with those from time-resolved spectroscopy measurements, where we show a slow decay in the photoconductivity from OPTP measurements, consistent with the absence of carrier localisation. In the absence of contradictory data, we conclude small polarons are unlikely to form. Accepting that polarons are ubiquitous in polar materials, the absence of small polarons implies the presence instead of large polarons (*i.e.*, those not strongly bound to a lattice site).

As mentioned in Response 3.1, we have also carried out additional calculations to verify our claims that small polarons do not form, and these further details have been added to the paper.

AMSET is not defined either in the main manuscript or the supplemental material.

Response 3.8: We apologise for this oversight. AMSET is the *ab-initio* scattering and transport model. We have now defined this in the first instance it is used in the main text:

“Deformation potential calculations were calculated using the method of Wei and Zunger¹²⁷⁻¹² with deformed structures generated and analyzed via the ab-initio Scattering and Transport (AMSET) package¹⁰⁰.”

Surprisingly the band gap of the material is not reported.

Response 3.9: Attempts have been made by several groups to determine the bandgap of CuSbSe₂ through experimental and computational approaches. Experimental approaches made use of a Tauc plot, formed from UV-visible spectroscopy measurements of the absorption coefficient spectrum. An example of such a Tauc plot is shown in Fig. R3 below. The bandgaps obtained from this approach range from 1.04 eV to 1.2 eV^{8,18-20}. However, it should be noted that all of these works that determined the bandgap from a Tauc plot assumed a direct bandgap ($(\alpha hv)^2$ vs hv), whereas we can see from the calculated electronic structure of CuSbSe₂ that it is an indirect bandgap semiconductor (refer to Fig. 2b in the main text). Furthermore, in these prior works, there is significant tailing of the absorption spectrum of CuSbSe₂ below the derived bandgap, which can be seen in the example shown in Fig. R3. This leads to the question of whether the actual bandgap of CuSbSe₂ is smaller than experimentally determined.

Figure R3. Reported optical absorption coefficient of CuSbSe₂ film. Inset: Tauc plot ($n = 2$, direct) for CuSbSe₂ film ($E_g = 1.04$ eV). Reprinted with permission from *Adv. Energy Mater.* 2015, 5, 1501203. Copyright 2015 WILEY-VCH Verlag GmbH & Co. KGaA, Weinheim.

Computationally, Yu *et al.* reported the indirect bandgap of CuSbSe₂ to be 1.17 eV, with a first direct transition of 1.26 eV⁴⁵. Maeda *et al.*⁴⁶ reported a smaller indirect bandgap of 0.93 eV, with a first direct transition of 1.04 eV. In our calculations of the band structure, shown in Fig. 2b, we computed an indirect bandgap of 1.16 eV.

Thus, we can say that the bandgap of CuSbSe₂ is small, most likely below the 1.3 eV bandgap that is optimal for solar absorbers under 1-sun illumination. However, it is difficult to precisely determine the bandgap, and we suspect that many of the experimentally-reported values in the prior literature are incorrect because 1) of the assumption of a direct bandgap in constructing the Tauc plot, and 2) the significant tailing in the absorption of CuSbSe₂ was ignored. We similarly see substantial tailing in our measured absorption spectra (refer to Supplementary Fig. 6). It is therefore called for to have a separate work that goes into detail into the absorption spectrum of CuSbSe₂ at the band-edge to precisely determine the bandgap of this material. However, this is beyond the scope of the current work, which is focussed on charge-carrier transport.

At the same time, we agree with the reviewer that it would be helpful to give an indication of what the bandgap of this material is. We have therefore added this information into the introduction when we first introduce this material:

*“Inspired by this recent work, herein we investigate a related layered Sb-based compound, CuSbSe₂. This material is a *přibramite*, which is the Se analogue to the chalcostibite CuSbS₂, and has experimentally- and computationally-determined bandgaps in the range of 0.9–1.2 eV⁴¹⁻⁴⁶. This is smaller than the bandgaps found for most Sb- and Bi-based perovskite-inspired materials recently investigated (Supplementary Table 1), and is suitable for harvesting the near-infrared portion of the solar spectrum, which is a substantial fraction of the energy in the AM 1.5G spectrum⁴⁷.”*

How the average in Table 1 is taken? Some values do not represent the average, e.g. the binding energy.

Response 3.10: In short, this is due to the fact that we are dealing with tensor valued quantities, some of which are different ranks (e.g., deformation potentials (E_{nk}) and elastic constants (c_{ijkl}) are rank 2 and 3 tensors, respectively). To arrive at the averages shown in Table 1, a tensor averaging scheme was employed for each tensor, and these values were used in the relevant formula. For rank 2 tensors, the tensor was diagonalized and the average of their eigenvalues taken. There are a number of common schemes used to average the tensor of elastic constants (e.g. Reuss, Hill, and Voigt averages). For this work, all of these methods produced similar values. We used the Hill average, calculated via the *Elate*⁴⁷ web app in our paper, which is itself the mean of Reuss and Voigt averages. Simple means were used for rank 1 tensors, except the effective masses for which the harmonic mean was taken. A Jupyter notebook is now available in the raw data file which explicitly shows all data processing steps.

We have added a footnote to Table 1 to explain this, and a Supplementary note:

Table 1:

^a *For details on how averaging for each quantity was carried out, see Supplementary Note 8.*

“Supplementary Note 8 / Tensor averaging

*Finding the average value for some quantities in Table 1 requires combining tensors of different ranks into a single value, e.g., deformation potentials (E_{nk}) and elastic constants (c_{ijkl}) are rank 2 and rank 3 tensors respectively. To arrive at the average shown in Table 1, a tensor averaging scheme was employed for each tensor, and these values were used in the relevant formula. For rank 2 tensors, the tensor was diagonalized and the average of their eigenvalues taken. There are a number of common schemes used to average the tensor of elastic constants (e.g., Reuss, Hill, and Voigt averages). For this work, all schemes produced similar values. We used the Reuss average, calculated via the *Elate*⁵¹ web app. Simple means were used for rank 1 tensors, except for the effective masses, for which the harmonic mean was taken. Once a tensor average for each quantity in each formula was found, these were used in the relevant equations to find the average values quoted in Table 1. A Jupyter notebook is available in the raw data file which explicitly shows all data processing steps.”*

I am uncertain about the authors' decision to relax the structure after applying strain. Their objective is to compute the acoustic deformation potential, and its derivatives are typically taken with respect to the structure at equilibrium.

Response 3.11: We are grateful to the reviewer for the opportunity to clarify this important point, which we carefully considered as part of this work. Indeed, the reviewer is correct that, in the literature, it is common for deformation potentials to be calculated based on the structure in the ground state. Subsequent calculations are then normally performed with a change in the unit cell volume, but without a change in the relative coordinates of the atomic basis. In this case, a deformed, but unequilibrated structure is used, since the atomic coordinates would not necessarily then give the lowest energy configuration in the deformed structure.

This method of calculating absolute deformation potential was popularised by Franceschetti, Wei and Zunger (FWZ) the mid 1990s to mid 2000s while investigating highly symmetric

diamond and zinc blende semiconductor materials^{11,48,49}. However, due to the high symmetry of the zinc blende structure, as well as the stiff bonds present in compounds in this family of materials, it is unlikely that the relative positions of the atoms within the unit cell would have changed significantly, even if the atomic basis were allowed to relax following deformation. Thus, we can make the reasonable assumption that the unequilibrated structure serves as a good approximation of the actual equilibrated structure.

In our work, we used the FWZ method for our high level HSE06 calculations of the deformation potential in Table 1 of the main text. However, we wished to test whether the assumption made in the FWZ approach holds for CuSbSe₂, which is part of a significantly different materials class than the zinc blende compounds. To do this, we performed additional deformation potential calculations at the r²SCAN level, as shown in Supplementary Fig. 11. We chose to use the r²SCAN functional here, as running multiple relaxations at HSE06 level was not feasible, and we only required a qualitative result. Through these calculations, we showed that using the unequilibrated structure leads to a substantial error for CuSbSe₂, for example equilibration after distortion along the c-axis results in a reduction of -59% (refer to Supplementary Table 4). This emphasises that using the unequilibrated structure in materials that are low symmetry and have deformable structures can lead to significant error. We would note that while the errors may be large, in the case of CuSbSe₂ they result in overall weaker coupling to acoustic phonons, and thus the decision to not relax and recalculate at HSE06 level for Table 1 does not change our conclusions that acoustic coupling is weak.

We note that the results of this analysis in Supplementary Fig. 11 were not fed into other sections presented in this manuscript, such as the Toyozawa and AMSET models, but serve more to highlight the complexity of semiconductor materials in literature today, and to suggest a benchmarking study of deformation potentials comparing equilibrated and unequilibrated structures may be justified using a qualitative functional (e.g., PBEsol).

To capture these points, we have modified the main text on page 23 of the main text:

“Please refer to Supplementary Note 7 for more details on how we calculated the acoustic deformation potential, especially Supplementary Fig. 10 and the associated discussion beneath it. The values in Table 1 are calculated self-consistently with a fixed internal structure using the HSE06 exchange-correlation functional.”

We have also provided a clarification to Figure 4:

“Fig. 4 | Computational analysis of CuSbSe₂ ... b, Percentage changes in bond lengths and interlayer distance of CuSbSe₂ as a function of strain along the c-axis. All calculated bond lengths shown are after relaxation of the atoms in the structure after distortion, i.e., calculations for equilibrated structures as shown (refer to discussion in Supplementary Note 7).”

Further details are in the Supplementary Information:

“Deformation potentials can be measured but are more readily calculated via ab-initio methods. The method of Wei and Zunger is explained in Supplementary Fig. 10. In this method, a fixed internal structure is maintained, which means in practice that the change in relative atomic coordinates of each species is not allowed in response to the deformation of the unit cell. That is, an unequilibrated structure is compared to the ground-state undeformed structure. Here we define an unequilibrated structure as one to which a homogenous strain has been applied, but the internal atomic coordinates have not subsequently been allowed to relax to a local minimum.”

The use of unequilibrated structures was reasonable in the types of materials originally investigated by Wei and Zunger, which were high-symmetry zinc blende semiconductors with no internal structural

degrees of freedom. However, this assumption may not be valid in the case of CuSbSe₂. We therefore performed additional deformation potential calculations at the *r*²SCAN level. In the case of CuSbSe₂, we found that allowing atomic relaxations following the deformation of the unit cell results in a mean reduction in the predicted deformation potentials by -25 % and -12.4% for CBM and VBM, respectively, as compared to the unequilibrated case (Supplementary Table 4). To probe the cause of this reduction, we compared cation-anion bond lengths before and after atomic relaxation. We found that when the unit cell was strained along the *c*-axis and allowed to relax, there were comparatively little changes in the intralayer bond lengths because these distortions were mostly taken up by an increase in interlayer spacing (Fig. 4b). Referring to Figs. 2b and 4c in the main text, we see that the band edges are dominated by intralayer bonding, and so relaxing these bonds towards their ground state configurations should minimize changes in the electronic structure, thus minimizing the deformation potentials.”

The method for calculating the prediction of the Acoustic Deformation Potential (ADP) and Polar Optical Phonon (POP) contributions in Supplemental Fig. 5 is unclear.

Response 3.12: As explained in Response 3.7, we have removed these ADP and POP calculations in the original Supplementary Fig. 5.

But for the benefit of the reviewer, we will elaborate here on the formalism used to calculate POP and ADP scattering components regardless. Mobilities were calculated using the *ab initio* Scattering and Transport package (AMSET)¹² which itself relies on Fermi’s golden rules of elastic/inelastic scattering, and the Boltzmann transport equation in the Momentum Relaxation Time Approximations (for elastic processes such as ADP scattering), or the Self-Energy Relaxation Time Approximation (inelastic processes such as POP scattering). We present Fermi’s golden rules for elastic scattering below:

$$\tau_{nk \rightarrow mk+q}^{-1} = \frac{2\pi}{h} |g_{nm}(\mathbf{k}, \mathbf{q})|^2 \delta(\Delta\epsilon_{k,q}^{nm})$$

Where $g_{nm}(\mathbf{k}, \mathbf{q}) = \langle m\mathbf{k} + \mathbf{q} | \Delta_q V | n\mathbf{k} \rangle$ is the scattering element between electronic state $\langle m\mathbf{k} + \mathbf{q} |$ and $\langle n\mathbf{k} + \mathbf{q} |$. These states are calculated via hybrid-DFT using an extremely dense *k*-point mesh, which is further interpolated onto a denser mesh via a linear tetrahedron scheme to improve resolution of the base calculation. The linear band structure presented in the Main Text is interpolated from this dense uniform base calculation. $\Delta_q V$ represents a general scattering potential. The full scattering element for POP and ADP scattering is given by the following equations respectively:

$$g_{nm}^{POP}(\mathbf{k}, \mathbf{q}) = \left[\frac{\hbar\omega_{po}}{2} \right]^{\frac{1}{2}} \sum_{\mathbf{G} \neq -\mathbf{q}} \left(\frac{1}{\hat{\mathbf{n}} \cdot \boldsymbol{\epsilon}_{\infty}} - \frac{1}{\hat{\mathbf{n}} \cdot \boldsymbol{\epsilon}_s \cdot \hat{\mathbf{n}}} \right)^{\frac{1}{2}} \times \frac{\langle m\mathbf{k} + \mathbf{q} | e^{i(\mathbf{q}+\mathbf{G}) \cdot \mathbf{r}} | n\mathbf{k} \rangle}{|\mathbf{q} + \mathbf{G}|}$$

$$g_{nm}^{ADP}(\mathbf{k}, \mathbf{q}) = \sqrt{k_{BT}} \sum_{\mathbf{G} \neq -\mathbf{q}} \left(\frac{E_{nk} \cdot \hat{\mathbf{S}}_l}{c_l \sqrt{\rho}} + \frac{E_{nk} \cdot \hat{\mathbf{S}}_{t_1}}{c_{t_1} \sqrt{\rho}} + \frac{E_{nk} \cdot \hat{\mathbf{S}}_{t_2}}{c_{t_2} \sqrt{\rho}} \right) \langle m\mathbf{k} + \mathbf{q} | e^{i(\mathbf{q}+\mathbf{G}) \cdot \mathbf{r}} | n\mathbf{k} \rangle$$

Inputs to these scattering elements, such as ω_{po} , and the elastic tensor c were calculated using DFPT, and finite differences respectively. Calculation of deformation potential tensor (E_{nk}) has been carried out via the method of Wei and Zunger and set up using helper tools within the AMSET package.

For a more detailed description, we refer the reviewer to the original AMSET publication¹², and in the particular the SI of this publication, where a detailed mathematical framework for the model is presented.

It would be nice for the authors to comment that the temperature dependence of the effective masses and phonon frequencies (anharmonicity, renormalization of the phonon self-energy) are not taken into account. In general, it will be beneficial for the manuscript to state all the approximations involved in computing mobilities, deformation potentials, and Fröhlich coupling.

Response 3.13: We thank the reviewer for these excellent suggestions, and we have now accordingly added these statements in to the Methods:

“Calculations of the crystal, electronic, phonon structure, and bulk-polaron partial charge density functions were carried out in the Kohn Sham Density Functional Theory (KS-DFT) framework¹⁴ using with the projector augmented wave (PAW) method¹⁵ as implemented using the Vienna ab-initio software package (VASP)¹⁶. The PBE.54 PAW potential set was used throughout (Cu 22Jun2005, Sb 06Sep2000, Se 06Sep2000). ... Band structure between high symmetry points was interpolated from a densely sampled uniform band structure calculation generated using the zero-weighted k-point method atop a weighted 4×6×2 grid. The dielectric function was calculated in the single particle approximation using the linear optics routine of Gajdos et al. implemented in VASP^{115,117-120} (LOPTICS=.TRUE.)... Deformation potential calculations were calculated using the method of Wei and Zunger¹²⁷⁻¹²⁹, with deformed structures generated and analyzed via the ab-initio Scattering and Transport (AMSET) package¹⁰⁰. The phonon band structure was calculated using the finite displacement method¹³⁰ within the harmonic approximation as implemented in Phonopy using a displacement of 0.15 Å^{131,132}, using the regularized and restored SCAN (r²-SCAN) meta-GGA functional¹³³. It should be noted that within the harmonic approximation, finite temperature effects are neglected...

The search for low energy bulk polarons was carried out using the ShakeNBreak method and package¹⁰¹. A 64-atom supercell was used to perform a spin-polarized calculation of (ISPIN=2) containing an unpaired electron or hole, which was enforced using NELECT and choosing NUPDOWN=1 INCAR tags at HSE06 level. Local distortions around atomic sites of ±30% and 0% were applied, followed by a stochastic rattle of all atoms in the cell with a standard deviation of 0.25 Å. An unperturbed supercell was also run. For holes, distortions centered on Cu, Se1, and Se2 were trialed, while for electrons distortions centered on Sb were trialed. Distorted supercells were relaxed within the Γ -point approximation using the Γ only version of VASP. Energy data was plotted using the tools within ShakeNBreak while partial charge densities of bands containing unpaired hole and electron were generated using pymatgen¹³⁶ and plotted using VESTA.”

A discussion of how the mobility, effective masses, band gaps, dielectric constants, and effective charges are compared to other Sb- and Bi-based compounds could be beneficial to strengthen the points of the manuscript.

Response 3.14: This is an excellent suggestion, and we have accordingly added a table to the Supplementary Information comparing these parameters for CuSbSe₂ with other Sb- and Bi-based compounds (copied below for convenience).

Mobility and effective mass: we have added the following paragraph to page 21 of the main text:

“Importantly, the mobilities measured in these CuSbSe₂ films are higher than typically found for small polarons. Hall effect measurements gave a macroscopic mobility of $1.01 \pm 0.01 \text{ cm}^2 \cdot \text{V}^{-1} \cdot \text{s}^{-1}$ at room temperature (Fig. 3b, 300 K), while the delocalized intra-grain mobility value extracted from the initial photoconductivity of OPTP measurements was $4.7 \pm 0.2 \text{ cm}^2 \cdot \text{V}^{-1} \cdot \text{s}^{-1}$ (refer to Supplementary Note 5 for details of how the mobility was determined from OPTP measurements). This difference in mobility values is due to the different length scales of Hall effect and OPTP measurements. Hall effect measurements investigate charge carrier transport throughout the whole sample, while the mobility extracted from the initial OPTP signal represents the transport within a shorter range, usually well within one grain^{25,27,28,48}.

Even though the mobilities values extracted over different length scales are different, comparing mobility values obtained from different materials over the same length scale is informative to put the nature of charge-carrier transport in context. BiOI, as another pnictogen-based compound which can avoid charge-carrier localization, has a peak OPTP mobility of $\sim 3 \text{ cm}^2 \cdot \text{V}^{-1} \cdot \text{s}^{-1}$ at 295 K for polycrystalline samples. In single crystals, the mobilities for BiOI obtained from time-of-flight measurements are $26 \text{ cm}^2 \cdot \text{V}^{-1} \cdot \text{s}^{-1}$ (out-of-plane) and $83 \text{ cm}^2 \cdot \text{V}^{-1} \cdot \text{s}^{-1}$ (in-plane)³⁹. By contrast, materials undergoing carrier localization (e.g., NaBiS₂, Cs₂AgSbBr₆, Cs₂AgBiBr₆ and AgBiS₂ without heat treatment) exhibit a substantial reduction in mobility from an initial delocalized state to localized mobilities in the range of $0.03\text{--}1.3 \text{ cm}^2 \cdot \text{V}^{-1} \cdot \text{s}^{-1}$ 2 ps after excitation, as obtained from OPTP measurements (Supplementary Table 1). These OPTP mobilities are all lower than that for CuSbSe₂, despite CuSbSe₂ having higher effective masses (Supplementary Table 1). This is consistent with the delocalized nature of charge-carriers in CuSbSe₂, and emphasizes the importance of avoiding carrier localization.”

Supplementary Table 1 | Comparison of key properties of CuSbSe₂ with other Sb- and Bi-based compounds. These are the charge-carrier mobility at room temperature, effective mass, bandgap and dielectric constants, along with the time taken for the photoconductivity signals to decay by 50% from the initial peak value. The charge-carrier mobilities shown are extracted from the photoconductivity spectra measured by OPTP.

Material	OPTP Mobility ($\text{cm}^2 \cdot \text{V}^{-1} \cdot \text{s}^{-1}$)	Effective mass (m_h^*/m_0 ; m_e^*/m_0)	Bandgap (eV)	Dielectric constant (ϵ_∞ ; ϵ_{stat})	Time taken to reach 50% photo-conductivity decay (ps)
CuSbSe ₂	4.7 ± 0.2	1.60; 0.43	0.9-1.2	11.3; 23.0	6.7
NaBiS ₂ ¹	0.29 (delocalized); 0.03 (localized)	1.04; 0.24	1.4	$\epsilon_\infty = 8.1$; $\epsilon_{stat} = 43.7$	~ 0.5
AgBiS ₂ ^{2,3}	As-prepared: 0.43 ± 0.05 (delocalized), 0.11 ± 0.05 (localized); annealed: 2.70 ± 0.10 (delocalized), 2.20 ± 0.10 (localized)	0.51; 0.24	1.45	19.43(x/y), 12.44(z); 115.61(x/y), 35.08(z)	<1 (as-prepared); 20-30 (heat treatment)
Cs ₂ AgSbBr ₆ ^{4,5}	0.5 (delocalized); 0.1 (localized)	0.234-0.969; 0.289-0.431	1.64	4.82; 13.69	1-2

$\text{Cs}_2\text{AgBiBr}_6^{6-10}$	3 (delocalized); 1.3 (localized)	0.14; 0.37	2-2.25	4.60; 12.76	1-2
BiOI^{11}	~ 3	0.26; 0.23	1.93	8.60; 43.33	200-300
$\text{Sb}_2\text{S}_3^{12-14}$	0.9 ± 0.1	0.64; 0.40	1.7-1.8	11.55(x), 10.97(y), 8.25(z); 98.94(x), 94.21(y), 13.14(z)	10-20
$\text{MA}_3\text{Bi}_2\text{I}_9^{15-17}$	Not reported	0.95; 0.54	2.1-2.2	5.43(x), 4.67 (z); 39.89(x), 9.62(z)	Not reported
$\text{Cs}_3\text{Bi}_2\text{I}_9^{18,19}$	Not reported	0.94 (perpendicular ar to (100)), 2.14 (perpendicular ar to (001)); 0.33 (perpendicular ar to (100)), 3.22 (perpendicular ar to (001))	1.95	5.38 (perpendicular ar to (100)), 4.09 (perpendicular ar to (001)); 9.90 (perpendicular ar to (100)), 7.85 (perpendicular ar to (001))	Not reported
BiSI^{20-23}	Not reported	0.95; 0.53	1.57	8.03; 37.81	Not reported
BiI_3^{24-27}	Not reported	2.01; 0.68	1.8	7.1 (in plane), 6.4 (out of plane); 54 (in plane), 8.6 (out of plane)	Not reported

Band gap: We modified the introduction as follows on page 7 of the main text (refer to Response 3.9 for more details):

“This material is a přibramite, which is the Se analogue to the chalcostibite CuSbS_2 , and has experimentally- and computationally-determined bandgaps in the range of 0.9–1.2 eV⁴¹⁻⁴⁶. This is smaller than the bandgaps found for most Sb- and Bi-based perovskite-inspired materials recently investigated (Supplementary Table 1), and is suitable for harvesting the near-infrared portion of the solar spectrum, which is a substantial fraction of the energy in the AM 1.5G spectrum⁴⁷.”

Dielectric constants and Born effective charges: We have added another table to the Supplementary Information comparing the Born effective charges of Bi- and Sb-based materials. We have also added the following paragraph, highlighted in yellow beneath Scheme 1 in the discussion section on page 35 of the main text:

“For materials with strong ionic-covalent bonding, the Born effective charges can be significantly larger than the formal oxidation states. The BECs for CuI are close to the oxidation state of the species, however, when considering the whole BEC tensor for SbI and SeI atoms, we observe BEC values higher than the formal oxidation states with displacements along the b direction, while net BECs (summing over columns) for displacements in the a and c directions are lower than the oxidation states. The low net dynamical charges of SbI and SeI for a and c displacements are reflected by their close ϵ_∞ and ϵ_{stat} values, and contribute to the low Fröhlich coupling constants in these directions (Table 1). The anomalously large contributions of the SbI and SeI atoms along the b-axis are of interest. These can

be explained by either a change in polarization of Sb1 and Se1 atoms upon displacement, or a direct transfer of charge between the two species, however it is difficult to say for sure without further investigation¹¹. The lone-pair on the Sb1 atom, and its origins in a symmetry breaking interaction between hybridized Sb-s, p orbitals and Se p orbitals can explain this. Changes in the symmetry (e.g. via sub-lattice displacement) of the Sb-Se coordination sphere will change how the lone pair is expressed, leading to strong deviations in BEC. Nevertheless, despite the larger ionic contributions to the dielectric constant along the b-axis, Fröhlich coupling constants remain <2 (Table 1).

To put CuSbSe₂ in context, we compare the BEC values with those of other Sb- or Bi-based compounds, as well as CH₃NH₃PbI₃. The stable 6s² lone pair of Pb²⁺ results in anomalously higher BEC values than the formal valence of Pb²⁺, hence leading to high ionic dielectric constants ($\epsilon_{\infty} = 6.1$; $\epsilon_{stat} = 25.7$ ¹¹²).

These high dielectric constants lead to stronger Fröhlich interactions ($\alpha = 2-3$ for methylammonium lead iodide perovskites²⁴) than CuSbSe₂ as discussed in Results. As Supplementary Table 5 shows, almost all pnictogen atoms in these compounds exhibit higher BEC values than their formal valences, despite the anisotropic values along certain directions. Compared to these compounds, the BEC values of Sb in CuSbSe₂ in the a and c directions are obviously lower than the formal oxidation state, resulting in the low ionic dielectric contribution, and thus weak Fröhlich interaction.”

Supplementary Table 5 | Born effective charge (BEC) of pnictogen atoms in Sb- and Bi-based compounds, along with the BEC value of Pb atom in CH₃NH₃PbI₃. The label a, b and c refer to principal crystallographic axes, while xx, yy and zz refer to diagonal components of the Born effective charge tensors.

Material	Born effective charge
CuSbSe ₂ (this work)	Sb: 1.03 (a), 5.65 (b), 1.44 (c)
Cs ₂ AgBiBr ₆ ⁴⁵	Bi: 4.79 (average)
BiOI ⁴⁶	Bi: 5.87 (a); 5.87 (b); 3.08 (c)
Sb ₂ S ₃ ⁴⁷	Sb1: 2.89 (xx); 5.62 (yy); 7.36 (zz); Sb2: 3.33 (xx); 7.25 (yy); 4.50 (zz)
Cs ₃ Bi ₂ I ₉ ⁴⁸	Bi: 3.9 (xx), 3.9 (zz)
BiSI ⁴⁹	Bi: 6.42 (a); 3.04 (b); 4.01 (c)
BiI ₃ ²⁷	Bi: 5.2 (a); 5.2 (b), 2.8 (c)
CH ₃ NH ₃ PbI ₃ ⁵⁰	Pb: 5.22 (a), 3.54 (b), 4.86 (c)

The use of three different levels of approximations (i) meta-GGA functional for force constants, (ii) hybrid functionals for electronic structure calculations, and (iii) PBE for DFPT is quite unusual and not justified. For example, why not use PBE for all calculations? Is there a particular reason?

Response 3.15: While it is preferable to minimise the mixing of different levels of theory where possible, in this case, it would not be appropriate carry out all calculations at a single level (i.e., only PBE, r²SCAN, or HSE06) for all of the computed properties/computational investigations. In each case, we chose a level of theory which balanced quantitative accuracy with computational cost. For example, it would have been inappropriate to report quantities

related to the electronic structure (e.g., DOS/BS) using GGA (PBE) alone. Conversely, while we might expect the force constants predicted by HSE06 to describe the system better than r^2 SCAN, the added cost would be extremely large and unjustified.

Specifically in reference to points (i-iii):

(i) A number of dispersion-corrected GGA and meta-GGA functionals were considered for computing the force constants/phonon dispersion. r^2 SCAN was eventually chosen due to a demonstrated ability to predict force constants and high-frequency eigenmodes at a reasonable cost¹⁵, and gave good agreement with experimental Raman and FTIR results (refer to Supplementary Fig. 2). Carrying out these supercell calculations at hybrid level would have added significant computational cost for limited benefit. PBE was the original choice of functional for this calculation, however when comparing results to Raman and FTIR data, we found it failed to describe Γ -point modes as accurately as r^2 SCAN, and so was not selected for the production run.

(ii) Local functionals, such as PBE are known to poorly describe electronic structure, while HSE06 has been found to describe key electronic structure descriptors, such as bandgap, much more accurately^{16,17}. While in other sections we used (meta-)GGA and still could achieve quantitative agreement with experiment, we deemed the error this introduces into bandgap-dependent properties, such as the high-frequency dielectric constant, to be too large. In such cases, we opted to use HSE06.

(iii) While, aesthetically, it would have been preferable to run DFPT calculations at either meta-GGA or hybrid level, this is not possible using the *ab-initio* calculator VASP, which is limited to LDA and GGA functionals using this formalism. A benchmarking study by Petousis *et al.* found GGA to model the dielectric constant with a mean absolute relative deviation of 16.2% as compared to an experimental dataset of 88 materials^{50,51}. GGA models the ionic contributions to the dielectric constant well, but the aforementioned error can be reduced further by calculating bandgap-dependent contributions to the dielectric constant at HSE06 level, a task GGA struggles with due to the well-known bandgap problem of DFT. This is why using a mixture of PBE and HSE06 derived quantities, as we have done on this work, is justified. It is not feasible or necessary to calculate the ionic contributions at hybrid-DFT level.

Thus, we could not use PBE for all calculations, since it poorly describes the electronic structure, and was not as accurate as r^2 SCAN for predicting Γ -point modes in the phonon dispersion curve, and was therefore not well suited to carrying out the force constant calculations.

To clarify the above points, we have added the following to the start of the theory section of Methods in order to explain our rationale for choosing the functionals used:

“For the computations made in this work, we carefully selected the functional to use that struck a balance between accuracy and computational cost. For example, we found that the regularized and restored SCAN (r^2 -SCAN) meta-GGA functional provided a much more accurate description of Γ -point modes, and was therefore more suitable for computing force constants and the phonon dispersion curve. PBE is also well known to poorly describe electronic structure, and we therefore used hybrid functionals (details below) for computing properties requiring an accurate prediction of the bandgap, such as the dielectric constant. PBE was, however, suitable for DFPT calculations of ionic dielectric properties. It was not feasible to run all calculations using hybrid functionals due to computational cost and incompatibility of DFPT with hybrid/mGGA functionals.”

Given the extensive discussion on mechanisms for weak charge-carrier-phonon coupling, there is also some uncertainty about whether a manuscript in the style of a "Communications" article is appropriate.

Response 3.16: According to the author guidelines for *Nature Communications*, Articles “may range from short communications through to more in-depth studies”. <https://www.nature.com/ncomms/submit/article> Indeed, *Nature Communications* has published many influential papers that have explored a topic in detail⁵²⁻⁵⁶, which we believe is important in this case. We have gone through the paper and trimmed down some of the discussion where it is not necessary to keep the paper as concise as possible.

References for the response letter

- 1 Webber, D. H. & Brutchey, R. L. Alkahest for V_2VI_3 chalcogenides: dissolution of nine bulk semiconductors in a diamine-dithiol solvent mixture. *J Am Chem Soc* **135**, 15722-15725 (2013). <https://doi.org:10.1021/ja4084336>
- 2 Zhao, X., Lu, M., Koeper, M. J. & Agrawal, R. Solution-processed sulfur depleted Cu(In, Ga)Se₂ solar cells synthesized from a monoamine–dithiol solvent mixture. *Journal of Materials Chemistry A* **4**, 7390-7397 (2016). <https://doi.org:10.1039/c6ta00533k>
- 3 Liu, F. *et al.* Low-temperature, solution-deposited metal chalcogenide films as highly efficient counter electrodes for sensitized solar cells. *Journal of Materials Chemistry A* **3**, 6315-6323 (2015). <https://doi.org:10.1039/C5TA00028A>
- 4 Buckley, J. J., Greaney, M. J. & Brutchey, R. L. Ligand Exchange of Colloidal CdSe Nanocrystals with Stibanates Derived from Sb₂S₃ Dissolved in a Thiol-Amine Mixture. *Chemistry of Materials* **26**, 6311-6317 (2014). <https://doi.org:10.1021/cm503324k>
- 5 Arnou, P. *et al.* Solution processing of CuIn(S,Se)₂ and Cu(In,Ga)(S,Se)₂ thin film solar cells using metal chalcogenide precursors. *Thin Solid Films* **633**, 76-80 (2017). <https://doi.org:10.1016/j.tsf.2016.10.011>
- 6 Tian, Q., Cui, Y., Wang, G. & Pan, D. A robust and low-cost strategy to prepare Cu₂ZnSnS₄ precursor solution and its application in Cu₂ZnSn(S,Se)₄ solar cells. *RSC Advances* **5**, 4184-4190 (2015). <https://doi.org:10.1039/C4RA12090F>
- 7 Colombara, D., Peter, L. M., Rogers, K. D., Painter, J. D. & Roncallo, S. Formation of CuSbS₂ and CuSbSe₂ thin films via chalcogenisation of Sb–Cu metal precursors. *Thin Solid Films* **519**, 7438-7443 (2011). <https://doi.org:10.1016/j.tsf.2011.01.140>
- 8 Xue, D. J. *et al.* CuSbSe₂ as a potential photovoltaic absorber material: studies from theory to experiment. *Advanced Energy Materials* **5**, 1501203 (2015). <https://doi.org:https://doi.org/10.1002/aenm.201501203>
- 9 <<http://abulafia.mt.ic.ac.uk/shannon/ptable.php>> (
- 10 Shockley, W. & Bardeen, J. Energy Bands and Mobilities in Monatomic Semiconductors. *Physical Review* **77**, 407-408 (1950). <https://doi.org:10.1103/PhysRev.77.407>
- 11 Wei, S. H. & Zunger, A. Predicted band-gap pressure coefficients of all diamond and zinc-blende semiconductors: Chemical trends. *Physical Review B* **60**, 5404-5411 (1999). <https://doi.org:10.1103/PhysRevB.60.5404>
- 12 Ganose, A. M. *et al.* Efficient calculation of carrier scattering rates from first principles. *Nat Commun* **12**, 2222 (2021). <https://doi.org:10.1038/s41467-021-22440-5>
- 13 Mosquera-Lois, I., Kavanagh, S. R., Walsh, A. & Scanlon, D. O. Identifying the ground state structures of point defects in solids. *npj Computational Materials* **9**, 25 (2023). <https://doi.org:10.1038/s41524-023-00973-1>

- 14 Pham, T. D. & Deskins, N. A. Efficient Method for Modeling Polarons Using Electronic Structure Methods. *Journal of Chemical Theory and Computation* **16**, 5264-5278 (2020). <https://doi.org:10.1021/acs.jctc.0c00374>
- 15 Ning, J., Furness, J. W. & Sun, J. Reliable Lattice Dynamics from an Efficient Density Functional Approximation. *Chemistry of Materials* **34**, 2562-2568 (2022). <https://doi.org:10.1021/acs.chemmater.1c03222>
- 16 Borlido, P. *et al.* Large-Scale Benchmark of Exchange–Correlation Functionals for the Determination of Electronic Band Gaps of Solids. *Journal of Chemical Theory and Computation* **15**, 5069-5079 (2019). <https://doi.org:10.1021/acs.jctc.9b00322>
- 17 Borlido, P. *et al.* Exchange–correlation functionals for band gaps of solids: benchmark, reparametrization and machine learning. *npj Computational Materials* **6**, 96 (2020). <https://doi.org:10.1038/s41524-020-00360-0>
- 18 Zhou, J. *et al.* Solvothermal crystal growth of CuSbQ₂ (Q=S, Se) and the correlation between macroscopic morphology and microscopic structure. *Journal of Solid State Chemistry* **182**, 259-264 (2009). <https://doi.org:https://doi.org/10.1016/j.jssc.2008.10.025>
- 19 Goyal, D., Goyal, C. P., Ikeda, H. & Malar, P. Role of growth temperature in photovoltaic absorber CuSbSe₂ deposition through e-beam evaporation. *Materials Science in Semiconductor Processing* **108**, 104874 (2020). <https://doi.org:https://doi.org/10.1016/j.mssp.2019.104874>
- 20 Chen, T. *et al.* Ultralow Thermal Conductivity and Enhanced Figure of Merit for CuSbSe₂ via Cd-Doping. *ACS Applied Energy Materials* **4**, 1637-1643 (2021). <https://doi.org:10.1021/acsaem.0c02820>
- 21 Tian, Q. *et al.* Versatile and Low-Toxic Solution Approach to Binary, Ternary, and Quaternary Metal Sulfide Thin Films and Its Application in Cu₂ZnSn(S,Se)₄ Solar Cells. *Chemistry of Materials* **26**, 3098-3103 (2014). <https://doi.org:10.1021/cm5002412>
- 22 Zhang, R. *et al.* Metal-metal chalcogenide molecular precursors to binary, ternary, and quaternary metal chalcogenide thin films for electronic devices. *Chem Commun (Camb)* **52**, 5007-5010 (2016). <https://doi.org:10.1039/c5cc09915c>
- 23 Arnou, P. *et al.* Hydrazine-Free Solution-Deposited CuIn(S,Se)₂ Solar Cells by Spray Deposition of Metal Chalcogenides. *ACS Appl Mater Interfaces* **8**, 11893-11897 (2016). <https://doi.org:10.1021/acsaami.6b01541>
- 24 Koskela, K. M., Strumolo, M. J. & Brutchey, R. L. Progress of thiol-amine ‘alkahest’ solutions for thin film deposition. *Trends in Chemistry* **3**, 1061-1073 (2021). <https://doi.org:10.1016/j.trechm.2021.09.006>
- 25 Otamiri, J. C., Andersson, S. L. T. & Andersson, A. Ammoxidation of toluene by YBa₂Cu₃O_{6+x} and copper oxides: Activity and XPS studies. *Applied Catalysis* **65**, 159-174 (1990). [https://doi.org:https://doi.org/10.1016/S0166-9834\(00\)81595-X](https://doi.org:https://doi.org/10.1016/S0166-9834(00)81595-X)
- 26 Velásquez, P. *et al.* A Chemical, Morphological, and Electrochemical (XPS, SEM/EDX, CV, and EIS) Analysis of Electrochemically Modified Electrode Surfaces of Natural Chalcopyrite (CuFeS₂) and Pyrite (FeS₂) in Alkaline Solutions. *The Journal of Physical Chemistry B* **109**, 4977-4988 (2005). <https://doi.org:10.1021/jp048273u>
- 27 Poulston, S., Parlett, P. M., Stone, P. & Bowker, M. Surface Oxidation and Reduction of CuO and Cu₂O Studied Using XPS and XAES. *Surface and Interface Analysis* **24**, 811-820 (1996). [https://doi.org:10.1002/\(sici\)1096-9918\(199611\)24:12<811::Aid-sia191>3.0.Co;2-z](https://doi.org:10.1002/(sici)1096-9918(199611)24:12<811::Aid-sia191>3.0.Co;2-z)
- 28 Wang, D., Miller, A. C. & Notis, M. R. XPS Study of the Oxidation Behavior of the Cu₃Sn Intermetallic Compound at Low Temperatures. *Surface and Interface Analysis*

- 24, 127-132 (1996). [https://doi.org/10.1002/\(sici\)1096-9918\(199602\)24:2<127::Aid-sia110>3.0.Co;2-z](https://doi.org/10.1002/(sici)1096-9918(199602)24:2<127::Aid-sia110>3.0.Co;2-z)
- 29 Biesinger, M. C. Advanced analysis of copper X-ray photoelectron spectra. *Surface and Interface Analysis* **49**, 1325-1334 (2017). <https://doi.org/10.1002/sia.6239>
- 30 Romand, M., Roubin, M. & Deloume, J. P. ESCA studies of some copper and silver selenides. *Journal of Electron Spectroscopy and Related Phenomena* **13**, 229-242 (1978). [https://doi.org/10.1016/0368-2048\(78\)85029-4](https://doi.org/10.1016/0368-2048(78)85029-4)
- 31 Qiu, X., Ji, S., Chen, C., Liu, G. & Ye, C. Synthesis, characterization, and surface-enhanced Raman scattering of near infrared absorbing Cu₃SbS₃ nanocrystals. *CrystEngComm* **15**, 10431-10434 (2013). <https://doi.org/10.1039/C3CE41861H>
- 32 Hsiang, H.-I., Yang, C.-T. & Tu, J.-H. Characterization of CuSbSe₂ crystallites synthesized using a hot injection method. *RSC Advances* **6**, 99297-99305 (2016). <https://doi.org/10.1039/C6RA20692A>
- 33 Wang, C. *et al.* Reactive close-spaced sublimation processed CuSbSe₂ thin films and their photovoltaic application. *APL Materials* **6**, 084801 (2018). <https://doi.org/10.1063/1.5028415>
- 34 Delobel, R., Baussart, H., Leroy, J.-M., Grimblot, J. & Gengembre, L. X-ray photoelectron spectroscopy study of uranium and antimony mixed metal-oxide catalysts. *Journal of the Chemical Society, Faraday Transactions 1: Physical Chemistry in Condensed Phases* **79**, 879-891 (1983). <https://doi.org/10.1039/F19837900879>
- 35 Welch, A. W. *et al.* Trade - Offs in Thin Film Solar Cells with Layered Chalcostibite Photovoltaic Absorbers. *Advanced Energy Materials* **7**, 1601935 (2017). <https://doi.org/10.1002/aenm.201601935>
- 36 Yang, B. *et al.* Hydrazine solution processed CuSbSe₂: Temperature dependent phase and crystal orientation evolution. *Solar Energy Materials and Solar Cells* **168**, 112-118 (2017). <https://doi.org/10.1016/j.solmat.2017.04.030>
- 37 Jagt, R. A. *et al.* Layered BiOI single crystals capable of detecting low dose rates of X-rays. *Nat Commun* **14**, 2452 (2023). <https://doi.org/10.1038/s41467-023-38008-4>
- 38 Lal, S. *et al.* Bandlike Transport and Charge-Carrier Dynamics in BiOI Films. *J Phys Chem Lett* **14**, 6620-6629 (2023). <https://doi.org/10.1021/acs.jpcllett.3c01520>
- 39 Wu, B. *et al.* Strong self-trapping by deformation potential limits photovoltaic performance in bismuth double perovskite. *Sci Adv* **7**, eabd3160 (2021). <https://doi.org/10.1126/sciadv.abd3160>
- 40 Devreese, J. T. Encyclopedia of Applied Physics. *Wiley-VCH Publishers, Inc* **14**, 383-409 (1996).
- 41 Peeters, F. M. & Devreese, J. T. in *Solid State Physics* Vol. 38 (eds Henry Ehrenreich, David Turnbull, & Frederick Seitz) 81-133 (Academic Press, 1984).
- 42 Devreese, J. Fröhlich Polarons. Lecture course including detailed theoretical derivations. *arXiv e-prints*, arXiv: 1012.4576 (2010). <https://doi.org/10.48550/arXiv.1012.4576>
- 43 Giustino, F. Electron-phonon interactions from first principles. *Reviews of Modern Physics* **89**, 015003 (2017). <https://doi.org/10.1103/RevModPhys.89.015003>
- 44 Franchini, C., Reticcioli, M., Setvin, M. & Diebold, U. Polarons in materials. *Nature Reviews Materials* **6**, 560-586 (2021). <https://doi.org/10.1038/s41578-021-00289-w>
- 45 Yu, L., Kokenyesi, R. S., Keszler, D. A. & Zunger, A. Inverse Design of High Absorption Thin - Film Photovoltaic Materials. *Advanced Energy Materials* **3**, 43-48 (2012). <https://doi.org/10.1002/aenm.201200538>

- 46 Maeda, T. & Wada, T. First-principles study of electronic structure of CuSbS₂ and CuSbSe₂ photovoltaic semiconductors. *Thin Solid Films* **582**, 401-407 (2015). <https://doi.org:10.1016/j.tsf.2014.11.089>
- 47 Gaillac, R., Pullumbi, P. & Coudert, F.-X. ELATE: an open-source online application for analysis and visualization of elastic tensors. *Journal of Physics: Condensed Matter* **28**, 275201 (2016). <https://doi.org:10.1088/0953-8984/28/27/275201>
- 48 Franceschetti, A., Wei, S.-H. & Zunger, A. Absolute deformation potentials of Al, Si, and NaCl. *Physical Review B* **50**, 17797-17801 (1994). <https://doi.org:10.1103/PhysRevB.50.17797>
- 49 Li, Y.-H., Gong, X. G. & Wei, S.-H. *Ab initio* all-electron calculation of absolute volume deformation potentials of IV-IV, III-V, and II-VI semiconductors: The chemical trends. *Physical Review B* **73**, 245206 (2006). <https://doi.org:10.1103/PhysRevB.73.245206>
- 50 Petousis, I. *et al.* Benchmarking density functional perturbation theory to enable high-throughput screening of materials for dielectric constant and refractive index. *Physical Review B* **93**, 115151 (2016). <https://doi.org:10.1103/PhysRevB.93.115151>
- 51 Petousis, I. *et al.* High-throughput screening of inorganic compounds for the discovery of novel dielectric and optical materials. *Sci Data* **4**, 160134 (2017). <https://doi.org:10.1038/sdata.2016.134>
- 52 Yang, Z. *et al.* Ultrafast self-trapping of photoexcited carriers sets the upper limit on antimony trisulfide photovoltaic devices. *Nat Commun* **10**, 4540 (2019). <https://doi.org:10.1038/s41467-019-12445-6>
- 53 Giannini, S. *et al.* Quantum localization and delocalization of charge carriers in organic semiconducting crystals. *Nature Communications* **10**, 3843 (2019). <https://doi.org:10.1038/s41467-019-11775-9>
- 54 Dhital, C. *et al.* Carrier localization and electronic phase separation in a doped spin-orbit-driven Mott phase in Sr₃(Ir_{1-x}Ru_x)₂O₇. *Nature Communications* **5**, 3377 (2014). <https://doi.org:10.1038/ncomms4377>
- 55 Fu, J. *et al.* Hot carrier cooling mechanisms in halide perovskites. *Nature Communications* **8**, 1300 (2017). <https://doi.org:10.1038/s41467-017-01360-3>
- 56 Fang, H.-H., Adjokatse, S., Shao, S., Even, J. & Loi, M. A. Long-lived hot-carrier light emission and large blue shift in formamidinium tin triiodide perovskites. *Nature Communications* **9**, 243 (2018). <https://doi.org:10.1038/s41467-017-02684-w>

Reviewer comments in Arial. Our responses and quotes from the paper in Times New Roman.

Reviewer 2:

The response from the authors well addresses concerns and questions raised by me.

Response: We would like to thank the reviewer for their time in evaluating our paper, and appreciate their constructive comments, which have improved our paper. We are glad that they are satisfied with our revisions and are happy to accept our paper for *Nature Communications* as-is.

Reviewer 3:

I would like to thank the authors for their great effort in addressing both my comments and those of other reviewers. The manuscript's clarity has significantly improved, particularly regarding the computational part of the work. I agree with the removal of mobility calculations which indeed can potentially divert attention from the manuscript's key messages. Now the manuscript lacks any substantial quantitative comparison between calculations and experiments. Nonetheless, qualitatively, computational analysis remains valuable in interpreting the data.

The only comment made by the authors in the rebuttal letter and is not reflected in the manuscript is the following:

"In regard to other methods of calculating electron-phonon coupling, we agree that this is an exciting and rapidly growing area of research, but would stress that due to the sheer number of methods available (e.g., ab-initio MD, Quantum Monte-Carlo, the Special Displacement Method, the ab-initio theory of polarons of Sio et al.), an in-depth discussion of recent developments is outside the scope of this work."

I believe it's worth acknowledging the above methods in the main manuscript as promising avenues for future research. Using these methods could potentially lead to more accurate calculations of polarons and electron-phonon coupling, thereby facilitating a better quantitative comparison with experimental results. Including these approaches in the manuscript as future directions may enhance its scope and contribute to advancing the field.

After this comment is addressed, I believe the manuscript is suitable for publication in *Nature Communications*.

Response: We are glad for the time from the reviewer in carefully evaluating our paper, both in the first submission and in the revision. We are delighted to see that the reviewer is satisfied to accept the paper for *Nature Communications* after addressing the one remaining comment, which we have now completed. We have accordingly modified the discussion section on page 27 as follows:

"Note that other, more sophisticated methods exist to calculate electron-phonon coupling and model polarons (including ab-initio molecular dynamics¹⁰⁶⁻¹⁰⁸, AHC theory¹⁰⁹⁻¹¹¹, Quantum Monte-Carlo simulations¹¹²⁻¹¹⁴ the Special Displacement Method¹¹⁵⁻¹¹⁷, and the ab-initio theory of polarons of Sio et al¹¹⁸.), but these are cutting-edge methods and go beyond the scope of this work. Nevertheless, fully investigating them in future work can lead to better quantitative agreement between theory and experiment, in addition to qualitative agreement."

Reviewer 4:

In this revised manuscript, the authors explore factors that can delocalize carriers in pnictogen-based solar absorbers, using CuSbSe₂ as a case study. They propose that carrier delocalization in CuSbSe₂ is influenced by (a) weak coupling to acoustic phonons due to a low deformation potential, and (b) weak coupling to optical phonons due to a relatively low ionic contribution to the dielectric constant compared to the electronic contribution. While the paper is well-written and is praiseworthy for the synthesis advancement, I believe it does not provide insights or results significant enough to warrant publication in Nature Communications.

The assertion that reducing carrier coupling with phonons—whether acoustic or optical—is beneficial for delocalization is a known principle that researchers with a solid understanding of chemistry are likely familiar with. The insights provided, such as the relationship between low deformation potential and reduced carrier localization for reducing coupling with acoustic modes, or the impact of electronic versus ionic contributions to the dielectric constant in determining the coupling with optical modes, are generally well-understood principles in the field. In this context, the connection between layered structures and low deformation potential is also a predictable outcome, as is the expectation that high bandgap materials will have a lower electronic contribution to the dielectric constant.

Response 4.1: We would like to firstly thank the reviewer for their time in taking up the task of appraising our revisions made in response to the comments from Reviewer 1, who unfortunately did not have time to do this themselves. We also appreciate the extra thoughts and opinions of the reviewer, and are grateful for the opportunity to clarify the misunderstanding around the conceptual novelty of our work.

The conceptual advance is not that reducing the strength of electron-phonon coupling reduces the likelihood of carrier localization, but rather *how* this can be achieved in heavy pnictogen-based semiconductors. We emphasized this point at the start of the discussion section:

“In light of the results presented, it can be seen that carrier localization is not present in CuSbSe₂ ... This is unusual among heavy pnictogen-based perovskite-inspired materials, and it is critical to unravel the underlying structural, electronic and chemical factors. In this discussion, we will examine computationally how the crystal structure and bonding in CuSbSe₂ result in the low deformation potentials and low Fröhlich coupling constants, as well as the effect on the electronic dimensionality in both the valence and conduction bands.”

Indeed, this is not straightforward with Bi- and Sb-based semiconductors. As we explained in the introduction of the paper, these materials have come to the fore of attention over the past decade in efforts to develop lower-toxicity alternative solar absorbers to lead-halide perovskites, based on ideas around defect tolerance^{1,2} (these are commonly referred to as “perovskite-inspired” materials, or PIMs). However, a significant drawback of these materials is the presence of carrier localization. Such carrier localization has been widely found with PIMs, including in Cs₂AgBiBr₆³, Cs₂AgSbBr₆⁴, Cu₂AgBiI₆^{5,6}, Ag-Bi-I semiconductors⁷, A₃B₂I₉ compounds (where A = Cs or Rb, B = Bi or Sb)⁸, Cs₃Bi₂Br₉⁹, BiI₃¹⁰, AgBiS₂¹¹, and NaBiS₂¹². Self-trapping has also been reported in Sb₂S₃ and Sb₂Se₃^{13,14}, although this remains under debate. Carrier localization in heavy pnictogen-based PIMs has been so widely found that it is being referred to as a “hallmark” of these materials^{3,6,12,13,15-18}, which implies that a growing number of the community have been resigned to always finding self-trapping in these materials, which fundamentally limits their performance in solar cells. The wider community is indeed aware of the fundamentals of electron-phonon coupling, but despite this knowledge, there have been no substantive suggestions put forward on how the chemistry of these pnictogen-based materials could be changed to

avoid this limitation, or indeed if it is fundamentally possible to achieve entirely band-like transport^{3,5,7,8,13-15,19}.

To our knowledge, the first hint that it is possible to find delocalized charge-carriers in PIMs is in our recent works on BiOI, where we showed high mobilities up to $83 \text{ cm}^2 \text{ V}^{-1} \text{ s}^{-1}$,²⁰ far exceeding the values typical of small polarons ($<0.1\text{--}10 \text{ cm}^2 \text{ V}^{-1} \text{ s}^{-1}$)^{12,21}, along with photoconductivities that did not decay within a ps²². These are exciting demonstrations that fully band-like transport in PIMs is possible, but they do not explain *why* this can happen, or *how* this may be found in other PIMs. These are the very questions we sought to address in this work, where we took CuSbSe₂ as a case-study (refer to pages 8–9 of the introduction for our detailed justification for focussing on this material).

In this work, we identify several key physical-chemical factors that enable band-like transport in CuSbSe₂. These are covered in the discussion section of the paper, and we encourage the reviewer to go back over the details in this section. To summarize:

- By having regular gaps in the crystal structure, the distortions to the unit cell during the propagation of an acoustic wave are mostly relaxed into these interlayer gaps, minimizing changes to the bonds that contribute to the electronic structure at the band extrema. This then minimizes the deformation potential. Remarkably, these low deformation potentials are not only in the *c*-axis direction, but also in the *a*- and *b*-axis directions;
- There are quasi-bonding interactions (see below for definition) between atoms across the interlayer gaps, such that the electronic dimensionality at band-edges is 2D or higher, which reduces the likelihood of the system being thermodynamically favoured to relax into a self-trapped state, especially when combined with the low acoustic coupling constant. This high electronic dimensionality also contributes to the low exciton binding energy of CuSbSe₂ ($9\pm 4 \text{ meV}$), such that free carriers occur in this material;
- The strength of Fröhlich coupling is minimized by having low ionic dielectric contributions compared to the electronic contributions. This is not only due to the small bandgap, but importantly, also because CuSbSe₂ has species with low Born effective charges compared to the formal charges of the elements

Arriving at these findings required the detailed and thorough computational analyses into the structure and bonding that we have performed, and indeed some of the important details are not predictable based on general background knowledge. For example, it is not intuitive that a distortion to the unit cell along the *a*-axis of CuSbSe₂ (due to the propagation of an acoustic wave; refer to Supplementary Fig. 11) would lead to a low acoustic deformation potential, since this distortion is within and along each layer (Fig. R1a), and is not punctuated with regular gaps in the structure (unlike the *c*-axis direction). Yet, the *a*-axis valence band deformation potential is lower than that along the *c*-axis (Table 1; copied below for convenience). This can be understood by first analyzing the bonding environment in CuSbSe₂. To do this, we computed the crystal orbital Hamilton population (COHP; shown in Fig. 2b, which is copied below). These COHP calculations showed that the valence band maximum (VBM) primarily has Cu-Se antibonding character, and the deformation potentials will therefore depend on how the CuSe₄ tetrahedra are distorted as the unit cell is deformed along the three principal axes. Distortion along the *a*-axis leads to a scissoring of the Cu-Se bonds, with minimal changes to the bond lengths. By contrast, distortion along the *c*- and *b*-axes lead to more changes in bond length. This shows the importance of considering the geometry and orientation of bonding groups to the principal axes, which can affect the size and isotropy of the deformation potentials.

Fig. R1 | Crystal structure of **a**, CuSbSe₂, **b**, MoS₂, with the thickness of each layer labelled. Part **a** is an original figure for this paper. Part **b** reproduced with permission from Zhao, W., Pan, J., Fang, Y., Che, X., Wang, D. Bu, K. & Huang, F. Metastable MoS₂: Crystal Structure, Electronic Band Structure, Synthetic Approach and Intriguing Physical Properties. *Chem. Eur. J.*, **24**, 15942 (2018). © 2018 Wiley-VCH Verlag GmbH & Co. KGaA, Weinheim.

Table 1 | Calculated properties related to carrier-phonon coupling in CuSbSe₂ along different principal axes.

	a	b	c	Average ^d
a_o (Å)	6.457	4.034	14.929	
E_d^{VBM} (eV)	1.16	1.93	2.11	1.73
E_d^{CBM} (eV)	6.60	6.32	6.62	6.51

Fig. 2 | **Optical and electronic properties of CuSbSe₂ ... b**, Electronic band structure of CuSbSe₂ (left panel; the highest occupied state set to 0 eV), along with electronic density of states curves (middle panel), and crystal orbital Hamilton population (COHP) diagram (right panel). The bonding and anti-bonding interactions are represented by blue and orange, respectively.

In our calculations of the deformation potential, we went further in our analyses than typically found in the wider literature. That is, we investigated the effects of relaxing the atomic positions after distorting the unit cell to simulate the effects of a propagating acoustic wave. In the conventional method of calculating deformation potentials by Wei and Zunger²³, the internal degrees of freedom are not relaxed after the volume change. This is because this prior analysis was made for II-VI compounds, where the atoms sit on high-symmetry sites. However, this relaxation is necessary for layered compounds, and resulted in smaller changes in bond lengths, thus leading to lower deformation potentials (Supplementary Table 4). Such detailed treatment is not commonly found, and this is a limitation of using deformation potentials from databases obtained from high-throughput calculations²⁴.

High electronic dimensionality is also important. This is illustrated by comparing CuSbSe₂ with MoS₂. Whereas we have found CuSbSe₂ to exhibit fully band-like transport, MoS₂ has self-trapping^{7,10,25,26}, even though both materials are layered (refer to Fig. R1). A key difference is that CuSbSe₂ has both Sb

and Se species across the interlayer gap, whereas MoS₂ only has the chalcogen^{27,28}. As shown in our COHP calculations in Fig. 2b and 4a, Sb 5p and Se 4p antibonding states dominate the conduction band minimum (CBM) density of states. By having both species positioned across the interlayer gap, there is partial hybridization between them (we refer to these as ‘quasi-bonds’), such that the interlayer interactions are strong (refer to Fig. 4a). As a result, the CBM has 3D electronic dimensionality, which substantially reduces the likelihood of small polaron formation. On the other hand, the VBM, dominated by Cu-Se bonding, is electronically 2D (Supplementary Fig. 10). This is because Cu from each layer is not involved in quasi-bonding with Se from the neighbouring layer (Fig. R1a). But because of the low deformation potentials at the VBM, holes are not self-trapped. Another point of difference is that MoS₂ has much thinner layers (Fig. R1), below its Bohr diameter (3.0 Å thickness vs. 3.2 Å Bohr diameter²⁹), and this could also favour exciton formation and localization.

Fig. 4 | Computational analysis of CuSbSe₂. **a**, Structure of CuSbSe₂, with key atoms labelled, and the interlayer distance defined as the perpendicular distance between Sb2 and Sb3. **b**, Percentage changes in bond lengths and interlayer distance of CuSbSe₂ as a function of strain along the *c*-axis. All calculated bond lengths shown are after relaxation of the atoms in the structure after distortion, *i.e.*, calculations for equilibrated structures as shown (refer to discussion in Supplementary Note 7). A disproportionately large change in the interlayer distance is observed as compared to bond lengths for a given strain. **c**, Calculated crystal orbital Hamiltonian population (COHP) per bond of in-layer (dash line) and interlayer (solid line) Sb-Se bonds. The bonding and anti-bonding interactions are represented by blue and orange, respectively. **d**, Fermi iso-surface 0.1 eV below the VBM (top figure) and above the CBM (bottom figures).

To complete our discussion, it is important to also discuss Fröhlich coupling. Prior work put forward ideas of tuning this parameter by adjusting the masses of species present, along with the bond strengths/ionicity, through their effects on the LO frequency and dielectric constants^{21,30,31}. However, while the relationship between the dielectric response factor (*i.e.*, difference between the inverse of the electronic and static contributions to the dielectric constant) and Born effective charge (BEC) is well-known, tuning BECs and thus the dielectric response / Fröhlich coupling has not been thoroughly explored in practice^{14,21,30,31}. Our work finds that Sb in CuSbSe₂ exhibits lower Born effective charges along the *a*- and *c*-axes than Sb or Bi in other compounds (Supplementary Table 5) – in fact *lower* than the +3 formal oxidation state, contributing significantly to the low dielectric response factor. This is an interesting, non-intuitive finding that requires further detailed computational analyses beyond the scope of this work (*e.g.*, probing in detail the changes in polarization upon displacing the Sb atom). These findings also make the important point that when it comes to materials design, we should not simply maximize BEC to obtain high dielectric constants that screen out charged defects, as previous works have suggested for defect tolerance¹, but rather should fine tune the BEC to also balance with the need to minimize Fröhlich coupling (favoured by lower BECs).

Overall, our detailed investigation into CuSbSe₂ puts forward several important and interesting findings that shape our understanding of how band-like transport can be achieved in heavy-pnictogen-based semiconductors. We show the importance of considering the geometry and orientation of structural groups relative to the principal axes. For example, in the case of CuSbSe₂, having CuSe₄ oriented at an angle to the principal axes results in distortions being relaxed as changes in bond angle rather than bond length, reducing deformation potentials. We also show the benefits of layered structures, not just in allowing distortions along the *c*-axis to be relaxed in interlayer gaps, but in having the degrees of freedom to allow bonds to relax back to their equilibrium lengths, further reducing deformation potentials. This also emphasizes that future high-throughput calculations of acoustic deformation potentials in 2D (and likely also 1D and 0D materials) need to allow re-equilibration after distortion. At the same time, we show the importance of surface atoms of the layers contributing to the band-edge electronic states (unlike transition metal dichalcogenides for instance). By having quasi-bonding across interlayer gaps, electronic dimensionality is increased, further reducing the likelihood of carrier localization. None of these factors have been discussed in previous works on carrier localization in halide perovskites and pnictogen-based semiconductors^{7,10,25,26}, and our findings pave the way to more successfully discovering heavy-pnictogen-based materials with band-like transport.

To make these points clearer and avoid misunderstandings, we have made the following changes.

Abstract:

*“Inorganic semiconductors based on heavy pnictogen cations (Sb³⁺ and Bi³⁺) have gained significant attention as potential nontoxic and stable alternatives to lead-halide perovskites for solar cell applications. A limitation of these novel materials, which is being increasingly commonly found, is carrier localization, which substantially reduces mobilities and diffusion lengths. Herein, the layered *přibramite* CuSbSe₂ is investigated and discovered to have delocalized free carriers, as shown through optical pump terahertz probe spectroscopy and temperature-dependent mobility measurements. Using a combination of theory and experiment, the critical enabling factors are found to be: 1) having a layered structure, which allows distortions to the unit cell during the propagation of an acoustic wave to be relaxed in the interlayer gaps, with minimal changes in bond length, thus limiting deformation potentials; 2) favourable quasi-bonding interactions across the interlayer gap giving rise to higher electronic dimensionality; 3) Born effective charges not being anomalously high, which, combined with the small bandgap (≤ 1.2 eV), result in a low ionic contribution to the dielectric constant compared to the electronic contribution, thus reducing the strength of Fröhlich coupling. These insights can drive forward the rational discovery of heavy pnictogen-based semiconductors that avoid carrier localization.”*

Discussion – effects of orientation (page 29):

“From Table 1, it can be seen that there is more anisotropy in the deformation potentials at the VBM than CBM. This anisotropy in the VBM deformation potentials can be understood by considering the inequivalent distortions of the Cu-Se tetrahedra caused by strains along the principal crystallographic axes. Whilst all Cu-Se bonds are equivalent in both bond length and strength, these tetrahedra are arbitrarily rotated with respect to the principal axes ...strain along the a-axis causes scissoring of pairs of Cu-Se bonds rather than significant changes in bond length, whereas strain along the b axis distorts all four bonds.”

Discussion – nature of layered structures and bonding (page 30):

“The magnitude of the deformation potential is substantially reduced by structural relaxation in this flexible crystal structure, as outlined in Supplementary Note 7 ... When the strain reached $\pm 5\%$, the changes in most bond lengths were below 1%, and the maximum change (Cu1-Se3 bond) was only around 2%. This is not explained by misalignment of the strain to bonding vectors, as we see differences of more than 4% for the same Cu1-Se3 bond in the unrelaxed case (i.e., for a uniform distribution of strain along the inter-atomic distances), and is also in contrast to the large change in interlayer distance of $\pm 20\%$ under $\pm 5\%$ c-axis strain. The fact that Cu-Se bonds exhibit more changes than Sb-Se bonds agrees with the COHP calculation results that indicate that the Cu-Se bonds are overall weaker due to the filled antibonding states in the VBM ... This phenomenon should be considered when calculating deformation potentials in complex materials with similar structures to BiOI^3 and CuSbSe_2 .”

Discussion – electronic dimensionality (page 34):

“The near-3D nature of the CBM is consistent with the lower CB being dominated by Sb-Se antibonding states, and there being weak interactions between the Sb and Se species across the interlayer gaps (Fig. 4c), which we refer to as quasi-bonding. By contrast, the 2D nature of the VBM is consistent with Cu-Se interactions, which dominate the upper VB, mostly occurring within each layer. The combination of the relatively high electronic dimensionality (especially in the CBM) and low g_{ac} values overall are consistent with the band-like transport in CuSbSe_2 .”

Conclusions (pages 40-41):

“In conclusion, we have found CuSbSe_2 to be a heavy pnictogen-based chalcogenide that can avoid charge-carrier localization, which we determined through a combination of experiment and computations. A novel thiol-amine solution processing method was employed to achieve phase-pure CuSbSe_2 thin films. OPTP measurements on CuSbSe_2 revealed a timescale of 6.7 ps to reach 50% photoconductivity decay, substantially slower than if carrier localization were present. Temperature-dependent Hall effect measurements confirmed the presence of large polarons based on the decrease in mobility with increases in temperature. Through DFT calculations, we found that both the acoustic and Fröhlich coupling constants are lower than those of many other heavy pnictogen-based materials, which support the finding that CuSbSe_2 has weaker charge-carrier-phonon coupling. Whilst the effect of the deformation potential on the acoustic coupling strength, and relative size of the dielectric response factor on the strength of Fröhlich coupling are well established, it was not clear how these parameters could be tuned to achieve delocalized charge-carriers in heavy pnictogen-based semiconductors. In this work, we performed detailed computational investigations to reveal the factors involved, focussing on the bonding/anti-bonding nature of the crystal orbitals at the band extrema, and changes in bond lengths and interlayer spacing as a function of distortions, as well as the Born effective charges of ions. In particular, we show that deformation potentials can be minimized by having distortions to the unit cell due to the propagation of an acoustic wave relaxed through changes in geometry rather than bond length. This could be achieved through a layered structure, which provides sufficient degrees of freedom to allow bonds to mostly relax back to their equilibrium lengths following distortion. This could also be achieved by having structural groups of atoms contributing to the orbitals

at band extrema (e.g., CuSe₄ tetrahedra) oriented at an angle to the principal axes, such that distortions are relaxed as changes in inter-group bond angles rather than intra-group bond lengths. Coupled with high electronic dimensionality (through significant interlayer bonding contributions to the band edges), strong coupling to acoustic phonons is avoided. Meanwhile, the weak Fröhlich coupling is due to the high electronic contribution (mostly due to the small bandgap) and low ionic contribution to the dielectric constants. The latter arises from the Born effective charges of Sb, Cu and Se not substantially deviating from their formal oxidation states (in contrast to lead-halide perovskites)¹²⁹. This makes the important point that when it comes to materials design and the Born effective charge of species, there is a balance required between reducing Fröhlich coupling (lower BECs) and defect tolerance through dielectric screening (higher BECs). Overall, the insights made in this work are valuable for the future design of solar absorbers that have band-like transport.”

Nevertheless, the manuscript presents interesting points regarding the nature of bonding/antibonding interactions near the valence band maximum (VBM) and conduction band minimum (CBM) and their impact on electronic dimensionality and deformation potential. Similarly, the discussion involving the effect of Born effective charges is also interesting. However, these insights are of limited practical use without known band structure data of a system. The authors do not provide sufficient guidance on predicting these properties from atomic composition or structure alone, which is a critical limitation in materials selection.

Response 4.2: We are glad to see that the reviewer appreciates the in-depth insights that we provide in our work. In reading through these comments, we understand that the reviewer is alluding to materials design through use of simple descriptors that will facilitate high throughput materials discovery (*i.e.*, without the need for computing electronic structure).

This is certainly our ultimate goal, and, indeed, would be the ultimate goal for the wider community. But we emphasize the novel nature of our findings. As explained in Response 4.1, the field has only very recently come to the realization that it is possible to find PIMs with band-like transport. Our work showing the chemical-physical factors that enable this is a significant step forward.

The next step is to verify the broader applicability of our design principles, which we fully expect to be generalizable (more on this in Response 4.8). That is, we would synthesize and test materials with similar (*e.g.*, CuSbS₂, CuBiS₂, CuBiSe₂) and dissimilar structures (*e.g.*, Cu₃SbSe₃), and different Born effective charges (*e.g.*, compare Sb vs. Bi as the pnictogen cation, and compounds where the stereochemical activity of the lone pair is expressed vs. not) to experimentally verify how robust these principles are. We emphasize right now most properties of materials cannot be simply predicted from composition and structure alone, and this has motivated the development of electronic structure databases, such as Materials Project³³. As such databases become larger and more information-rich (*e.g.*, like those including quantum-chemical bonding analysis)³⁴, we expect to be able to query chemical descriptors—such as the COHP analysis we have done here—in addition to structural and compositional descriptors. We therefore fully anticipate that by using the detailed case study we provide here, it will be possible to screen for similar features and test the generalizability of our results in other materials.

Such work would pave the way for a targeted study of the most promising materials using state-of-the-art computations with the aim of developing simple chemical and structural descriptors for predicting carrier localization. But this step is a significant undertaking. If we look at similar efforts being performed now in the much more mature field of carrier recombination, this is still a significant challenge³⁵. Thus, we appreciate the reviewer’s point, and agree with them on this as the ultimate goal.

But fulfilling this goes well beyond the scope of this work, and will be undertaken in many follow-on works.

As detailed in the response above and better clarified in our manuscript, our work identifies these relevant material properties which disfavour carrier localization in this system, as well as identifying the mechanisms through which they act (*i.e.*, effects on acoustic deformation potentials, electronic dimensionality and Fröhlich coupling) and the compositional/structural features from which they arise (*e.g.*, rigid structural groups surrounded by deformable regions, surface atoms which contribute to band edge states etc.), pushing forward this objective of establishing clear structure-property relationships in the context of carrier localization.

To capture these points, we have added a new future work section to the end of the discussion (page 38):

“Future directions

Based on our investigations, we propose that free volumes (e.g. interlayer gaps) in the structure can help to minimize the effect of structural distortions on the bonding environment and lower the deformation potential. We proposed that this does not necessarily need to be in the form of a layered structure, but could also be achieved in motifs where there is a regular soft layer of species (e.g., molecular species) that do not contribute to orbitals at the band extrema. At the same time, quasi-bonding across these regular gaps between species contributing to the band-edge density of states is important for increasing the electronic dimensionality, which reduces the likelihood of self-trapping. This could be found more generally, for example, in materials that exhibit stereochemical activity of the pnictogen cation (e.g., CuSbS_2^{127} , CuBiS_2^{70} and CuBiSe_2^{128}), resulting in a layered structure, with both the pnictogen and chalcogen placed about the interlayer gaps, allowing quasi-bonding to take place between them. Finally, materials with low ionic contributions to the dielectric constant are desired to minimize Fröhlich coupling, but this needs to be balanced with the effect on the capture cross-section of charged defects.

We believe that these insights, gained from investigating CuSbSe_2 , are generalizable because the key structural and electronic features can be found in other materials, and the ways in which they affect carrier localization are rationalized based on the fundamentals of electron-phonon coupling theory (rather than bespoke theory specific to only CuSbSe_2). The important next step will be to test the wider applicability of these principles in broader sets of materials, particularly making use of the computational approaches we developed in this work. These efforts could ultimately lead to the development of simple descriptors for the high-throughput inverse design of heavy-pnictogen-based semiconductors with band-like transport.”

While the authors have addressed previous technical comments adequately, the manuscript does not convincingly advance the field of pnictogen-based solar absorbers. Therefore, I cannot recommend it for publication in its current form. Nonetheless, the discussions are valuable and may be suitable for a different publication venue.

Response 4.3: We are glad that the reviewer is satisfied that we have addressed the previous technical comments from the original reviewers. We hope that the elaborate explanations in Responses 4.1 and 4.2 clarify the conceptual novelty and importance of this work, and the many important future directions opened up. We note that Reviewers 2 and 3, who have appraised this work since its first submission, recognized the conceptual advance, and now recommend this work for publication in *Nature Communications*. Their comments in the latest peer review can be found at the start of this document.

For the convenience of Reviewer 4, we have copied out some of their comments from the first round of peer review:

“This research holds significance for the design and optimization of solar absorbers. The unique characteristics of CuSbSe₂, particularly its ability to resist charge carrier localization, position it as a potential solar absorption material. This contributes to the advancement of more efficient photovoltaic materials.” (Reviewer 2)

“Unlike other Sb- and Bi-based inorganic semiconductors, they have demonstrated that CuSbSe₂ exhibits delocalized free carriers through mobility measurements as a function of temperature and optical pump terahertz probe spectroscopy; those results together with the solution processing method are the most important contributions.” (Reviewer 3)

Furthermore, the authors can investigate these concerns to improve the discussion part in the manuscript.

1. The authors are encouraged to give temperature dependent resistivity plot for more clarity of the carrier transport.

Response 4.4: We have accordingly included a plot showing the change in resistivity with temperature. There is a decrease in resistivity with increasing temperature, despite a reduction in charge-carrier mobility due to high carrier concentrations with rising temperatures, as more charge-carriers are excited across the bandgap.

Supplementary Fig. 8 | Temperature-dependent resistivity of CuSbSe₂ thin films determined using Hall effect measurements.

2. The authors should properly investigate exact temperature dependence of the mobility data and then discuss about the transport mechanism.

Response 4.5: We appreciate this comment from the reviewer. It is common to fit the temperature dependence of the mobility *versus* temperature, and use the exponent to comment on the dominant scattering mechanism. However, a recent computational work analyzed over 23 000 materials, and found that the temperature dependence of the mobility is in fact not a good indicator of the scattering mechanism (<https://doi.org/10.48550/arXiv.2210.01746>). Rather, if a power law is fit to the mobility *versus* temperature, the exponent was found to depend more on the phonon frequency. Given the

controversy in the field, we believe that it would not help our paper to speculate on the transport mechanism based on the fitted exponent.

Fig. 3 | Spectroscopic and temperature-dependent studies on carrier-phonon coupling in CuSbSe₂ ... b, Temperature-dependent mobility of CuSbSe₂ thin films determined using Hall effect measurements, fit using a power law model that indicates $\mu \propto T^{-1.2}$. The point at the lowest temperature (121 K) is not included in the fit due to its higher standard deviation.

We would like to mention to the reviewer that in our original submission, we included in the Supplementary Information our calculations of the temperature-dependence of mobility. These calculations were performed using the *ab-initio* Scattering and Transport (AMSET) package²⁴. Using AMSET, we found that over the entire temperature range considered, polar optical polaron (Fröhlich-like) scattering dominates over acoustic deformation potential (ADP) scattering, and is unrelated to its temperature dependence. This can be seen from the fact that the overall mobility is close to the POP mobilities and well below the ADT mobilities for the entire temperature range (Fig. R2 vs. Fig. 3b).

If we fit a power-law decay model to the experimentally-measured mobility data, it gives a power law exponent of -1.2 (Fig. 3b, above). Meanwhile, fitting a power law to the calculated temperature-dependent overall mobility reveals that $\mu \propto T^{-1.1}$ (Fig. R2, below) demonstrating strong consistency between experiment and computations.

Fig. R2 | Computational analysis of mobility. Calculated temperature-dependent total mobility (black circles) and mobilities when acoustic deformation potential (ADP, red upward triangles) scattering or polar optical phonon (POP, pink downward triangles) scattering dominates. The power law fit (red line) indicates that $\mu \propto T^{-1.1}$. The mobility values limited by ionized impurity scattering (IMP) are too high to be included in the figure.

However, we note that the calculated mobilities exceed the experimental values. This can be for a wide range of reasons, such as optical deformation potential scattering, grain boundary scattering, or defect scattering. In the previous revision, we reached an agreement with the other reviewers to remove the calculated mobilities from the paper because the difference between these values and experimental values was too distracting to the main message. For our paper, what matters most is an understanding of whether large or small polarons exist in CuSbSe₂. The experimental temperature-dependent Hall data clearly shows large polaron behaviour, and this is verified through optical pump terahertz probe spectroscopy measurements, as well as computations (see Response 4.6 for more details).

We agree with the reviewer that it would be helpful to wider community for us to show the fit to the temperature-dependent mobility data, and we have accordingly modified Fig. 3b in the main text as shown above. In addition, we added the following to the main text on page 21:

“We fit a power law model to the mobility data, and found the exponent to be -1.2 (Fig. 3b), however interpreting the scattering mechanism based on this exponent has recently been revealed as not straightforward¹⁰¹.”

3. Can the authors compare the magnetoresistance of the materials having localized carriers with those having delocalized carriers? Magnetoresistance of small polaron and large polaron should be different.

Response 4.6: We thank the reviewer for this interesting suggestion. The role of polarons on accounting for the colossal magnetoresistance of manganites has been studied³⁶⁻³⁸, although the exact mechanism in this specific class of materials is still not well understood³⁹, but is associated with small polarons. Furthermore, magnetoresistance is affected by other factors, such as spin-orbit coupling⁴⁰. If we look at the wider literature on polarons, it is rare to compare the magnetoresistance as a way to determine whether large or small polarons exist⁴¹. Given the many factors that can contribute to magnetoresistance, it is difficult to use this to make an unambiguous comment on the nature of polarons.

We understand that the motivation behind the reviewer’s question was to verify that large polarons exist in CuSbSe₂. We have thoroughly done this using a combination of experiment, spectroscopy and computations. Experimentally, we have shown that the mobility decreases with increasing temperature, which is characteristic of large polarons and is completely the opposite behaviour to small polarons (Fig. 3b in the main text). Through optical pump terahertz probe spectroscopy, we provide direct experimental evidence for the absence of rapid carrier localization (Fig. 3a; refer to details in Supplementary Note 5). For computations, we have thoroughly evaluated the deformation potential, acoustic coupling constant, Fröhlich coupling constant, phonon dispersion curve and Born effective charges. In addition, as part of the previous revision, we performed new computations that directly show the absence of small polarons, as detailed below:

We used state-of-the-art calculations using the *ShakeNBreak* method⁴², to explicitly model polarons in CuSbSe₂ as dilute charges in a supercell. As applied to bulk polarons, the *ShakeNBreak* method is effectively an evolution of the bond distortion method with the addition of a stochastic rattle to the perturbed/distorted structures. The bond distortion method has been used previously to search for low-energy polaronic states and thus calculate phonon properties, such as formation energy and mobility in the dilute limit⁴³. Perturbation of the structure is necessary, as while simply adding an extra charge (an unpaired electron or hole) to an unperturbed supercell and allowing the system to relax to its local minimum may in some cases result in spontaneous localisation of an electron/hole polaron, typically the structure must first be biased towards a polaronic configuration by distorting atoms around likely polaron sites before localization can occur. Intuitively one can think of this as overcoming the energy barrier between delocalized and localized solutions. These distorted structures are relaxed to their local minima in turn, and by comparing the energies of unperturbed and distorted (perturbed) structures, we

can assess whether energy minimising distortions are present in the system. By inspecting the charge of the band containing the unpaired electron or hole in each relaxed structure, we can assess whether distortions result in a localized (0D) polaronic solution or otherwise. From our results using 64 atom supercells, we conclude that polaronic states in CuSbSe_2 occupy a volume $>1555.52 \text{ \AA}^3$, which is four times the volume of a unit cell volume and would therefore be classified as large polarons. These calculations were run at HSE06 level to negate self-interaction errors, which may have otherwise resulted in spurious delocalized solutions. Localization was assessed qualitatively by inspecting the partial charge density function of the band containing the unpaired electron or hole, with all solutions being $>0\text{D}$ by inspection.

For the reviewer's convenience, we show below Supplementary Fig. 8:

Supplementary Fig. 8 | Computations to directly determine whether small polarons form in CuSbSe_2 . **a**, Addition of an unpaired hole followed by distortion around the Cu, Se1 and Se2 sites and/or rattling of the lattice results in a relatively small change in DFT total energy of about ~ 12 meV compared to the unperturbed state. Cu and the two Se sites were selected, since the valence band is dominated by these states (refer to Fig. 2b in the main text). The partial electron density function of the unpaired hole for the **b**, unperturbed and the **c**, perturbed structures show a quasi-2D valence band as

the electron density is confined to a single layer. **d**, Addition of an unpaired electron followed by distortion around the Sb site results in the formation of a metastable state 0.12 eV above the unperturbed structure. Analysis was performed on distortions around Sb because Sb 5p orbitals dominate the lower conduction band (refer to Fig. 2b in the main text). The partial electron density of the unpaired electron in the **e**, unperturbed structure shows a delocalised 3D solution, while the higher energy **f**, perturbed structure shows quasi-1D character because the electron density is confined to ribbons running through the interlayer space. In this analysis, we considered a supercell that was four times the volume of a unit cell, and we did not observe any localization of either the electron or hole charge density (0D character) to within one unit cell. That is, these computations support conclusions that small polarons do not form in CuSbSe₂

Therefore, we have strong evidence, from multiple methods, that show delocalized charge-carriers exist in CuSbSe₂.

4. The stability of CuSbSe₂, especially under operational conditions relevant to solar cells, is not thoroughly explored. Given the focus on earth-abundant and environmentally friendly materials, understanding long-term stability in diverse environmental conditions is essential.

Response 4.7: This is certainly an important point, and as part of the previous revision, in response to comments from Reviewer 2, we conducted stability measurements of CuSbSe₂ stored in ambient air without encapsulation (room temperature, ~80% relative humidity for 3 weeks). This, of course, is highly relevant to solar cells under operation. Solar cells are encapsulated, but materials which are stable without encapsulation hold promise for exhibiting long-term stability. We examined the stability of the oxidation state of the species present, especially Cu(I). For the reviewer's convenience, we copy out key parts of our paper that we added on the topic of stability.

Main text, page 14:

“Finally, we note that CuSbSe₂ contains Cu in the +1 oxidation state, whereas the +2 oxidation state is typically more thermodynamically stable under ambient conditions. We therefore examined the chemical stability of the cation species in the optimized CuSbSe₂ films by X-ray photoelectron spectroscopy, as detailed in Supplementary Note 3. We found from the Cu 2p core levels and LMM Auger peaks that Cu remained in the +1 oxidation state after storage in ambient air (with approximately 80% relative humidity) for 3 weeks, and Sb also remained in the +3 oxidation state. However, we found that a layer of oxide (likely Sb₂O₃) formed on the surface of the films after storage in air, whereas there was no evidence of cuprous oxide or hydroxide species, showing Cu(I) to remain stable in its tetrahedral environment in the structure.”

Key plots from Supplementary Note 3 (the reviewer is encouraged to read all of what we wrote in this section):

Supplementary Fig. 3 | XPS survey spectra of CuSbSe₂ thin films. Measurements for **a**, as-prepared and **b**, aged CuSbSe₂ samples.

Supplementary Fig. 4 | XPS spectra of CuSbSe₂ thin films (as-prepared and aged). **a**, Cu 2p core levels and **b**, Cu LMM Auger spectra of as-prepared (black line) and aged CuSbSe₂ samples (red line).

Supplementary Fig. 5 | XPS spectra of CuSbSe₂ thin films for Sb 3d. Measurements and fitting for **a**, as-prepared and **b**, aged CuSbSe₂ samples.

To complement these XPS measurements of chemical stability, we have now measured and added in the diffraction measurements of CuSbSe₂ thin films at the beginning and end of a 3-week period stored

in ambient air (refer to Supplementary Fig. 6a, copied below). Here, we found the phase to remain unchanged, as expected.

At the same time, we recognize the reviewer's point regarding "diverse environmental conditions". We therefore conducted another stability test, this time at 85 °C and 85% relative humidity, and under 1-sun illumination. These are conditions that solar cells need to be exposed to in accelerated degradation testing. We previously performed bulk phase stability measurements under identical conditions for triple-cation lead-halide perovskite thin films⁴⁴, meaning that we can put the stability of CuSbSe₂ in context. Whereas the halide perovskite film degraded after just 8 h (refer to Fig. 1 in Ref. 44), CuSbSe₂ remained unchanged over the entire 24 h test.

The high stability of CuSbSe₂ without encapsulation is highly promising for their application in photovoltaics. We have added the following to Supplementary Note 3:

Structural Stability: We also examined the phase stability of the CuSbSe₂ thin films in diverse environments. For the stability test in an ambient environment, as-prepared and aged samples were prepared as described above for XPS measurements, and their XRD patterns were obtained. As shown in Supplementary Fig. 6a, after 3 weeks of storage in an ambient environment (room temperature and approximately 80% relative humidity), the XRD pattern of the CuSbSe₂ thin film was unchanged, showing the material to be phase-stable without encapsulation.

Another stability test was conducted at 85 °C and 85% relative humidity, and under 1-sun illumination. The details of the experimental methods can be found in our previous publication⁴¹. CuSbSe₂ thin films were kept under such conditions for 3, 8 and 24 hours, respectively. The macroscopic appearance and XRD pattern of the CuSbSe₂ thin films were not changed after 24-hour exposure to this condition (Supplementary Fig. 6b). CuSbSe₂ therefore shows better stability under these conditions than triple-cation perovskite thin films, which began showing degradation products after only 8 h⁴¹.

Supplementary Fig. 6 | Phase stability of CuSbSe₂ thin films in diverse environments. a, Comparison of the XRD patterns of as-prepared (red) and aged CuSbSe₂ films (stored at room temperature with approximately 80% relative humidity for 3 weeks, purple) with the CuSbSe₂ reference pattern (black, ICSD database, coll. code 418754). Main peaks are indicated by dashed lines along with the corresponding Miller indices. **b,** Appearance and XRD patterns of CuSbSe₂ thin films under 1-sun illumination, at 85 °C and 85% relative humidity over 24 h. Main peaks are indicated by dashed lines along with the corresponding Miller indices. The XRD peak from the Si substrate is indicated.

We also added the following to the main text on pages 14 and 15:

“We also found CuSbSe₂ to be phase stable in ambient air over this 3-week period (Supplementary Fig. 6a), and is also more stable under damp-heat conditions (85% relative humidity, 85 °C temperature, and under 1-sun illumination) than lead-halide perovskites (Supplementary Fig. 6b).”

5. While the work identifies mechanisms to avoid carrier localization in CuSbSe₂, it still acknowledges the prevalence of carrier localization in many pnictogen-based semiconductors. This limitation suggests that achieving truly delocalized charge-carriers remains a significant challenge, possibly limiting the broad applicability of the findings to other materials within this class. Furthermore, the unique structural features of CuSbSe₂, such as the stereochemical activity of the lone pair on Sb³⁺, might not be present in other materials, limiting the generalizability of the results.

Response 4.8: Our findings on the chemical-physical factors behind the band-like transport in CuSbSe₂ are important new insights for the field (as explained in detail in Response 4.1), but at the same time are grounded in the fundamentals, and the key structural and electronic features can be found in other materials (detailed below). As such, we would fully expect our findings to be generalizable.

Thus far, the community working on heavy pnictogen-based PIMs have primarily focussed on materials with low electronic dimensionality (e.g., Cs₂AgBiBr₆ and Cs₂AgSbBr₆, which are structurally 3D, but electronically 0D because of the mismatch in orbitals at the band extrema in alternating octahedra), high exciton binding energy (e.g., BiI₃) or high levels of cation disorder (e.g., AgBiS₂). We believe that these are the underlying reasons for the prevalence of carrier localization in the PIMs investigated thus far. As we explained in Response 4.1, it was only in mid-2023 that groups started reporting heavy-pnictogen-based PIMs with purely band-like transport, but the underlying reasons were not understood. Our work on CuSbSe₂ is an important start in filling these important gaps in the knowledge of the field. Having investigated this material in depth, we now understand that compounds comprised of thick layers benefit from relaxing distortions to the interlayer gaps; having polyhedra oriented at an angle to the principal axes allow distortions to be manifest as changes in bond angle rather than bond length. Both factors minimize deformation potentials. At the same time, forming quasi-bonds between layers can enhance electronic dimensionality, reducing localization. All of these features can be found in CuSbS₂, CuBiS₂ and CuBiSe₂, which we would thus expect to also exhibit band-like transport.

The reviewer makes an interesting point about the stereochemical activity of the lone pair on Sb in CuSbSe₂. The main benefits of a second-order Jahn-Teller distortion in the case of CuSbSe₂ is to firstly help to prevent cation disorder between Cu and Sb, and secondly contribute to forming a layered structure that still has Sb in one layer quasi-bonding to Se in the neighbouring layer, allowing a higher electronic dimensionality.

This stereochemical activity of Sb³⁺ (and indeed of other pnictogens) can be found in other materials. For example, CuSbS₂⁴⁵, CuBiS₂⁴⁶ and CuBiSe₂⁴⁷ all exhibit stereochemical activity of the pnictogen, resulting in the pnictogen and chalcogen occurring across the interlayer gaps, with quasi-bonding likely occurring between them. More generally, 1D materials, such as Sb₂S₃ and Sb₂Se₃ can also exhibit stereochemical activity of the pnictogen, which takes place when there is sufficient energy matching in the valence orbitals of the cation and anion (e.g., stereochemical activity of the valence lone pair takes place in PbO, but not PbS⁴⁸).

To capture this point on stereochemical activity, we have added the following to the future work section of the discussion on page 38:

“At the same time quasi-bonding across these regular gaps between species contributing to the band-edge density of states is important for increasing the electronic dimensionality, which reduces the likelihood of self-trapping. This could be found more generally, for example, in materials that exhibit stereochemical activity of the pnictogen cation (e.g., CuSbS₂¹²⁷, CuBiS₂⁷⁰ and CuBiSe₂¹²⁸), resulting in a layered structure, with both the pnictogen and chalcogen placed about the interlayer gaps, allowing quasi-bonding to take place between them.”

As discussed in Response 4.2, we share with the reviewer the vision to ultimately design PIMs with band-like transport from simple descriptors. The next step will be to test the robustness of the design principles obtained from investigating the specific case of CuSbSe₂ on other, related materials. We have already begun this effort, and are investigating CuBiS₂, CuBiSe₂ and Cu₃BiSe₃. In addition to similar layered materials, we can extend the principles found here to hypothesize that band-like transport could also be found in materials with quasi-1D structures, provided that they have 2D or 3D electronic dimensionality. From the Toyozawa model, the likelihood of carrier localization increases as the dimensionality is reduced. But we emphasize that it is electronic dimensionality rather than structural dimensionality that matters. Traditionally, the community has assumed that both are the same. For example, prior work on antimony chalcogenides quoted the fact that these are quasi-1D materials as being a factor in their exhibiting facile self-trapping¹³. But our work here has shown that electronic dimensionality can substantially deviate from the structural dimensionality, and could be different in the conduction and valence bands.

[Data redacted for confidentiality reasons. We intend to publish this data separately]

Thus, our work on CuSbSe₂ makes important insights that change the future direction of the field, demonstrating its impact. Furthermore, in the current work, we have extensively developed a range of advanced computational methodologies to thoroughly study carrier localization, and we can directly apply these towards future efforts on the broader materials space.

To capture these points, we have modified the future work part of the discussion on page 39:

“We believe that these insights, gained from investigating CuSbSe₂, are generalizable because the key structural and electronic features can be found in other materials, and the ways in which they affect carrier localization are rationalized based on the fundamentals of electron-phonon coupling theory (rather than bespoke theory specific to only CuSbSe₂).”

6. Although the localization effect is minimal in CuSbSe₂, but the macroscopic mobility is still much lower for example compared to single crystalline MAPbI₃. For application in solar cells, this is a crucial aspect that needs to be addressed for CuSbSe₂.

Response 4.9: We note that the samples investigated experimentally are polycrystalline thin films. Single-crystalline MAPbI₃ has orders of magnitude higher mobility than polycrystalline MAPbI₃. It is well established that many scattering effects, due to grain boundaries and structural defects, limit mobilities in polycrystalline materials. Certainly, future efforts focussed on practical applications of CuSbSe₂ could focus on improving the grain size. When it comes to single-crystalline CuSbSe₂, charge-carrier mobilities of 87 cm² V⁻¹ s⁻¹ have been reported⁴⁹, which is similar to the values reported for single crystal BiOI (83 cm² V⁻¹ s⁻¹; which also exhibits delocalized charge-carriers), and substantially higher than the values for Cs₂AgBiBr₆ double perovskites (11 cm² V⁻¹ s⁻¹; which exhibits small polarons).

As part of the previous revision, we added a comparison of the mobilities of CuSbSe₂ to other PIMs. For the convenience of the reviewer, we have copied out this discussion in the main text, as well as the table we added to the Supplementary Information:

“Importantly, the mobilities measured in these CuSbSe₂ films are higher than typically found for small polarons. Hall effect measurements gave a macroscopic mobility of 1.01±0.01 cm²·V⁻¹·s⁻¹ at room

temperature (Fig. 3b, 300 K), while the delocalized intra-grain mobility value extracted from the initial photoconductivity of OPTP measurements was $4.7 \pm 0.2 \text{ cm}^2 \cdot \text{V}^{-1} \cdot \text{s}^{-1}$ (refer to Supplementary Note 5 for details of how the mobility was determined from OPTP measurements). This difference in mobility values is due to the different length scales of Hall effect and OPTP measurements. Hall effect measurements investigate charge carrier transport throughout the whole sample, while the mobility extracted from the initial OPTP signal represents the transport within a shorter range, usually well within one grain^{25,27,28,48}.

Even though the mobilities values extracted over different length scales are different, comparing mobility values obtained from different materials over the same length scale is informative to put the nature of charge-carrier transport in context. BiOI, as another pnictogen-based compound which can avoid charge-carrier localization, has a peak OPTP mobility of $\sim 3 \text{ cm}^2 \cdot \text{V}^{-1} \cdot \text{s}^{-1}$ at 295 K for polycrystalline samples. In single crystals, the mobilities for BiOI obtained from time-of-flight measurements are $26 \text{ cm}^2 \cdot \text{V}^{-1} \cdot \text{s}^{-1}$ (out-of-plane) and $83 \text{ cm}^2 \cdot \text{V}^{-1} \cdot \text{s}^{-1}$ (in-plane)³⁹. By contrast, materials undergoing carrier localization (e.g., NaBiS₂, Cs₂AgSbBr₆, Cs₂AgBiBr₆ and AgBiS₂ without heat treatment) exhibit a substantial reduction in mobility from an initial delocalized state to localized mobilities in the range of $0.03\text{--}1.3 \text{ cm}^2 \cdot \text{V}^{-1} \cdot \text{s}^{-1}$ 2 ps after excitation, as obtained from OPTP measurements (Supplementary Table 1). These OPTP mobilities are all lower than that for CuSbSe₂, despite CuSbSe₂ having higher effective masses (Supplementary Table 1). This is consistent with the delocalized nature of charge-carriers in CuSbSe₂, and emphasizes the importance of avoiding carrier localization.”

Supplementary Table 1 | Comparison of key properties of CuSbSe₂ with other Sb- and Bi-based compounds. These are the charge-carrier mobility at room temperature, effective mass, bandgap and dielectric constants, along with the time taken for the photoconductivity signals to decay by 50% from the initial peak value. The charge-carrier mobilities shown are extracted from the photoconductivity spectra measured by OPTP.

Material	OPTP Mobility ($\text{cm}^2 \cdot \text{V}^{-1} \cdot \text{s}^{-1}$)	Effective mass (m_h^*/m_0 ; m_e^*/m_0)	Bandgap (eV)	Dielectric constant (ϵ_∞ ; ϵ_{stat})	Time taken to reach 50% photoconductivity decay (ps)
Polycrystalline CuSbSe ₂ (this work)	4.7 ± 0.2	1.60; 0.43	0.9-1.2	11.3; 23.0	6.7
Single crystal CuSbSe ₂ ¹	87 (measured by Hall effect measurements)				
NaBiS ₂ ²	0.29 (delocalized); 0.03 (localized)	1.04; 0.24	1.4	$\epsilon_\infty = 8.1$; $\epsilon_{stat} = 43.7$	~ 0.5
AgBiS ₂ ^{3,4}	As-prepared: 0.43 ± 0.05 (delocalized), 0.11 ± 0.05 (localized); annealed: 2.70 ± 0.10 (delocalized), 2.20 ± 0.10 (localized)	0.51; 0.24	1.45	19.43(x/y), 12.44(z); 115.61(x/y), 35.08(z)	<1 (as-prepared); 20-30 (heat treatment)
Cs ₂ AgSbBr ₆ ^{5,6}	0.5 (delocalized); 0.1 (localized)	0.234-0.969; 0.289-0.431	1.64	4.82; 13.69	1-2

$\text{Cs}_2\text{AgBiBr}_6^{7-11}$	3 (delocalized); 1.3 (localized)	0.14; 0.37	2-2.25	4.60; 12.76	1-2
Polycrystalline $\text{BiOI}^{12,13}$	~3	0.26; 0.23	1.93	8.60; 43.33	200-300
Single crystal BiOI^{12}	26 (perpendicular), 83 (parallel), measured by time of flight measurements				
$\text{Sb}_2\text{S}_3^{14-16}$	0.9 ± 0.1	0.64; 0.40	1.7-1.8	11.55(x), 10.97(y), 8.25(z); 98.94(x), 94.21(y), 13.14(z)	10-20
$\text{MA}_3\text{Bi}_2\text{I}_9^{17-19}$	Not reported	0.95; 0.54	2.1-2.2	5.43(x), 4.67 (z); 39.89(x), 9.62(z)	Not reported
$\text{Cs}_3\text{Bi}_2\text{I}_9^{20,21}$	Not reported	0.94 (perpendicular ar to (100)), 2.14 (perpendicular ar to (001)); 0.33 (perpendicular ar to (100)), 3.22 (perpendicular ar to (001))	1.95	5.38 (perpendicular ar to (100)), 4.09 (perpendicular ar to (001)); 9.90 (perpendicular ar to (100)), 7.85 (perpendicular ar to (001))	Not reported
BiSI^{22-25}	Not reported	0.95; 0.53	1.57	8.03; 37.81	Not reported
BiI_3^{26-29}	Not reported	2.01; 0.68	1.8	7.1 (in plane), 6.4 (out of plane); 54 (in plane), 8.6 (out of plane)	Not reported

We recognize the reviewer's point, and have added the following to the end of the discussion to capture the point on the need to optimize thin film processing to further improve mobilities in polycrystalline thin films (page 39):

“On a more practical level specific to CuSbSe_2 , whilst the charge-carrier mobilities ($1.01 \pm 0.01 \text{ cm}^2 \cdot \text{V}^{-1} \cdot \text{s}^{-1}$ at room temperature) exceed those found for polycrystalline pnictogen-based semiconductors with self-trapping (Supplementary Table 1), they fall below the highest values achieved in single crystal CuSbSe_2 ($87 \text{ cm}^2 \cdot \text{V}^{-1} \cdot \text{s}^{-1}$)⁴⁴. Given also that the intra-grain local mobility measured by OPTP exceeds the macroscopic mobility (as explained earlier), the charge-carrier mobilities of the CuSbSe_2 polycrystalline thin films prepared in these work are likely limited by grain boundary or structural defect scattering. Future efforts should therefore focus on reducing the density of these structural defects through processing or post-processing strategies.”

References

- 1 Brandt, R. E., Stevanović, V., Ginley, D. S. & Buonassisi, T. Identifying defect-tolerant semiconductors with high minority-carrier lifetimes: beyond hybrid lead halide perovskites. *MRS Communications* **5**, 265-275 (2015). <https://doi.org/10.1557/mrc.2015.26>
- 2 Brandt, R. E. *et al.* Searching for “Defect-Tolerant” Photovoltaic Materials: Combined Theoretical and Experimental Screening. *Chemistry of Materials* **29**, 4667-4674 (2017). <https://doi.org/10.1021/acs.chemmater.6b05496>
- 3 Wright, A. D. *et al.* Ultrafast Excited-State Localization in Cs₂AgBiBr₆ Double Perovskite. *J Phys Chem Lett* **12**, 3352-3360 (2021). <https://doi.org/10.1021/acs.jpcclett.1c00653>
- 4 Righetto, M. *et al.* Alloying Effects on Charge-Carrier Transport in Silver–Bismuth Double Perovskites. *The Journal of Physical Chemistry Letters* **14**, 10340-10347 (2023). <https://doi.org/10.1021/acs.jpcclett.3c02750>
- 5 Buizza, L. R. V. *et al.* Interplay of Structure, Charge-Carrier Localization and Dynamics in Copper-Silver-Bismuth-Halide Semiconductors. *Advanced Functional Materials* **32**, 2108392 (2022). <https://doi.org/https://doi.org/10.1002/adfm.202108392>
- 6 Buizza, L. R. V. *et al.* Charge-Carrier Mobility and Localization in Semiconducting Cu₂AgBiI₆ for Photovoltaic Applications. *ACS Energy Lett* **6**, 1729-1739 (2021). <https://doi.org/10.1021/acsenerylett.1c00458>
- 7 Lal, S. *et al.* The Role of Chemical Composition in Determining the Charge-Carrier Dynamics in (AgI)_x(BiI₃)_y Rudorffites. *Advanced Functional Materials* **34**, 2315942 (2024). <https://doi.org/https://doi.org/10.1002/adfm.202315942>
- 8 McCall, K. M., Stoumpos, C. C., Kostina, S. S., Kanatzidis, M. G. & Wessels, B. W. Strong Electron–Phonon Coupling and Self-Trapped Excitons in the Defect Halide Perovskites A₃M₂I₉ (A = Cs, Rb; M = Bi, Sb). *Chemistry of Materials* **29**, 4129-4145 (2017). <https://doi.org/10.1021/acs.chemmater.7b01184>
- 9 Jin, J. *et al.* Octahedral Distortion and Excitonic Behavior of Cs₃Bi₂Br₉ Halide Perovskite at Low Temperature. *The Journal of Physical Chemistry C* **127**, 3523-3531 (2023). <https://doi.org/10.1021/acs.jpcc.2c07642>
- 10 Scholz, M., Oum, K. & Lenzer, T. Pronounced exciton and coherent phonon dynamics in BiI₃. *Physical Chemistry Chemical Physics* **20**, 10677-10685 (2018). <https://doi.org/10.1039/C7CP07729G>
- 11 Righetto, M. *et al.* Cation-Disorder Engineering Promotes Efficient Charge-Carrier Transport in AgBiS₂ Nanocrystal Films. *Adv Mater*, e2305009 (2023). <https://doi.org/10.1002/adma.202305009>
- 12 Huang, Y. T. *et al.* Strong absorption and ultrafast localisation in NaBiS₂ nanocrystals with slow charge-carrier recombination. *Nat Commun* **13**, 4960 (2022). <https://doi.org/10.1038/s41467-022-32669-3>
- 13 Yang, Z. *et al.* Ultrafast self-trapping of photoexcited carriers sets the upper limit on antimony trisulfide photovoltaic devices. *Nat Commun* **10**, 4540 (2019). <https://doi.org/10.1038/s41467-019-12445-6>
- 14 Tao, W. *et al.* Coupled Electronic and Anharmonic Structural Dynamics for Carrier Self-Trapping in Photovoltaic Antimony Chalcogenides. *Adv Sci (Weinh)* **9**, e2202154 (2022). <https://doi.org/10.1002/advs.202202154>
- 15 Wu, B. *et al.* Strong self-trapping by deformation potential limits photovoltaic performance in bismuth double perovskite. *Sci Adv* **7**, eabd3160 (2021). <https://doi.org/10.1126/sciadv.abd3160>

- 16 Jia, Z. *et al.* Charge-Carrier Dynamics of Solution-Processed Antimony- and Bismuth-Based Chalcogenide Thin Films. *ACS Energy Letters* **8**, 1485-1492 (2023). <https://doi.org/10.1021/acsenerylett.3c00140>
- 17 Kentsch, R. *et al.* Exciton Dynamics and Electron–Phonon Coupling Affect the Photovoltaic Performance of the Cs₂AgBiBr₆ Double Perovskite. *The Journal of Physical Chemistry C* **122**, 25940-25947 (2018). <https://doi.org/10.1021/acs.jpcc.8b09911>
- 18 Wang, Y. *et al.* Cation disorder engineering yields AgBiS₂ nanocrystals with enhanced optical absorption for efficient ultrathin solar cells. *Nature Photonics* **16**, 235-241 (2022). <https://doi.org/10.1038/s41566-021-00950-4>
- 19 Liu, C. *et al.* Asynchronous Photoexcited Electronic and Structural Relaxation in Lead-Free Perovskites. *Journal of the American Chemical Society* **141**, 13074-13080 (2019). <https://doi.org/10.1021/jacs.9b04557>
- 20 Jagt, R. A. *et al.* Layered BiOI single crystals capable of detecting low dose rates of X-rays. *Nat Commun* **14**, 2452 (2023). <https://doi.org/10.1038/s41467-023-38008-4>
- 21 Rondiya, S. R., Jagt, R. A., MacManus-Driscoll, J. L., Walsh, A. & Hoye, R. L. Z. Self-trapping in bismuth-based semiconductors: Opportunities and challenges from optoelectronic devices to quantum technologies. *Applied Physics Letters* **119**, 220501 (2021). <https://doi.org/10.1063/5.0071763>
- 22 Lal, S. *et al.* Bandlike Transport and Charge-Carrier Dynamics in BiOI Films. *J Phys Chem Lett* **14**, 6620-6629 (2023). <https://doi.org/10.1021/acs.jpcclett.3c01520>
- 23 Wei, S. H. & Zunger, A. Predicted band-gap pressure coefficients of all diamond and zinc-blende semiconductors: Chemical trends. *Physical Review B* **60**, 5404-5411 (1999). <https://doi.org/10.1103/PhysRevB.60.5404>
- 24 Ganose, A. M. *et al.* Efficient calculation of carrier scattering rates from first principles. *Nat Commun* **12**, 2222 (2021). <https://doi.org/10.1038/s41467-021-22440-5>
- 25 Bhuyan, S., Jindal, V., Jana, D. & Ghosh, S. Signatures of self-trapping of trions in monolayer MoS₂. *Journal of Physics D: Applied Physics* **51**, 435102 (2018). <https://doi.org/10.1088/1361-6463/aadfc2>
- 26 Kang, M. *et al.* Holstein polaron in a valley-degenerate two-dimensional semiconductor. *Nature Materials* **17**, 676-680 (2018). <https://doi.org/10.1038/s41563-018-0092-7>
- 27 Alsaleh, N. M., Singh, N. & Schwingenschlögl, U. Role of interlayer coupling for the power factor of CuSbS₂ and CuSbSe₂. *Physical Review B* **94**, 125440 (2016). <https://doi.org/10.1103/PhysRevB.94.125440>
- 28 Wang, D.-D., Gong, X.-G. & Yang, J.-H. Unusual interlayer coupling in layered Cu-based ternary chalcogenides CuMCh₂ (M = Sb, Bi; Ch = S, Se). *Nanoscale* **13**, 14621-14627 (2021). <https://doi.org/10.1039/D1NR04045F>
- 29 Yu, Y. *et al.* Exciton-dominated Dielectric Function of Atomically Thin MoS₂ Films. *Scientific Reports* **5**, 16996 (2015). <https://doi.org/10.1038/srep16996>
- 30 Buizza, L. R. V. & Herz, L. M. Polarons and Charge Localization in Metal-Halide Semiconductors for Photovoltaic and Light-Emitting Devices. *Adv Mater* **33**, e2007057 (2021). <https://doi.org/10.1002/adma.202007057>
- 31 Hoye, R. L. Z. *et al.* The Role of Dimensionality on the Optoelectronic Properties of Oxide and Halide Perovskites, and their Halide Derivatives. *Advanced Energy Materials* **12**, 2100499 (2021). <https://doi.org/10.1002/aenm.202100499>
- 32 Ming, W., Shi, H. & Du, M.-H. Large dielectric constant, high acceptor density, and deep electron traps in perovskite solar cell material CsGeI₃. *Journal of Materials Chemistry A* **4**, 13852-13858 (2016). <https://doi.org/10.1039/C6TA04685A>

- 33 Jain, A. *et al.* Commentary: The Materials Project: A materials genome approach to accelerating materials innovation. *APL Materials* **1**, 011002 (2013). <https://doi.org/10.1063/1.4812323>
- 34 Naik, A. A., Ertural, C., Dhamrait, N., Benner, P. & George, J. A Quantum-Chemical Bonding Database for Solid-State Materials. *Scientific Data* **10**, 610 (2023). <https://doi.org/10.1038/s41597-023-02477-5>
- 35 Mosquera-Lois, I., Kavanagh, S. R., Ganose, A. M. & Walsh, A. Machine-learning structural reconstructions for accelerated point defect calculations. *npj Computational Materials* **10**, 121 (2024). <https://doi.org/10.1038/s41524-024-01303-9>
- 36 Teresa, J. M. D. *et al.* Evidence for magnetic polarons in the magnetoresistive perovskites. *Nature* **386**, 256-259 (1997). <https://doi.org/10.1038/386256a0>
- 37 Millis, A. J., Mueller, R. & Shraiman, B. I. Fermi-liquid-to-polaron crossover. II. Double exchange and the physics of colossal magnetoresistance. *Physical Review B* **54**, 5405-5417 (1996). <https://doi.org/10.1103/PhysRevB.54.5405>
- 38 Srivastava, C. M., Srivastava, N. B., Singh, L. N. & Bahadur, D. Small polaron transport and colossal magnetoresistance in $\text{La}_{2/3}\text{Ca}_{1/3}\text{MnO}_3$. *Journal of Applied Physics* **105** (2009). <https://doi.org/10.1063/1.3123764>
- 39 Salamon, M. B. & Jaime, M. The physics of manganites: Structure and transport. *Reviews of Modern Physics* **73**, 583-628 (2001). <https://doi.org/10.1103/RevModPhys.73.583>
- 40 Banerjee, A. & Paul, G. Room-temperature Magnetoresistance in Hybrid Halide Perovskites: Effect of Spin-Orbit Coupling. *Physical Review Applied* **14**, 064018 (2020). <https://doi.org/10.1103/PhysRevApplied.14.064018>
- 41 Franchini, C., Reticcioli, M., Setvin, M. & Diebold, U. Polarons in materials. *Nature Reviews Materials* **6**, 560-586 (2021). <https://doi.org/10.1038/s41578-021-00289-w>
- 42 Mosquera-Lois, I., Kavanagh, S. R., Walsh, A. & Scanlon, D. O. Identifying the ground state structures of point defects in solids. *npj Computational Materials* **9**, 25 (2023). <https://doi.org/10.1038/s41524-023-00973-1>
- 43 Pham, T. D. & Deskins, N. A. Efficient Method for Modeling Polarons Using Electronic Structure Methods. *Journal of Chemical Theory and Computation* **16**, 5264-5278 (2020). <https://doi.org/10.1021/acs.jctc.0c00374>
- 44 Guo, X. *et al.* Air-stable bismuth sulfobromide (BiSBr) visible-light absorbers: optoelectronic properties and potential for energy harvesting. *Journal of Materials Chemistry A* **11**, 22775-22785 (2023). <https://doi.org/10.1039/D3TA04491B>
- 45 Yang, B. *et al.* CuSbS_2 as a Promising Earth-Abundant Photovoltaic Absorber Material: A Combined Theoretical and Experimental Study. *Chemistry of Materials* **26**, 3135-3143 (2014). <https://doi.org/10.1021/cm500516v>
- 46 Dufton, J. T. *et al.* Structural and electronic properties of CuSbS_2 and CuBiS_2 : potential absorber materials for thin-film solar cells. *Phys Chem Chem Phys* **14**, 7229-7233 (2012). <https://doi.org/10.1039/c2cp40916j>
- 47 Förster, H.-J., Bindi, L. & Stanley, C. J. Grundmannite, CuBiSe_2 , the Se-analogue of emplectite, a new mineral from the El Dragón mine, Potosí, Bolivia. *European Journal of Mineralogy* **28**, 467-477 (2016). <https://doi.org/10.1127/ejm/2016/0028-2513>
- 48 Walsh, A., Payne, D. J., Egdell, R. G. & Watson, G. W. Stereochemistry of post-transition metal oxides: revision of the classical lone pair model. *Chemical Society Reviews* **40**, 4455-4463 (2011). <https://doi.org/10.1039/C1CS15098G>
- 49 Chen, T. *et al.* Ultralow Thermal Conductivity and Enhanced Figure of Merit for CuSbSe_2 via Cd-Doping. *ACS Applied Energy Materials* **4**, 1637-1643 (2021). <https://doi.org/10.1021/acsaem.0c02820>